# Learning a Fast Mixing Exogenous Block MDP using a Single Trajectory

**Alexander Levine**[*]
The University of
Texas at Austin

**Peter Stone**
The University of Texas at Austin
Sony AI

**Amy Zhang**
The University of
Texas at Austin

## Abstract

In order to train agents that can quickly adapt to new objectives or reward functions, efficient unsupervised representation learning in sequential decision-making environments can be important. Frameworks such as the Exogenous Block Markov Decision Process (Ex-BMDP) have been proposed to formalize this representation-learning problem (Efroni et al., 2022b). In the Ex-BMDP framework, the agent's high-dimensional observations of the environment have two latent factors: a *controllable* factor, which evolves deterministically within a small state space according to the agent's actions, and an *exogenous* factor, which represents time-correlated noise, and can be highly complex. The goal of the representation learning problem is to learn an encoder that maps from observations into the controllable latent space, as well as the dynamics of this space. Efroni et al. (2022b) has shown that this is possible with a sample complexity that depends only on the size of the *controllable* latent space, and not on the size of the noise factor. However, this prior work has focused on the episodic setting, where the controllable latent state resets to a specific start state after a finite horizon.

By contrast, if the agent can only interact with the environment in a single continuous trajectory, prior works have not established sample-complexity bounds. We propose **STEEL**, the first provably sample-efficient algorithm for learning the controllable dynamics of an Ex-BMDP from a single trajectory, in the function approximation setting. STEEL has a sample complexity that depends only on the sizes of the *controllable* latent space and the encoder function class, and (at worst linearly) on the *mixing time* of the exogenous noise factor. We prove that STEEL is correct and sample-efficient, and demonstrate STEEL on two toy problems. Code is available at: `https://github.com/midi-lab/steel`.

## 1 Introduction

This work considers the *unsupervised representation learning* problem in sequential control environments. Suppose an agent (e.g., a robot) is able to make observations and take actions in an environment for some period of time, but does not yet have an externally-defined task to accomplish. We want the agent to learn a *model* of the environment that may be useful for many downstream tasks: the question is then how to efficiently explore the environment to learn such a model.

Sequential decision-making tasks are often modeled as *Markov Decision Processes* (MDPs). In the unsupervised setting, an MDP consists of a set of possible observations $\mathcal{X}$, a set of possible actions $\mathcal{A}$, a distribution over initial observations $\pi_0 \in \mathcal{P}(\mathcal{X})$, and a transition function $\mathcal{T} : \mathcal{X} \times \mathcal{A} \to \mathcal{P}(\mathcal{X})$. The agent does not have direct access to $\mathcal{T}$. Instead, at each timestep $t$, the agent observes $x_t \in \mathcal{X}$ and selects action $a_t$. The next observation $x_{t+1}$ is then sampled as $x_{t+1} \sim \mathcal{T}(x_t, a_t)$.

In the totally generic MDP setting, the only model learning possible is to directly learn the transition function $\mathcal{T}$. However, if the space of possible observations is large, this task becomes intractable. Therefore, prior works have attempted to simplify the problem by assuming that the MDP has some underlying structure, which a learning algorithm can exploit. One such structural assumption is the Ex-BMDP (Exogenous Block MDP) framework, introduced by Efroni et al. (2022b). The Ex-

---

[*]Correspondence to: alevine0@cs.utexas.edu

BMDP framework captures situations where the space of observations $\mathcal{X}$ is very large, but the parts of the environment that the agent has control over can be represented by a much smaller latent state.

An Ex-BMDP has an observation space $\mathcal{X}$, a controllable (or *endogenous*) latent state space $\mathcal{S}$, and an exogenous state space $\mathcal{E}$. In the version of this setting that we consider, the controllable state $s_t \in \mathcal{S}$ evolves *deterministically* according to a latent transition function $T : \mathcal{S} \times \mathcal{A} \rightarrow \mathcal{S}$. That is: $s_{t+1} = T(s_t, a_t)$. The exogenous state $e_t \in \mathcal{E}$ represents *time correlated noise*: it evolves stochastically, *independently of actions*, according to a transition function $\mathcal{T}_e : \mathcal{E} \rightarrow \mathcal{P}(\mathcal{E})$. That is: $e_{t+1} \sim \mathcal{T}_e(e_t)$. Neither $s$ nor $e$ is directly observed. Instead, the observation $x_t$ is sampled as $x_t \sim \mathcal{Q}(s_t, e_t)$, where $\mathcal{Q} \in \mathcal{S} \times \mathcal{E} \rightarrow \mathcal{P}(\mathcal{X})$ is the *emission function*. We make a *block* assumption on $\mathcal{Q}$ with respect to $\mathcal{S}$: that is, we assume that for distinct latent states $s, s' \in \mathcal{S}$, the sets of possible observations that can be sampled from $\mathcal{Q}(s, \cdot)$ and $\mathcal{Q}(s', \cdot)$ are disjoint. In other words, there exists a deterministic partial inverse of $\mathcal{Q}$, which is $\phi^* : \mathcal{X} \rightarrow \mathcal{S}$, such that if $x \sim \mathcal{Q}(s, e)$, then $\phi^*(x) = s$. Hence, it is always possible in principle to infer $s$ from $x$.[1]

As in the general MDP setting, the agent only directly observes $x_t \in \mathcal{X}$, and chooses actions $a_t$ in response. However, rather than attempting to learn the full transition dynamics $\mathcal{T}$ of the system (which is determined by $T$, $\mathcal{T}_e$, and $\mathcal{Q}$ together), the objective of the agent is instead to efficiently model only the *latent encoder* $\phi^*$ and the *latent transition function* $T$. Together, these models allow the agent to plan or learn in downstream tasks using the encoded representations $\phi^*(x)$, modeling only the parts of the environment that the agent can actually control (the latent state $s \in \mathcal{S}$) while ignoring the potentially-complex dynamics of time-correlated noise.

Specifically, the aim of *efficient* representation learning in this setting is to learn $\phi^*$ and $T$, using a number of environment steps of exploration that is dependent only on $|\mathcal{S}|$ and the size of the function class $\mathcal{F}$ that the encoder $\phi^*$ belongs to, and is *not* dependent on the size of $\mathcal{X}$ or $\mathcal{E}$. This allows $\mathcal{X}$ and $\mathcal{E}$ to be very large or potentially even infinite, but still allows for representation learning to be tractable. Efroni et al. (2022b) proposes an algorithm, PPE, with this property. However, PPE only works in a finite-horizon setting, where the agent interacts with the environment in episodes of fixed length $H$. After each episode, the controllable state (almost) always resets to a deterministic start state $s_0 \in \mathcal{S}$ at the beginning of each episode. In this work, we instead consider the *single-trajectory, no-reset setting*, where the agent interacts with the environment in a single episode of unbounded length, with no ability to reset the state. This better models real-world cases, where, for example, expensive human intervention would be required to "reset" the environment that a robot trains in: we would rather not require this intervention. This no-reset Ex-BMDP setting was previously considered by Lamb et al. (2023) and Levine et al. (2024); however, the algorithms presented in those works do not have sample-complexity guarantees.

By contrast, the algorithm presented in this work is guaranteed to learn $\phi^*$ and $T$ using samples polynomial in $|\mathcal{S}|$ and $\log |\mathcal{F}|$, with no dependence on $|\mathcal{E}|$ and $|\mathcal{X}|$. We only require that the *mixing time* of the exogenous noise is bounded. In other words, the requirement is that $t_{\mathrm{mix}}$, the mixing time of the Markov chain on $\mathcal{E}$ induced by $\mathcal{T}_e$, is at most some known quantity $\hat{t}_{\mathrm{mix}}$. Note that we do *not* require that the *endogenous* state $s$ mixes quickly under any particular policy (although we do require – as do Lamb et al. (2023) and Levine et al. (2024) – that all states in $\mathcal{S}$ are *eventually* reachable from one another). In this setting, we derive an algorithm with the following asymptotic sample complexity (where $\mathcal{O}^*(f(x)) := \mathcal{O}(f(x) \log(f(x)))$):

$$\mathcal{O}^*\Big( N D |\mathcal{S}|^2 |\mathcal{A}| \cdot \log \frac{|\mathcal{F}|}{\delta} + |\mathcal{S}||\mathcal{A}|\hat{t}_{\mathrm{mix}} \cdot \log \frac{N|\mathcal{F}|}{\delta} + \frac{|\mathcal{S}|^2 D}{\epsilon} \cdot \log \frac{|\mathcal{F}|}{\delta} + \frac{|\mathcal{S}|\hat{t}_{\mathrm{mix}}}{\epsilon} \cdot \log \frac{|\mathcal{F}|}{\delta} \Big), \quad (1)$$

where $N$ is a predetermined upper-bound on $|\mathcal{S}|$, $\delta$ is the failure rate of the algorithm, $D$ is the maximum distance between any two latent states in $\mathcal{S}$ (at most $|\mathcal{S}|$), and $\epsilon$ is the minimum accuracy of the output learned encoder $\phi$ on any latent state class $s \in \mathcal{S}$. Note that this expression is at worst polynomial in $|\mathcal{S}|$, and linear in $\hat{t}_{\mathrm{mix}}$ and $\log |\mathcal{F}|$.

Our algorithm proceeds iteratively, at each iteration taking a certain sequence of actions repeatedly in a loop. Because the latent state dynamics $T$ are deterministic, this process is guaranteed to (after some transient period) enter a cycle of latent states, of bounded length. Because the latent states in

---

[1]Unlike most prior works on Ex-BMDPs, we do *not* make a block assumption on $\mathcal{E}$: we allow the same $x$ to be emitted by $\mathcal{Q}(s, e)$ and $\mathcal{Q}(s, e')$, for distinct $e, e'$. Technically, then, our Ex-BMDP framework represents a restricted class of Partially-Observed MDPs (POMDPs), rather than MDPs, because the complete state is not encoded within the observed $\mathcal{X}$.

this cycle are repeatedly re-visited, the algorithm is then able to predictably collect many samples of the same latent state, without the need to "re-set" the environment. Furthermore, because this looping can be continued indefinitely, the algorithm can "wait out" the mixing time of the exogenous dynamics, in order to collect near-i.i.d. samples of each latent state. We call our algorithm **S**ingle-**T**rajectory **E**xploration for **E**x-BMDPs via **L**ooping, or **STEEL**. In summary, we: (1) introduce **STEEL**, the first provably sample-efficient algorithm for learning Ex-BMDPs in a general function-approximation setting from a single trajectory, (2) prove the correctness and sample complexity of **STEEL**, and (3) empirically test **STEEL** on two toy problems to demonstrate its efficacy.

## 2 RELATED WORKS

### 2.1 REPRESENTATION LEARNING FOR EX-BMDP AND EXO-MDPS

The Ex-BMDP model was originally introduced by Efroni et al. (2022b), who propose the **PPE** algorithm to learn the endogenous state encoder $\phi(\cdot)$ and latent transition dynamics $T$ of an Ex-BMDP. PPE has explicit sample-complexity guarantees that are polynomial in $|\mathcal{S}|$ and $\log|\mathcal{F}|$: crucially, the sample complexity does not depend explicitly on $|\mathcal{E}|$ or $|\mathcal{X}|$. However, unlike the method proposed in this work, PPE is restricted to the *episodic, finite horizon setting* with (nearly) *deterministic resets*. After each episode, the endogenous state is (nearly) always reset to the *same* starting latent state $s_0 \in \mathcal{S}$, and the exogenous state $e_0 \in \mathcal{E}$ is i.i.d. resampled from a fixed starting distribution. Similarly to this work, PPE assumes that the latent transition dynamics $T$ are (close to) deterministic. Because both $s_0$ and $T$ are nearly deterministic, PPE can collect i.i.d. samples of observations $x$ associated with any latent state $s \in \mathcal{S}$ simply by executing the same sequence of actions starting from $s_0$ after each reset. By contrast, in our setting, we cannot reset the Ex-BMDP state, so it is more challenging to collect samples of a given latent state $s$.

Other works *have* considered the Ex-BMDP setting without latent state resets. Lamb et al. (2023) and Levine et al. (2024) consider a setting similar to ours, where the agent interacts with the environment in a single trajectory. However, these methods do not provide sample-complexity guarantees, and instead are only guaranteed to converge to the correct encoder in the limit of infinite samples.

Efroni et al. (2022a) considers a related "Exo-MDP" setting, and proposes the **ExoRL** algorithm. In this setting, while the environment is episodic, the latent state $s_0$ resets to a starting value sampled randomly from a fixed *distribution* after each episode. Additionally, the latent transition dynamics may be non-deterministic. However, unlike our work, Efroni et al. (2022a) does **not** consider the general function-approximation setting for state encoders. Instead, the observation $x$ is explicitly factorized into $d$ factors, and the controllable state $s$ consists of some unknown subset of $k$ of these factors: the representation learning problem is reduced to identifying which $k$ of the $d$ factors are action-dependent. ExoRL guarantees a sample-complexity polynomial in $2^k = |\mathcal{S}|$ and $\log(d)$.[2]

Recently, Mhammedi et al. (2024) have proposed an algorithm for provably sample-efficient policy optimization in the episodic Ex-BMDP setting with rewards, that can handle nondeterministic latent dynamics. However, the algorithm requires *simulator access* to the environment: this means that the agent is able to reset the environment to *any observation $x \in \mathcal{X}$* that has been previously observed. This requirement is considerably stronger than even the requirement of deterministic resets to a single latent state found in Efroni et al. (2022b).

In addition to our main claim that our proposed algorithm is the first provably sample-efficient algorithm for representation learning in the single-trajectory Ex-BMDP setting, another property of our method is that we do not make a "block" assumption on the *exogenous* state $e$. For a fixed $s \in \mathcal{S}$, in our setting, the same observation $x \in \mathcal{X}$ may be emitted by multiple distinct exogenous states $e, e' \in \mathcal{E}$. Prior works (Efroni et al., 2022b;a; Lamb et al., 2023; Levine et al., 2024) have stated assumptions that require that $e$ may be uniquely inferred from $x$.[3] Removing this restriction allows one to model a greater range of phenomena. For example, suppose an agent can turn on or turn off

---

[2]Because ExoRL allows for nondeterministic starting latent states, one might be able to adapt ExoRL to the infinite-horizon no-reset setting by considering the single episode as a chain of "pseudo-episodes" with nondeterministic start state (c.f., Xu et al. (2024)). However, one would have to ensure that $s$ mixes sufficiently between episodes, which may be challenging given that $s$ is not directly observed and has unknown, controllable dynamics. Even still, the resulting algorithm would not apply to our general function-approximation setting.

[3]Although it is not immediately clear why this restriction is necessary for the proposed algorithms.

| | No-Reset Setting | Sample-Complexity Guarantees | Function Approximation | Nondeterministic Reset State | Partially-Observed Exogenous State | Nondeterministic Latent Transitions |
|---|---|---|---|---|---|---|
| **STEEL** | ✓ | ✓ | ✓ | ✓ | ✓ | ✗ |
| (Efroni'22b) | ✗ | ✓ | ✓ | ✗ | ? | ✗ |
| (Lamb'23) | ✓ | ✗ | ✓ | ✓ | ? | ✗ |
| (Levine'24) | ✓ | ✗ | ✓ | ✓ | ? | ✗ |
| (Efroni'22a) | ✗ | ✓ | ✗ | ✓ | ✗ | ✓ |

Table 1: Comparison to Prior Works for learning Ex-BMDP Latent Dynamics

a "noisy TV": i.e., the agent can control whether or not some source of temporally-correlated noise is observable. This is allowed in our version of the Ex-BMDP formulation, but is not allowed if $e$ must be fully inferable from $x$. One prior work, Wu et al. (2024), also (implicitly) removes the block restriction on the exogenous state $e$. That work extends Ex-BMDP representation learning to the partially-observed state setting, with the assumption that the observation history within some known window is sufficient to infer the latent state $s$. However, Wu et al. (2024) does not provide any sample-complexity guarantees. Wang et al. (2022) and Kooi et al. (2023) consider similar settings with *continuous* controllable latent state. However, the proposed methods require explicitly modeling the exogenous noise state $e$, and there are no sample complexity guarantees. Trimponias & Dietterich (2023) considers the sample-efficiency of reward-based reinforcement learning in Ex-BMDPs assuming known endogenous and exogenous state encoders; however, it does not address the sample complexity of the representation learning problem.

## 2.2 REPRESENTATION LEARNING FOR BLOCK MDPS AND LOW-RANK MDPS

The Ex-BMDP framework can be considered as a generalization of the Block MDP framework (Dann et al., 2018; Du et al., 2019). Like the Ex-BMDP setting, the Block MDP setting models environments where the observed state space $\mathcal{X}$ of the overall MDP is much larger than an action-dependent latent state space $\mathcal{S}$. Some works in the Block MDP framework (Mhammedi et al., 2023; Misra et al., 2020) also allow for nondeterministic latent state transitions: that is $s_{t+1} \sim \mathcal{T}_s(s_t, a_t)$. However, unlike the Ex-BMDP setting, there is no exogenous latent state $e \in \mathcal{E}$ or exogenous dynamics $\mathcal{T}_e$: the observation is simply sampled as $x_t \sim \mathcal{Q}(s_t)$. In other words, the Block MDP setting does not allow for *time correlated noise outside of the modelled latent state $s$*. Therefore, even when stochastic latent-state transition are allowed, any time-correlated noise must be captured in $\mathcal{S}$, and so impacts the sample complexity (which is typically polynomial in $|\mathcal{S}|$).

The Low-Rank MDP framework can also be considered as an extension the Block MDP framework, but is an *orthogonal* extension to the Ex-BMDP framework. In Low Rank MDPs, there exist functions $\phi : \mathcal{X} \times \mathcal{A} \to \mathbb{R}^d$ and $\mu : \mathcal{X} \to \mathbb{R}^d$, such that $\Pr(x_{t+1} = x'|x_t = x, a_t = a) = \phi(x, a)^T \mu(x')$. The sample complexity depends only polynomially on $d$ and logarithmically on the size of the function classes for the state encoders $\phi$ and $\mu$; it should not depend explicitly on $|\mathcal{X}|$. Works under this framework include Agarwal et al. (2020); Uehara et al. (2022) and Cheng et al. (2023). Other works in the Low Rank MDP framework use a reward signal and only explicitly learn part of the representation (the encoder $\phi$), including Mhammedi et al. (2023) and Jiang et al. (2017); see Mhammedi et al. (2023) for a recent, thorough comparison of these works. Note that while BMDPs can be formulated as low-rank MDPs with $d = |\mathcal{S}|$, this does not hold for Ex-BMDPs: the rank of the transition probabilities on $\mathcal{X}$ depends on $|\mathcal{E}|$ – as noted by Efroni et al. (2022b).

## 3 NOTATION AND ASSUMPTIONS

- The Ex-BMDP, $M$, has observation space $\mathcal{X}$, with discrete endogenous states $\mathcal{S}$ that have deterministic, controllable dynamics, and possibly continuous exogenous states $\mathcal{E}$ with nondeterministic dynamics that do not depend on actions. Concretely, we have that $s_{t+1} = T(s_t, a_t)$, where $T$ is a deterministic function, and $e_{t+1} \sim \mathcal{T}_e(e_t)$. Let $x_t \sim \mathcal{Q}(s_t, e_t)$, for $x_t \in \mathcal{X}$, $s_t \in \mathcal{S}$, $e_t \in \mathcal{E}$, with the *block assumption* on $\mathcal{S}$. That is, a given $x \in \mathcal{X}$ can be emitted only by one particular $s \in \mathcal{S}$, which we define as $\phi^*(x_t) = s_t$. We assume that $M$ is accessed in one continuous trajectory. The initial endogenous state is an arbitrary $s_{\text{init}} \in \mathcal{S}$, and the initial exogenous state $e_{\text{init}} \sim \pi_{\mathcal{E}}^{\text{init}}$, where $\pi_{\mathcal{E}}^{\text{init}} \in \mathcal{P}(\mathcal{E})$.

- The exogenous dynamics on $\mathcal{E}$ are *irreducible and aperiodic*, with stationary distribution $\pi_{\mathcal{E}}$. There is a known upper bound $\hat{t}_{\text{mix}}$ on the *mixing time* $t_{\text{mix}}$, where (as defined in

Levin & Peres (2017) and elsewhere) $t_{\mathrm{mix}} := t_{\mathrm{mix}}(1/4)$, where $t_{\mathrm{mix}}(\epsilon)$ is defined such that:

$$\forall e \in \mathcal{E}, \ \| \Pr(e_{t+t_{\mathrm{mix}}(\epsilon)} = e' | e_t = e) - \pi_{\mathcal{E}}(e') \|_{\mathrm{TV}} \leq \epsilon. \tag{2}$$

This assumption bounds how "temporally correlated" the noise in the Ex-BMDP is: it ensures that the exogenous noise state $e_t$ at time $t$ is relatively unlikely to affect $e_{t+\hat{t}_{\mathrm{mix}}}$.

- We have a known upper bound on the number of endogenous latent states, $N \geq |\mathcal{S}|$. Additionally, we assume that all endogenous latent states can be reached from one another in at most $D$ steps, for some finite $D$ (note that we *do not* assume that all pairs of states in $\mathcal{S}$ can be reached from one another in *exactly* $D$ steps). We assume that there is a known upper bound on this diameter: $\hat{D} \geq D$. Trivially, if all endogenous latent states are reachable from one another then $D \leq N - 1$, so if a tighter bound is not available then we can use $\hat{D} := N - 1$. (In fact, it is not very important to use a tight bound here: $\hat{D}$ does not appear in the *asymptotic* sample complexity.)

- There is an encoder hypothesis class $\mathcal{F} : \mathcal{X} \to \{0, 1\}$, with realizablity for one-vs-rest classification of endogenous states. That is,

$$\forall s \in \mathcal{S}, \exists f \in \mathcal{F} : \forall x \in \mathcal{X}, f(x) = \mathbb{1}_{\phi^*(x)=s}. \tag{3}$$

  In other words, for every latent state $s \in \mathcal{S}$, there is some function $f \in \mathcal{F}$ such that $f(x) = 1$ if and only if $\phi^*(x) = s$.

- The algorithm has access to a training oracle for $\mathcal{F}$, which, given two finite multi-sets $D_0$ and $D_1$ each with elements from $\mathcal{X}$, returns a classifier $f \in \mathcal{F}$. The only requirement that we have for this oracle is that, if there exist any classifiers $\mathcal{F}^* \subset \mathcal{F}$, such that, for $f^* \in \mathcal{F}^*$, $\forall x \in D_0, f^*(x) = 0$ and $\forall x \in D_1, f^*(x) = 1$, then the oracle will return some member of $\mathcal{F}^*$. Note that an optimizer that minimizes the 0-1 loss on $D_0 \cup D_1$ will satisfy this requirement. However, it is not strictly necessary to minimize the 0-1 loss in particular.

- General notation: Let $\mathcal{M}(\mathcal{A})$ be the set of all multisets of the set $\mathcal{A}$. Let $\perp$ represent an undefined value. For lists $x, y$, let $x \cdot y$ represent their concatenation: that is, $[a, b] \cdot [c, d] = [a, b, c, d]$. For multisets $A$ and $B$, let $A \uplus B$ be their union, where their multiplicities are additive. Let $\%$ be the modulo operator, so that $a\%b \equiv a \pmod{b}$ and $0 \leq a\%b \leq b - 1$.

## 4 ALGORITHM

The STEEL algorithm is presented in full in Appendix A; in this section, we give a high-level overview of the algorithm, and present bounds on its sample-complexity. We state the sample-complexity and correctness of STEEL in the following theorem, which is proved in Appendix B.

**Theorem 1.** *For an Ex-BMDP $M = \langle \mathcal{X}, \mathcal{A}, \mathcal{S}, \mathcal{E}, \mathcal{Q}, T, \mathcal{T}_e, \pi_{\mathcal{E}}^{init} \rangle$ starting at an arbitrary endogenous latent state $s_{init} \in \mathcal{S}$, with $|\mathcal{S}| \leq N$, where the exogenous Markov chain $\mathcal{T}_e$ has mixing time at most $\hat{t}_{mix}$, and where all states in $\mathcal{S}$ are reachable from one another in at most $\hat{D}$ steps; and corresponding encoder function class $\mathcal{F}$ such that Equation 3 holds, the algorithm $STEEL(M, \mathcal{F}, N, \hat{D}, \hat{t}_{mix}, \delta, \epsilon)$ will output a learned endogenous state space $\mathcal{S}'$, transition model $T'$, and encoder $\phi'$, such that, with probability at least $1 - \delta$,*

- *$|\mathcal{S}'| = |\mathcal{S}|$, and under some bijective function $\sigma : \mathcal{S} \to \mathcal{S}'$, it holds that*

$$\forall s \in \mathcal{S}, a \in \mathcal{A} : \sigma(T(s, a)) = T'(\sigma(s), a), \ and, \tag{4}$$

- *Under the same bijection $\sigma$,*

$$\forall s \in \mathcal{S}, \Pr_{\substack{x \sim \mathcal{Q}(s,e), \\ e \sim \pi_{\mathcal{E}}}} (\phi'(x) = \sigma(\phi^*(x))) \geq 1 - \epsilon, \tag{5}$$

  *where $\pi_{\mathcal{E}}$ is the stationary distribution of $\mathcal{T}_e$.*

*Furthermore, the number of steps that STEEL executes on $M$ scales as:*

$$\mathcal{O}^*\left( ND|\mathcal{S}|^2|\mathcal{A}| \cdot \log \frac{|\mathcal{F}|}{\delta} + |\mathcal{S}||\mathcal{A}|\hat{t}_{mix} \cdot \log \frac{N|\mathcal{F}|}{\delta} + \frac{|\mathcal{S}|^2 D}{\epsilon} \cdot \log \frac{|\mathcal{F}|}{\delta} + \frac{|\mathcal{S}|\hat{t}_{mix}}{\epsilon} \cdot \log \frac{|\mathcal{F}|}{\delta} \right),$$

*where $\mathcal{O}^*(f(x)) := \mathcal{O}(f(x) \log(f(x)))$.*

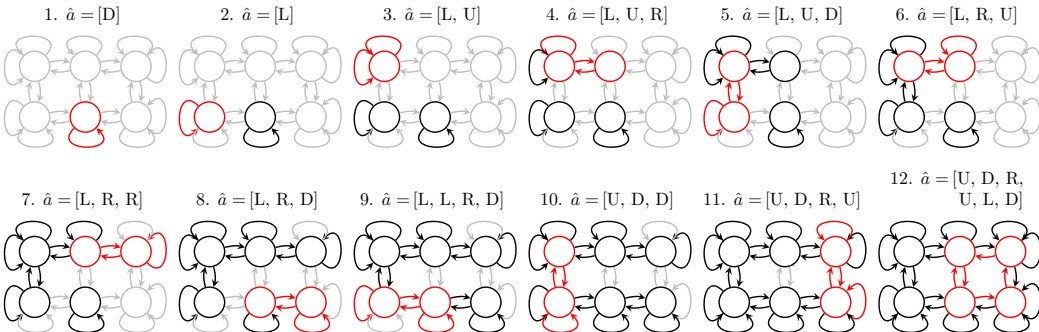

Figure 1: STEEL discovers the latent dynamics $\mathcal{S}$ and $T$ by iteratively adding cycles to the learned dynamics graph. In this simple example, the initially-unknown "true" dynamics consist of 6 states arranged in a grid, where the agent can move (U)p, (D)own, (L)eft, or (R)ight. STEEL takes 12 iterations to discover the full dynamics: each pane corresponds to an iteration, and shows the still-unknown parts of the dynamics graph in grey, the already-known parts of the dynamics graph in black, and the cycle being explored in red. States are represented as circles and transitions as arrows.

We now present a high-level overview of STEEL. The algorithm proceeds in three phases. In the first phase, the algorithm learns the transition dynamics; in the second phase, it collects additional samples of observations of each latent state in $\mathcal{S}$; in the final phase, the encoder $\phi'$ is learned.

**STEEL Phase 1: Learning latent dynamics.** In the first phase, STEEL constructs a learned latent state space $\mathcal{S}'$ and learned transition dynamics $T'$ by iteratively adding *cycles* to the known transition graph. At each iteration, a sequence of actions $\hat{a}$ is chosen such that, starting *anywhere* in the known $T'$, taking the actions in $\hat{a}$ is guaranteed to traverse a transition not already in $T'$. (We explain the process of constructing $\hat{a}$ below.)

STEEL then takes the actions in $\hat{a}$ repeatedly, collecting a sequence of observations $x_{CF}$. Because the transitions in $T$ are deterministic, this sequence of transitions must eventually (after at most $|\mathcal{S}||\hat{a}|$ steps) enter a *cycle* of latent states, of length $n_{\text{cyc}}|\hat{a}|$, for some $n_{\text{cyc}} \leq |\mathcal{S}|$. Because $\hat{a}$ was chosen to always escape the known transitions in $T'$, this cycle cannot be contained in $T'$, so adding the states and transitions of the new cycle to $\mathcal{S}'$ and $T'$ is guaranteed to expand the known dynamics graph by at least one edge: this process will discover the full transition dynamics after at most $|\mathcal{S}||\mathcal{A}|$ iterations. See Figure 1 for an example of how STEEL constructs $\mathcal{S}'$ and $T'$ by adding cycles to the dynamics. Throughout this process, STEEL also collects a dataset $\mathcal{D}(s) \in \mathcal{M}(\mathcal{X})$ for each newly-discovered latent state $s \in \mathcal{S}'$, so that each observation in $\mathcal{D}(s)$ has latent state $s$. The observations within each dataset $\mathcal{D}(s)$ are collected at least $\hat{t}_{\text{mix}}$ steps apart from one another, so they are near-i.i.d.

***Constructing*** $\hat{a}$. To ensure that $\hat{a}$ always escapes the known transition graph given by $T'$, at each iteration, STEEL uses a recurrent procedure to construct $\hat{a}$. At the beginning of this procedure, $\hat{a}$ is initialized as empty, and a set of latent states $\mathcal{B}$ is initialized with all of the learned states in $\mathcal{S}'$. At each step of the procedure, $\mathcal{B}$ represents the set of known states $s \in \mathcal{S}'$ that can be reached by starting at any arbitrary state $s' \in \mathcal{S}'$ and taking the action sequence given by the partially-constructed action list $\hat{a}$, following the known dynamics in $T'$. At each step, STEEL chooses a state $s \in \mathcal{B}$, and plans the shortest path from $s$ to any unknown transition in $T'$. The corresponding sequence of actions, $\hat{a}'$, is then appended to $\hat{a}$, and $\mathcal{B}$ is updated by replacing each state $b \in \mathcal{B}$ with the state that results from starting at $b$ and taking the sequence of actions $\hat{a}'$, according to the learned partial transition function $T'$. If this path reaches an unknown transition in $T'$, then $b$ is removed from $\mathcal{B}$ and not replaced. By construction, we know that the state $s$ will be removed, so $\mathcal{B}$ will shrink by at least one state at each step of the process. By the end, $\mathcal{B}$ is empty, and we are guaranteed that taking action sequence $\hat{a}$ from any state in $\mathcal{S}'$ will lead to an unknown transition in $T'$, as desired. Note that at each step, the shortest distance to an unknown transition has length at most $D + 1$, and the procedure continues for at most $|\mathcal{S}|$ steps, so $|\hat{a}| \leq |\mathcal{S}'| \cdot (D + 1)$.

***Identifying latent states***. At each iteration, to identify the distinct latent states in the cycle, STEEL uses a subroutine called CycleFind. CycleFind additionally collects the datasets $\mathcal{D}(s)$ for each newly-discovered latent state. CycleFind itself has two phases:

$n'_{cyc} = 3$; $n_{cyc} = 3$: $D_1$ contains only the red latent state; $D_0$ does not contain this latent state

Figure 2: CycleFind determines the period of the cycle in $x_{CF}$. See Section 4 under **"CycleFind Phase 1."** We show the sequence $x_{CF}$ sampled from $M$: specifically, we show every $|\hat{a}|$'th observation, where the same actions $\hat{a}$ are taken between each one. The observations' latent states are color-coded as red, blue, and green: a pattern repeats every $3|\hat{a}|$ steps, so $n_{cyc} = 3$. $D_1$ consists of the first observation in each $(n'_{cyc}|\hat{a}|)$-cycle, and $D_0$ the other observations taken between executions of $\hat{a}$. (Spans of length $\geq \hat{t}_{mix}$ are skipped to ensure certain subsets of the datasets are near-i.i.d.)

**CycleFind Phase 1: Finding the cycle's periodicity.** To identify the latent states in the cycle in $x_{CF}$, CycleFind first determines cycle's period, $n_{cyc}|\hat{a}|$. To find $n_{cyc}$, CycleFind tests all possible values of $n_{cyc}$ from $N$ to 1, in decreasing order. To check whether some candidate value, $n'_{cyc}$, is in fact $n_{cyc}$, CycleFind constructs two datasets, $D_0$ and $D_1$ from $x_{CF} = [x_1, ...]$. These datasets are constructed so that if $n'_{cyc} = n_{cyc}$, then $D_1$ contains observations of only one controllable latent state $s$, and $D_0$ contains no observations of $s$. Therefore, if one attempts to train a classifier $f$ to perfectly distinguish all observations in $D_0$ from those in $D_1$, then such a classifier is unlikely to exist in $\mathcal{F}$ if $n'_{cyc} > n_{cyc}$, but is guaranteed to exist if $n'_{cyc} = n_{cyc}$, by the realizability assumption. In this way, CycleFind uses the training oracle to determine $n_{cyc}$.

Specifically, the datasets used are (see Figure 2 for illustration.):

$$D_0 := \{x_{\bar{t}_{mix} + (2\bar{t}_{mix} + n'_{cyc}|\hat{a}|)i + j \cdot |\hat{a}| + \text{offs.}} \, | \, i \in \{0, ..., k-1\}, \; j \in \{1, ..., n'_{cyc} - 1\}\} \tag{6}$$

$$D_1 := \{x_{(2\bar{t}_{mix} + n'_{cyc} \cdot |\hat{a}|)i + \text{offs.}} \, | \, i \in \{0, ..., k-1\}\} \tag{7}$$

where $\bar{t}_{mix} := \lceil \hat{t}_{mix}/(n'_{cyc} \cdot |\hat{a}|) \rceil \cdot n'_{cyc} \cdot |\hat{a}|$ is $\hat{t}_{mix}$ rounded up to the nearest multiple of $n'_{cyc} \cdot |\hat{a}|$; the number of samples needed for $D_1$ is $k$ (which depends on $\log(|F|)$, $\log(\delta)$, and other parameters); and offs. $:= \max((N-1) \cdot |\hat{a}|, \hat{t}_{mix})$ is a constant offset to ensure that the Ex-BMDP endogenous state has entered the terminal cycle induced by $\hat{a}$, and that the exogenous state has mixed.

To see why $D_1$ and $D_0$ are perfectly distinguishable only if $n'_{cyc} = n_{cyc}$, consider the sequence of repeated latent states that compose the cycle, $[s^{cyc}_0, ..., s^{cyc}_{n_{cyc} \cdot |\hat{a}| - 1}]$. If we only consider every $|\hat{a}|$'th state in the cycle, then the resulting sequence, $[s^{cyc}_0, s^{cyc}_{|\hat{a}|}, ..., s^{cyc}_{(n_{cyc}-1) \cdot |\hat{a}|}]$, cannot contain any repeated states.[4] Therefore,

$$\forall n, m, \; n \equiv m \pmod{n_{cyc}} \Leftrightarrow \phi^*(x_{|\hat{a}|n + \text{offs.}}) = \phi^*(x_{|\hat{a}|m + \text{offs.}}), \tag{8}$$

and in particular, if $n'_{cyc} = n_{cyc}$, then $n \equiv m \pmod{n'_{cyc}} \Leftrightarrow \phi^*(x_{|\hat{a}|n + \text{offs.}}) = \phi^*(x_{|\hat{a}|m + \text{offs.}})$. Then, through modular arithmetic, we can conclude that $D_1$ contains observations of only one latent state of the cycle, while $D_0$ contains observations of all of the other latent states.

---

[4]Otherwise, due to the deterministic dynamics and repeated application of the same actions $\hat{a}$, the endogenous dynamics would immediately enter an even shorter cycle the first time a state is repeated in $[s^{cyc}_0, s^{cyc}_{|\hat{a}|}, ..., s^{cyc}_{(n_{cyc}-1) \cdot |\hat{a}|}]$, implying a shorter period.

Meanwhile, if $n'_{\text{cyc}} > n_{\text{cyc}}$, then for each observation in $D_1$, there is a corresponding observation in $D_0$ with the same latent state, that is nearly-identically and independently distributed (i.e, they are collected $\geq \hat{t}_{\text{mix}}$ steps apart). In particular, consider the (unknown) subset $D_0^{(j')} \subseteq D_0$, defined as:

$$D_0^{(j')} := \{x_{\bar{t}_{\text{mix}} + (2\bar{t}_{\text{mix}} + n'_{\text{cyc}}|\hat{a}|)i + j' \cdot |\hat{a}| + \text{offs.}} \,|\, i \in \{0, ..., k-1\}\} \tag{9}$$

where $j' := (-\lceil \hat{t}_{\text{mix}}/(n'_{\text{cyc}} \cdot |\hat{a}|)\rceil \cdot n'_{\text{cyc}} - 1)\% n_{cyc} + 1$. From arithmetic and applying Equation 8, we see that all observations in $D_0^{(j')}$ and all observations in $D_1$ have the same endogenous latent state. Additionally, all observations in $D_0^{(j')}$ and $D_1$ are collected at least $\hat{t}_{\text{mix}}$ steps apart.

**CycleFind Phase 2: Identifying latent states in the cycle.** Once $n_{\text{cyc}}$ is known, CycleFind can identify the latent states which re-occur every $n_{\text{cyc}}|\hat{a}|$ steps in $x_{CF}$. These latent states are not necessarily distinct from each other, and may also have been discovered already in a previous iteration of CycleFind. Therefore, CycleFind extracts from $x_{CF}$ datasets $\mathcal{D}'_i$ for each position in the cycle: $i \in \{0, ..., n_{\text{cyc}}|\hat{a}| - 1\}$. CycleFind also uses datasets $\mathcal{D}(s)$ collected in previous iterations representing the already-discovered states in $\mathcal{S}'$. CycleFind determines whether two datasets (either collected in this call to CycleFind, or collected in previous calls) represent the same latent state by attempting to learn a classifier $f \in \mathcal{F}$ that distinguishes them: if they both consist of near-i.i.d. samples of the same latent state, then it is highly unlikely that such a classifier exists.

To ensure the near-i.i.d. property "well enough," we only need that samples are separated by $\hat{t}_{\text{mix}}$ steps within each individual $\mathcal{D}'_i$; and that, when trying to distinguish two datasets $\mathcal{D}'_i, \mathcal{D}'_j$ which were both collected during this round of CyceFind, there are two (sufficiently large) *subsets* of $\mathcal{D}'_i$ and $\mathcal{D}'_j$ respectively such that all samples in the two subsets are collected at least $\hat{t}_{\text{mix}}$ steps apart – ensuring this second condition only doubles the number of samples we must collect. Thus, we do not need to "wait" $\hat{t}_{\text{mix}}$ steps between collecting each usable sample from $x_{CF}$; rather, we collect a usable sample *for each latent state* once for every roughly $2\max(\hat{t}_{\text{mix}}, n_{\text{cyc}}|\hat{a}|)$ steps. This is why $\hat{t}_{\text{mix}}$ does not appear in the largest term (in $|\mathcal{S}|$) of our asymptotic sample complexity.

If it is determined that some $\mathcal{D}_i$ represents a newly-discovered latent state, then a new state $s'$ is inserted into $\mathcal{S}'$ and $\mathcal{D}'(s')$ is initialized as $\mathcal{D}_i$. Once all states in $[s_0^{\text{cyc}}, ..., s_{n_{\text{cyc}} \cdot |\hat{a}| - 1}^{\text{cyc}}]$ have been identified, the action sequence $\hat{a}$ can be used to add them to the learned transition dynamics $T'$.

**STEEL Phase 2: Collecting additional samples to train encoder.**[5] Once we have the complete latent dynamics graph, the determinism of the latent dynamics allows us to use open-loop planning to efficiently re-visit each latent state, in order to collect enough samples to learn a highly-accurate encoder. Note that we can navigate to any arbitrary latent state in $D$ steps, so we can visit every latent state in $|\mathcal{S}|D$ steps. STEEL collects datasets $\mathcal{D}(s)$ for each latent state $s$ where, within each $\mathcal{D}(s)$, the samples are collected at least $\hat{t}_{\text{mix}}$ steps apart: therefore, it can add one sample to *each* dataset $\mathcal{D}(s)$ at worst roughly every $\max(|\mathcal{S}|D, \hat{t}_{\text{mix}})$ steps.

**STEEL Phase 3: Training the encoder.** Finally, STEEL trains the encoder. Specifically, for each latent state $s \in \mathcal{S}'$, it trains a binary classifier $f_s \in \mathcal{F}$ to distinguish $\mathcal{D}(s)$ from $\uplus_{s' \in \mathcal{S}' \setminus \{s\}} \mathcal{D}(s')$. To ensure that *only* the correct binary classifier, $f_{\sigma(\phi^*(x))}(x)$, returns 1, we ensure that *each* $f_s$ has an accuracy of $1 - \epsilon/|\mathcal{S}|$ on *each* latent state. We guarantee the accuracy of each classifier on each latent state separately and apply a union bound: note that because we use a union bound here, we do not need the samples in different datasets $\mathcal{D}(s), \mathcal{D}(s')$ to be independent, which is why we are able to collect samples more frequently that every $\hat{t}_{\text{mix}}$ steps. Finally, we define $\phi'(x) := \arg\max_s f_s(x)$.

## 5 SIMULATION EXPERIMENTS

We test the STEEL algorithm on two toy problems: an infinite-horizon environment inspired by the "combination lock" environment from Efroni et al. (2022b), and a version of the "multi-maze"

---

[5]In some scenarios, one might not need to learn an encoder at all. Note that the latent state $s$ of the agent is known at the last environment timestep $t$ of Phase 1 of STEEL. At this point, the full latent dynamics are already known. Thus, if the agent is "deployed" only once, immediately after training such that the latent state does not reset, then one could keep track of $s$ in an entirely open-loop manner while planning or learning rewards, without ever needing to use an encoder. In this case, the sample complexity terms involving $\epsilon$ disappear.

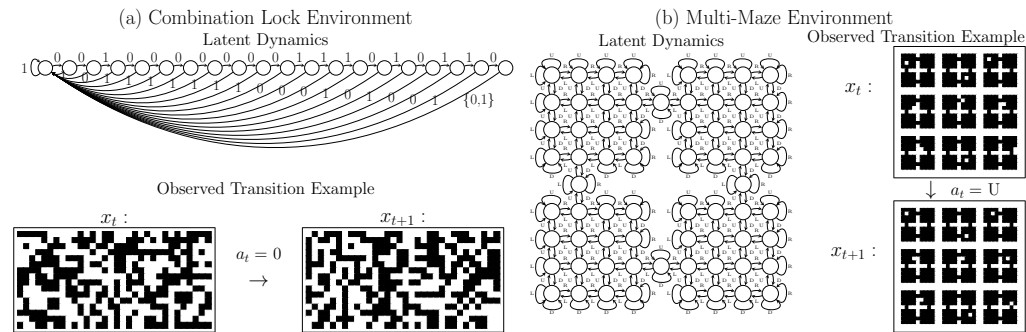

Figure 3: Visualisations of the simulation experiment environments. For both environments, we show the ground-truth latent dynamics $T$ (in the case of the combination lock, we show an arbitrary instance of $T$, for some $[a_0^*, ...a_{K-1}^*]$), and an example transition in the observed space $\mathcal{X}$.

environment from Lamb et al. (2023). In our combination lock environment, $\mathcal{A} = \{0, 1\}$, $\mathcal{S} = \{0, .., K-1\}$, and there is some sequence of "correct" actions $[a_0^*, ...a_{K-1}^*]$, such that $T(i, a_i^*) = i+1$, but $T(i, 1 - a_i^*) = 0$. In other words, in order to progress through the states, the agent must select the correct next action from the (arbitrary) sequence $[a_0^*, ...a_{K-1}^*]$; otherwise, the latent state is reset to 0. The observation space $\mathcal{X} = \{0, 1\}^L$, where $L \gg K$. Some arbitrary subset of size $K$ of the components in $\mathcal{X}$ are indicators for each latent state in $\mathcal{S}$: that is, $\forall i \in \mathcal{S}, \exists j \in \{0, ..., L-1\} : (x_t)_j = 1 \leftrightarrow s_t = i$. The other $L - K$ components in $\mathcal{X}$ are independent two-state Markov chains with states $\{0,1\}$, each with different arbitrary transition probabilities (bounded such that no transition probability for any of the two-state chains is less than $0.1$). Because each component is time-correlated, they all must be contained in $\mathcal{E}$, so $|\mathcal{E}| = 2^{L-K}$. In the multi-maze environment, the agent learns to navigate a four-room maze (similar to the one in Sutton et al. (1999)) using actions $\mathcal{A} = \{\text{Up, Down, Left, Right}\}$. The latent state space has size $|\mathcal{S}| = 68$. However, the observation $x \in \mathcal{X}$ in fact consists of *nine copies* of this maze, each containing a different apparent "agent." Eight of these "agents" move according to random actions: the true controllable agent is only present in one of the mazes. Because the eight distractor mazes can be in any configuration and have temporally-persistent state, we have that $|\mathcal{E}| = 68^8$. For both environments, we use the hypothesis class $\mathcal{F} := \{(x \to (x)_i | i \in \{0, \dim(\mathcal{X}) - 1\}\}$. In other words, the hypothesis class assumes that for each latent state $s$, there is some component of $i$ of the observations such that $\phi^*(x) = s$ if and only if $(x)_i = 1$. The two environments are visualized in Figure 3 .

There are four sources of potential variability in these simulation experiments: (1) the random elements of the environments' dynamics, $\mathcal{T}_e$, $\mathcal{Q}$, and $e_{\text{init}}$; (2) the starting latent state $s_{\text{init}}$; (3) steps in Algorithm 1 that allow for arbitrary choices (e.g., the choice of action $\hat{a} = [a]$ in the first invocation of CycleFind); and (4) the parameters of the environment, such as the "correct" action sequence $[a_0^*, ...a_{K-1}^*]$ in the combination lock. STEEL is designed in such a way that, with high probability (i.e., if the algorithm succeeds), *no choice that the algorithm makes* in terms of control flow or ac-

|  | Combo. Lock $(K = 20)$ | Combo. Lock $(K = 30)$ | Combo. Lock $(K = 40)$ | Multi-Maze |
|---|---|---|---|---|
| Fixed Env. Accuracy | 20/20 | 20/20 | 20/20 | 20/20 |
| Fixed Env. Steps | $1886582 \pm 0$ | $4286241 \pm 0$ | $7914856 \pm 0$ | $41003875 \pm 0$ |
| Variable Env. Accuracy | 20/20 | 20/20 | 20/20 | 20/20 |
| Variable Env. Steps | $2.00 \cdot 10^6$ $\pm 1.28 \cdot 10^5$ | $4.78 \cdot 10^6$ $\pm 4.36 \cdot 10^5$ | $9.59 \cdot 10^6$ $\pm 1.13 \cdot 10^6$ | $4.13 \cdot 10^7$ $\pm 1.11 \cdot 10^6$ |

Table 2: Success rate and number of steps taken for STEEL on both simulation environments. For all experiments, we set $\delta = \epsilon = .05$. For the combination lock experiments, we set $L = 512$, and use the (intentionally loose) upper bounds $N = \hat{D} = K + 10$ $(= |\mathcal{S}| + 10)$ and $\hat{t}_{\text{mix}} = 40$. For the multi-maze environment, we use $N = \hat{D} = 80$ $(> |\mathcal{S}| = 68)$, and $\hat{t}_{\text{mix}} = 300$. See Appendix D for how we chose the (loose) bounds $\hat{t}_{\text{mix}} \geq t_{\text{mix}}$.

tions will depend on exogenous noise.[6] Therefore, if we hold (2-4) constant, we expect the number of environment steps taken to be constant, regardless of the exogenous noise. To verify this, we test both environments for 20 simulations, in both a "fixed environment" setting with (2-4) held constant, and a "variable environment" setting with (2-4) set randomly. We test the combination lock environment with latent states $K \in \{20, 30, 40\}$. We measure the success rate in exactly learning $\phi^*(x)$ and $T$ (up to permutation) and the number of steps taken. Results are shown in Table 2.

STEEL correctly learned the latent dynamics $T$ and optimal encoder $\phi^*$ in every simulation run; and we verify that the step counts do not depend on exogenous noise. In the combination lock experiments for large $K$, which are hard exploration problems, the total step counts were many orders of magnitude smaller than either the size of the observation space ($\approx 10^{154}$); or the reciprocal-probability of a uniformly-random policy navigating from state 0 to state $K - 1$ ($\approx 10^{12}$ for $K = 40$). This shows that STEEL is effective at learning latent dynamics for hard exploration problems under high-dimensional, time-correlated noise. For the multi-maze experiment (which is *not* a hard exploration task), STEEL took a few orders of magnitude *greater* steps than reported in Lamb et al. (2023) or Levine et al. (2024) for the same environment ($\approx 10^3 - 10^4$ steps). However, unlike these prior methods, STEEL is *guaranteed* to discover the correct encoder with high probability; this requires the use of conservative bounds when defining sample counts $d$, $n_{\text{samp. cyc.}}$ and $n_{\text{samp.}}$ in Algorithms 1 and 2, and in making other adversarial assumptions in the design of the algorithm that ensure that it is correct and sample-efficient even in pathological cases. Additionally, note that the encoder hypothesis class $\mathcal{F}$ used in this experiment has no spatial priors. By contrast, Lamb et al. (2023) choose a neural-network encoder for this environment with strong spatial priors that favor focusing attention on a single maze, using sparsely-gated patch encodings (and Levine et al. (2024) use this same network architecture in order to compare to Lamb et al. (2023)) – this difference in priors over representations may also account for some of the gap in apparent sample efficiency.

In Appendix F, we present an additional set of experiments on a family of tabular Ex-BMDPs which are known to be particularly challenging to "multistep inverse" methods, such as those proposed by Lamb et al. (2023) and Levine et al. (2024). We find that, for sufficiently large instances of these environments, STEEL can in fact empirically outperform these prior "practical" methods.

## 6 LIMITATIONS AND CONCLUSION

A major limitation of STEEL that may constrain its real-world applicability is its strict determinism assumption on $T$. In the episodic setting, Efroni et al. (2022b) can get away with allowing rare deviations from deterministic latent transitions (no more often on average than once every $4|\mathcal{S}|$ *episodes*) because the environment resets "erase" these deviations before they can propagate for too long. By contrast, in the single-trajectory setting, the STEEL algorithm is fragile to even rare deviations in latent dynamics; ameliorating this issue may require significant changes to the algorithm.

A second barrier to practical applicability of STEEL is the need for an optimal training oracle for $\mathcal{F}$. While this is tractable for, e.g., linear models (with the realizability assumption ensuring linear separability), it becomes computationally intractable for anything much more complicated. However, this kind of assumption is common in sample-complexity results; and could be worked around in adapting STEEL to practical settings.[7]

Two additional limitations to our work are the assumption of reachability of all endogenous latent states $s \in \mathcal{S}$, and the requirement that an upper-bound on the mixing time of the exogenous noise be known *a priori*. However, in Appendix E, we argue that these assumptions are in fact *necessary*, for any algorithm in the single-trajectory, no-resets Ex-BMDP setting.

Finally, the core assumption that $\mathcal{S}$ is finite and small is of course a major limitation: sample-efficient reinforcement learning in combinatorial and continuous state spaces is a broad area of ongoing and future work. Despite these limitations, STEEL represents what we hope is an important contribution to representation learning in scenarios where resetting the environment during training is not possible, and observations are impacted by high-dimensional, time-correlated noise.

---

[6]This property is theoretically important because it ensures that the decision to collect a given observation is independent of all previous observations, given the ground truth dynamics and initial latent state.

[7]Similarly to how Efroni et al. (2022b) adapts PPE to practical settings in their experimental section.

ACKNOWLEDGEMENTS

A portion of this research has taken place in the Learning Agents Research Group (LARG) at the Artificial Intelligence Laboratory, The University of Texas at Austin. LARG research is supported in part by the National Science Foundation (FAIN-2019844, NRT-2125858), the Office of Naval Research (N00014-18-2243), Army Research Office (W911NF-23-2-0004, W911NF-17-2-0181), Lockheed Martin, and Good Systems, a research grand challenge at the University of Texas at Austin. The views and conclusions contained in this document are those of the authors alone. Peter Stone serves as the Executive Director of Sony AI America and receives financial compensation for this work. The terms of this arrangement have been reviewed and approved by the University of Texas at Austin in accordance with its policy on objectivity in research. Alexander Levine is supported by the NSF Institute for Foundations of Machine Learning (FAIN-2019844). Amy Zhang and Alexander Levine are supported by National Science Foundation (2340651) and Army Research Office (W911NF-24-1-0193).

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

## A  FULL ALGORITHM

The STEEL algorithm is presented here in full as Algorithm 1, with a major subroutine, CycleFind, split out as Algorithm 2.

---

**Algorithm 1:** STEEL

---

**Input:** Access to Ex-BMDP $M$, access to training oracle for function class $\mathcal{F}$, knowledge of upper bounds $N$, $\hat{D}$, $\hat{t}_{\text{mix}}$, parameters $\delta, \epsilon$.

Initialize learned latent state set $\mathcal{S}'$, initially empty;

Initialize table of collected datasets for each latent state: $\mathcal{D} : \mathcal{S}' \to \mathcal{M}(\mathcal{X})$;

Initialize learned latent dynamics: $T' : (\mathcal{S}' \cup \{\perp\}) \times \mathcal{A} \to (\mathcal{S}' \cup \{\perp\})$. (When a state $s$ is added to $\mathcal{S}'$, we initially set $\forall a \in \mathcal{A}, \ T(s, a) := \perp$. Also, we set $\forall a \in \mathcal{A}, \ T(\perp, a) := \perp$ as a permanent definition.);

// Phase 1: Discover latent dynamics $T$.

Chose arbitrary $a \in \mathcal{A}$;

$\mathcal{S}', \mathcal{D}, T', s_{\text{curr.}} \leftarrow \text{CycleFind}([a], \mathcal{S}', \mathcal{D}, T')$; // Special case for first iteration

**while** $\exists s \in \mathcal{S}', a \in \mathcal{A} : T'(s, a) := \perp$ **do**

    Initialize $\mathcal{B} \leftarrow \mathcal{S}'$, and initialize action list $\hat{a} \leftarrow [\,]$;

    **while** $\mathcal{B}$ *non-empty* **do**

        Chose arbitrary $s \in \mathcal{B}$;

        $\mathcal{B} \leftarrow \mathcal{B} \setminus \{s\}$;

        Let $\hat{a}' :=$ a minimum-length sequence of actions such that
$T'(T'(T'(...T'(s, \hat{a}'_0), \hat{a}'_1), \hat{a}'_2), ..., \hat{a}'_{|\hat{a}'|-1}) = \perp$. (This can be found using Dijkstra's algorithm.);

        $\hat{a} \leftarrow \hat{a} \cdot \hat{a}'$;

        $\mathcal{B} \leftarrow \{s'' \in \mathcal{S}' \,|\, \exists s' \in \mathcal{B} : T'(T'(T'(...T'(s', \hat{a}'_0), \hat{a}'_1), \hat{a}'_2), ..., \hat{a}'_{|\hat{a}'|-1}) = s''\}$;

    $\mathcal{S}', \mathcal{D}, T', s_{\text{curr.}} \leftarrow \text{CycleFind}(\hat{a}, \mathcal{S}', \mathcal{D}, T')$;

// Phase 2: Collect additional latent samples to train encoder.

Let $d := \lceil \frac{3|\mathcal{S}'| \ln(16|\mathcal{S}'|^2|\mathcal{F}|/\delta)}{\epsilon} \rceil$;

**while** $\exists s \in \mathcal{S}' : |\mathcal{D}(s)| < d$ **do**

    Let $\mathcal{C} := \{s \in \mathcal{S}' \,|\, |\mathcal{D}(s)| < d \wedge s \neq s_{\text{curr.}}\}$;

    Use $T'$ to plan a cycle of actions $\bar{a}$ starting at $s_{\text{curr.}}$ that visits all states $\mathcal{C}$ and then returns to $s_{\text{curr.}}$, by greedily applying Dijkstra's algorithm repeatedly;

    If $|\bar{a}| < \hat{t}_{\text{mix}}$, use $T'$ to extend $\bar{a}$ by repeatedly inserting the shortest-length self-loop of some state visited in $\bar{a}$ into $\bar{a}$ until $|\bar{a}| \geq \hat{t}_{\text{mix}}$;

    Execute all actions in $\bar{a}$ once without collecting data;

    **while** $\forall s \in \mathcal{C} : |\mathcal{D}(s)| < d$ **do**

        **for** $a$ *in* $\bar{a}_0, ..., \bar{a}_{|\bar{a}|-1}$ **do**

            Take action $a$ on $M$;

            $s_{\text{curr.}} \leftarrow T'(s_{\text{curr.}}, a)$;

            **if** $s_{curr.}$ *is being visited for the first time in this execution through $\bar{a}$* **then**

                Let $x_{\text{curr.}} :=$ the observed state of $M$;

                $\mathcal{D}(s_{\text{curr.}}) \leftarrow \mathcal{D}(s_{\text{curr.}}) \uplus \{x_{\text{curr.}}\}$;

// Phase 3: Train latent state encoder $\phi'$.

**for** $s \in \mathcal{S}'$ **do**

    Let $D_0 := \uplus_{s' \in \mathcal{S}' \setminus \{s\}} \mathcal{D}(s')$; $D_1 := \mathcal{D}(s)$;

    Apply training oracle to distinguish $D_0$ from $D_1$, yielding $f_s \in \mathcal{F}$;

**return** $\mathcal{S}', T'$, and $\phi'(x) := \arg\max_s f_s(x)$;

---

## B  PROOFS

### B.1  STEEL

Here, we explain the STEEL algorithm (Algorithm 1), and prove the correctness of Theorem 1. STEEL proceeds in three phases: in the first phase, we learn a tabular representation of the endogenous latent states $\mathcal{S}$ and associated dynamics $T$ of the Ex-BMDP. For each $s \in \mathcal{S}$, we also begin to collect a dataset $\mathcal{D}(s)$, where for each $x \in \mathcal{D}(s)$, we have that $\phi^*(x) = s$, and additionally where all samples in $\mathcal{D} := \bigcup_{s \in \mathcal{S}} \mathcal{D}(s)$ were collected from the Ex-BMDP $M$ at least $\hat{t}_{\text{mix}}$ time steps apart.

---

**Algorithm 2:** CycleFind Subroutine

---

**Input:** Action list $\hat{a}$; current learned state set $\mathcal{S}'$, datasets $\mathcal{D}$, and transition dynamics $T'$. Also, access to Ex-BMDP $M$, access to training oracle for function class $\mathcal{F}$, knowledge of upper bounds $N$, $\hat{t}_{\mathrm{mix}}$, and $\hat{D}$, and parameters $\delta$, $\epsilon$.

```
// Phase 1: find length of cycle, n_cyc · |â|.
```

Let $n_{\text{samp. cyc.}} := \left\lceil \ln\left(\frac{\delta}{4|\mathcal{A}| \cdot N \cdot (N-1) \cdot |\mathcal{F}|}\right) \Big/ \ln\left(\frac{9}{16}\right) \right\rceil$;

Let $c_{\mathrm{init}} := (2\hat{t}_{\mathrm{mix}} + 3N \cdot |\hat{a}| - 2) \cdot n_{\text{samp. cyc.}} - \hat{t}_{\mathrm{mix}} - N \cdot |\hat{a}| + 1 + \max((N-1) \cdot |\hat{a}|, \hat{t}_{\mathrm{mix}})$;

Collect a sequence of observation $x_{CF} := [x_1, ...x_{c_{\mathrm{init}}}]$ from $M$ by taking the actions in $\hat{a}$ repeatedly in a loop, for a total of $c_{\mathrm{init}}$ actions. (Action $\hat{a}_{i \% |\hat{a}|}$ is taken after observing $x_i$.);

Let $\bar{x}_i := x_{i \cdot |\hat{a}| + \max((N-1) \cdot |\hat{a}|, \hat{t}_{\mathrm{mix}})}$;

Initialize $n_{\mathrm{cyc}} \leftarrow 1$; // Default value if no other $n'_{\mathrm{cyc}}$ is $n_{\mathrm{cyc}}$

**for** $n'_{cyc}$ *in [N, N-1...,3,2]* **do**

> Let $q := \lceil \hat{t}_{\mathrm{mix}} / (n'_{\mathrm{cyc}} \cdot |\hat{a}|) \rceil$, $r := q \cdot n'_{\mathrm{cyc}}$, $k := \lfloor \frac{c_{\mathrm{init}} + r \cdot |\hat{a}| - \max((N-1) \cdot |\hat{a}|, \hat{t}_{\mathrm{mix}})}{2r \cdot |\hat{a}| + n'_{\mathrm{cyc}} \cdot |\hat{a}|} \rfloor$;
>
> Let $D_0 := \{\bar{x}_{r + (2r + n'_{\mathrm{cyc}})i + j} | i \in \{0, ..., k-1\}, \ j \in \{1, ..., n'_{\mathrm{cyc}} - 1\}\}$;
>
> Let $D_1 := \{\bar{x}_{(2r + n'_{\mathrm{cyc}})i} | i \in \{0, ..., k-1\}\}$;
>
> Apply training oracle to distinguish $D_0$ from $D_1$, yielding $f \in \mathcal{F}$;
>
> **if** $(\forall x \in D_0, f(x) = 0$ *and* $\forall x \in D_0, f(x) = 1)$ **then**
>
> > $n_{\mathrm{cyc}} \leftarrow n'_{\mathrm{cyc}}$;
> >
> > **break**;

```
// Phase 2: Assemble datasets for observations from cycle, identify
    new latent states, and update S', D, and T'.
```

Let $n_{\text{samp.}} := \left\lceil \ln\left(\frac{\delta}{4|\mathcal{A}| \cdot N^4 \cdot (\hat{D}+1) \cdot |\mathcal{F}|}\right) \Big/ \ln\left(\frac{9}{16}\right) \right\rceil$;

Let $c := 2 \cdot n_{\mathrm{cyc}} \cdot |\hat{a}| \cdot \left((n_{\text{samp.}} - 1) \cdot \left\lceil \frac{\hat{t}_{\mathrm{mix}}}{|\hat{a}| \cdot n_{\mathrm{cyc}}} \right\rceil + 1\right) + \hat{t}_{\mathrm{mix}} + \max((N - n_{\mathrm{cyc}}) \cdot |\hat{a}|, \hat{t}_{\mathrm{mix}})$;

Extend the sequence of observation $x_{CF}$ to length at least $c$ by taking the actions $\hat{a}$ repeatedly in a loop on $M$, for $\max(0, c - c_{\mathrm{init}})$ additional steps, so that $x_{CF} = [x_1, ...x_c]$;

Let $n_0 := \max((N - n_{\mathrm{cyc}}) \cdot |\hat{a}|, \hat{t}_{\mathrm{mix}})$, $n'_0 := n_0 + (n_{\text{samp.}} - 1) \cdot |\hat{a}| \cdot n_{\mathrm{cyc}} \cdot \left\lceil \frac{\hat{t}_{\mathrm{mix}}}{|\hat{a}| \cdot n_{\mathrm{cyc}}} \right\rceil + |\hat{a}| \cdot n_{\mathrm{cyc}} + \hat{t}_{\mathrm{mix}}$;

$\forall i \in \{0, ..., n_{\mathrm{cyc}} \cdot |\hat{a}| - 1\}$, Let:
$$\mathcal{D}'_i = \Big\{ x_j | \exists k \in \{0, ..., n_{\text{samp.}} - 1\}, \ \exists \text{ offset } \in \{n_0, n'_0\} :$$

$$j = k \cdot |\hat{a}| \cdot n_{\mathrm{cyc}} \cdot \left\lceil \frac{\hat{t}_{\mathrm{mix}}}{|\hat{a}| \cdot n_{\mathrm{cyc}}} \right\rceil + \text{ offset } + (i - \text{ offset})\%(n_{\mathrm{cyc}} \cdot |\hat{a}|) \Big\};$$

**for** $i \in \{0, ..., n_{cyc} \cdot |\hat{a}| - 1\}$ **do**

> Initialize **StateAlreadyFound?** $\leftarrow$ False;
>
> **for** $s \in \mathcal{S}'$ **do**
>
> > Let $D_0 := \mathcal{D}(s)$; $D_1 := \mathcal{D}'_i$;
> >
> > Apply training oracle to distinguish $D_0$ from $D_1$, yielding $f \in \mathcal{F}$;
> >
> > **if** *not* $(\forall x \in D_0, f(x) = 0$ *and* $\forall x \in D_0, f(x) = 1)$ **then**
> >
> > > $s_i^{\mathrm{cyc}} \leftarrow s$;
> > >
> > > (Optionally; and only if $\mathcal{D}(s)$ was not initialized or modified already during this call to CycleFind:) $\mathcal{D}(s) \leftarrow \mathcal{D}(s) \uplus \mathcal{D}'_i$;
> > >
> > > **StateAlreadyFound?** $\leftarrow$ True;
> > >
> > > **break**;
>
> **if** *not* **StateAlreadyFound?** **then**
>
> > Insert new state $s'$ into $\mathcal{S}'$;
> >
> > $\mathcal{D}(s') \leftarrow \mathcal{D}'_i$;
> >
> > $s_i^{\mathrm{cyc}} \leftarrow s'$;

**for** $i \in \{0, ..., n_{cyc} \cdot |\hat{a}| - 1\}$ **do**

> $T'(s_i^{\mathrm{cyc}}, a_{i\%|\hat{a}|}) \leftarrow s_{(i+1)\%(|\hat{a}| \cdot n_{\mathrm{cyc}})}^{\mathrm{cyc}}$;

$s_{\mathrm{curr.}} := s_{\max(c, c_{\mathrm{init}})\%(n_{\mathrm{cyc}} \cdot |\hat{a}|)}^{\mathrm{cyc}}$;

**return** $\mathcal{S}', \mathcal{D}, T', s_{\mathrm{curr.}}$;

---

Then, in the second phase, we use the learned dynamics model $T'$ to efficiently collect additional samples for each learned latent state $s \in \mathcal{S}'$ and add them to $\mathcal{D}(s)$, until $|\mathcal{D}(s)| \geq d$, where:

$$d := \lceil \frac{3|\mathcal{S}'| \ln(16|\mathcal{S}'|^2|\mathcal{F}|/\delta)}{\epsilon} \rceil. \tag{10}$$

Finally, in the third phase, we use $\mathcal{D}$ to learn an encoder $\phi'$, which approximates $\phi^*$ with high probability when the exogenous state $e$ of the Ex-BMDP is sampled from its stationary distribution $\pi_e$.

STEEL relies on the CycleFind subroutine, which is described in detail and proven correct in Section B.1.1. This subroutine is given a list of actions $\hat{a}$ and the previously-learned states $\mathcal{S}'$, datasets $\mathcal{D}$, and dynamics $T'$. It identifies and collects samples of all latent states in *some* state cycle which is traversed by taking the actions $\hat{a}$ repeatedly, and also identifies the latent state transitions in this cycle.

We restate Theorem 1 here:

**Theorem 1.** *For an Ex-BMDP $M = \langle \mathcal{X}, \mathcal{A}, \mathcal{S}, \mathcal{E}, \mathcal{Q}, T, \mathcal{T}_e, \pi_{\mathcal{E}}^{init} \rangle$ starting at an arbitrary endogenous latent state $s_{init} \in \mathcal{S}$, with $|\mathcal{S}| \leq N$, where the exogenous Markov chain $\mathcal{T}_e$ has mixing time at most $\hat{t}_{mix}$, and where all states in $\mathcal{S}$ are reachable from one another in at most $\hat{D}$ steps; and corresponding encoder function class $\mathcal{F}$ such that Equation 3 holds, the algorithm STEEL$(M, \mathcal{F}, N, \hat{D}, \hat{t}_{mix}, \delta, \epsilon)$ will output a learned endogenous state space $\mathcal{S}'$, transition model $T'$, and encoder $\phi'$, such that, with probability at least $1 - \delta$,*

- *$|\mathcal{S}'| = |\mathcal{S}|$, and under some bijective function $\sigma : \mathcal{S} \to \mathcal{S}'$, it holds that*

$$\forall s \in \mathcal{S}, a \in \mathcal{A} : \sigma(T(s,a)) = T'(\sigma(s), a), \; and, \tag{4}$$

- *Under the same bijection $\sigma$,*

$$\forall s \in \mathcal{S}, \Pr_{\substack{x \sim \mathcal{Q}(s,e), \\ e \sim \pi_{\mathcal{E}}}} (\phi'(x) = \sigma(\phi^*(x))) \geq 1 - \epsilon, \tag{5}$$

*where $\pi_{\mathcal{E}}$ is the stationary distribution of $\mathcal{T}_e$.*

*Furthermore, the number of steps that STEEL executes on $M$ scales as:*

$$\mathcal{O}^* \Big( ND|\mathcal{S}|^2|\mathcal{A}| \cdot \log \frac{|\mathcal{F}|}{\delta} + |\mathcal{S}||\mathcal{A}|\hat{t}_{mix} \cdot \log \frac{N|\mathcal{F}|}{\delta} + \frac{|\mathcal{S}|^2 D}{\epsilon} \cdot \log \frac{|\mathcal{F}|}{\delta} + \frac{|\mathcal{S}|\hat{t}_{mix}}{\epsilon} \cdot \log \frac{|\mathcal{F}|}{\delta} \Big),$$

*where $\mathcal{O}^*(f(x)) := \mathcal{O}(f(x) \log(f(x)))$.*

For the sake of our proof, we will treat the ground-truth properties of the Ex-BMDP, such as the latent dynamics $T$, as (unknown, arbitrary) *fixed quantities*, not as random variables. Similarly, we will treat the initial latent state $s_{init}$ as an arbitrary but fixed quantity, rather than a random variable. Furthermore, in the proof, we will treat decisions that are specified (implicitly or explicitly) as "arbitrary" in Algorithms 1 and 2 (such as the choice of the action $\hat{a} = [a]$ in the first invocation of CycleFind, or the choice of "shortest" paths in cases of ties when Dijkstra's algorithm is used) as being made deterministically, such as by a pseudorandom process – crucially, we require that these choices are made in a way that does not depend of the observations of the Ex-BMDP.

This leaves the exogenous noise transitions $\mathcal{T}_e$, the emission function $\mathcal{Q}$, and the initial exogenous latent state $e_{init}$ as the *only* sources of randomness in the algorithm. We notate the sample space over these three processes together as $\Omega$. Throughout the algorithm, we will ensure that decisions such as control flow choices, choices of actions, and choices of how to assemble datasets, are made *deterministically, independently of $\Omega$*, with high probability. That is, unless the algorithm *fails*, these decisions will be fully determined by $s_0$, $T$, and algorithm parameters. While *whether or not* the algorithm fails will depend (solely) on $\Omega$, we will ultimately bound the *total probability of failure* as less that $\delta$ by union bound, so that at each step in the proof, we can treat the algorithm's choices as statistically *independent* of $\Omega$.

STEEL begins by repeatedly applying the CycleFind subroutine. CycleFind identifies a *cycle* in the latent dynamics $T$ of the Ex-BMDP, and collects observations of the states in that cycle. In Phase 1

of STEEL, throughout these application of CycleFind, the algorithm maintains a representation of the learned Ex-BMDP "so far", in the form of an incomplete set of learned states $\mathcal{S}'$, learned latent dynamics $T' : (\mathcal{S}' \cup \{\perp\}) \times \mathcal{A} \to (\mathcal{S}' \cup \{\perp\})$, and a table of datasets corresponding to each latent state $\mathcal{D} : \mathcal{S}' \to \mathcal{M}(\mathcal{X})$.

We first describe the CycleFind subroutine in detail:

### B.1.1 CycleFind Subroutine

Here, we describe CycleFind, and prove its correctness. CycleFind accepts as input a list of actions $\hat{a} := [\hat{a}_0, ..., \hat{a}_{|\hat{a}|-1}]$, and the Ex-BMDP $M$ starting at an arbitrary state $x_0$. CycleFind also takes the representation of the learned Ex-BMDP "so far", in the form $\mathcal{S}'$, $\mathcal{D}$ and $T'$. We assume that, for each $s \in \mathcal{S}'$ all of the previously-observed observations in $\mathcal{D}(s)$ were collected with gaps of at least $\hat{t}_{\text{mix}}$ steps between them. Also, we assume that $\forall s \in \mathcal{S}', |\mathcal{D}(s)| \geq n_{\text{samp.}}$, where, in terms of the upper bounds $N \geq |S|$ and $\hat{t}_{\text{mix}} \geq t_{\text{mix}}$, and total failure probability $\delta$,

$$n_{\text{samp.}} := \left\lceil \ln\left(\frac{\delta}{4|\mathcal{A}| \cdot N^4 \cdot (\hat{D} + 1) \cdot |\mathcal{F}|}\right) \Big/ \ln\left(\frac{9}{16}\right) \right\rceil. \tag{11}$$

CycleFind first proceeds to take the actions $\hat{a}_0, ..., \hat{a}_{|\hat{a}|-1}$ repeatedly in a loop. CycleFind then uses this collected sequence of observations (which may need to be extended by cycling through $\hat{a}$ for additional iterations) to learn new states and update $\mathcal{S}'$, $\mathcal{D}$, and $T'$.

We will show that sequence of latent states visited by CycleFind is eventually periodic; that is, it guaranteed to eventually get stuck in a cycle of latent states, with a period in the form $n_{\text{cyc}} \cdot |\hat{a}|$, for some $n_{\text{cyc}} \leq N$. The goal of CycleFind is to:

1. Identify the period of this cycle. (That is, determine $n_{\text{cyc}}$.)

2. Use this period to extract from the sequence of observed states some new multisets of observations $\mathcal{D}'_i \in \mathcal{M}(\mathcal{X})$ for $i \in \{0, ..., n_{\text{cyc}} \cdot |\hat{a}| - 1\}$, which each contain only one unique latent state, corresponding to the position $i$ in the cycle. These multisets will only contain observations collected at least $\hat{t}_{\text{mix}}$ timesteps apart, so will be close-to-i.i.d. samples. (Depending on $n_{\text{cyc}}$, we may need to perform additional cycles of data collection at this step.)

3. Identify which of these multisets $\mathcal{D}'_i$ have the same latent states that have been previously identified in $\mathcal{S}'$, and which have a new latent state, and determine among the new multisets which ones have the same latent states to each other, and which are distinct. This allows us to update $\mathcal{S}'$ and $\mathcal{D}$ with the new samples from $\mathcal{D}'_i$, while maintaining the property that $\forall s \in \mathcal{S}', \forall x, x' \in \mathcal{D}(s), \phi^*(x) = \phi^*(x')$, and $\forall s, s' \in \mathcal{S}', \forall x \in \mathcal{D}(s), x' \in \mathcal{D}(s'), \phi^*(x) \neq \phi^*(x')$.) We also update the learned transitions $T'$.

4. Return the updated learned state set $\mathcal{S}'$, datasets $\mathcal{D}$, transition function $T'$, and the current latent state of $M$, $s_{\text{curr.}} \in \mathcal{S}'$.

Specifically, CycleFind has the following property:

**Proposition 1.** *For any action sequence $\hat{a}$ of length at most $(D + 1)N$, there exists at least one sequence of ground-truth states in $\mathcal{S}$, $[s_0^{cyc*}, s_1^{cyc*}, ..., s_{|\hat{a}| \cdot n_{cyc} - 1}^{cyc*}]$, for some $n_{cyc} \leq N$, such that $\forall i \in \{0, 1, ..., |\hat{a}| \cdot n_{cyc} - 1\}, T(s_i^{cyc*}, \hat{a}_{i\%|\hat{a}|}) = s_{(i+1)\%(|\hat{a}| \cdot n_{cyc})}^{cyc*}$. Given a sequence of actions $\hat{a}$, learned partial state set $\mathcal{S}'$, transition dynamics $T'$, and datasets $\mathcal{D}$ which meet the following inductive assumptions:*

- *There exists an injective mapping $\sigma^{-1} : \mathcal{S}' \to \mathcal{S}$ such that*

$$\forall s \in \mathcal{S}', a \in \mathcal{A}, \ T'(s, a) = \perp \vee \sigma^{-1}(T'(s, a)) = T(\sigma^{-1}(s), a) \tag{12}$$

*and additionally,*

$$\forall s \in \mathcal{S}', \forall x \in \mathcal{D}(s), \phi^*(x) = \sigma^{-1}(s). \tag{13}$$

- *$\forall s \in \mathcal{S}', |\mathcal{D}(s)| \geq n_{samp.}$; and for each $s$, the samples in $\mathcal{D}(s)$ were all sampled from $M$ at least $\hat{t}_{mix}$ steps apart. Additionally, the choice to add any sample $x$ to $\mathcal{D}(s)$ was made*

> *fully deterministically (as a function of $T$, $s_{init.}$, the timestep $t$ at which $x$ was collected, and algorithm parameters), and independently of the random processes captured by $\Omega$.*
>
> - *The choice of action sequence $\hat{a}$ is similarly fully deterministic and independent of $\Omega$.*

*then, with probability at least:*

$$1 - \frac{\delta}{2 \cdot |\mathcal{A}| \cdot N} \tag{14}$$

*CycleFind will return updated $\mathcal{S}'$, $\mathcal{D}$, $T'$, and $s_{curr.}$ which meet the same inductive assumptions, and for which additionally:*

> - *The image of the updated $\mathcal{S}'_{(new)}$, $\sigma^{-1}(\mathcal{S}'_{(new)})$ is a (non-strict) superset of $\sigma^{-1}(\mathcal{S}')$, which additionally includes all unique states in some $[s_0^{cyc*}, s_1^{cyc*}, ..., s_{|\hat{a}| \cdot n_{cyc}-1}^{cyc*}]$.*
>
> - *The transition matrix $T'_{(new)}$ is a (non-strict) superset of the old transition matrix $T'$ (in the sense that its domain is now $\mathcal{S}'_{(new)} \supseteq \mathcal{S}'_{(old)}$, and if $T'_{(old)}(s,a) \neq \bot$ then $T'_{(new)}(s,a) \neq \bot$), and $T'_{(new)}$ additionally includes the transitions corresponding to the cycle; that is:*
>
> $$\forall i \in \{0, ..., |\hat{a}| \cdot n_{cyc} - 1\}, \exists s, s' \in \mathcal{S}' : \sigma^{-1}(s) = s_i^{cyc*} \wedge \sigma^{-1}(s') = s_{(i+1)\%(|\hat{a}| \cdot n_{cyc})}^{cyc*} \wedge$$
> $$T'_{(new)}(s, \hat{a}_{i\%|\hat{a}|}) = s'. \tag{15}$$
>
> - *The final observation $x$ sampled by CycleFind from $M$ is such that $\sigma^{-1}(s_{curr.}) = \phi^*(x)$.*

*Additionally, CycleFind will take at most:*

$$\max\left( (2\hat{t}_{mix} + 3N \cdot |\hat{a}| - 2) \cdot n_{samp.\ cyc.} - N \cdot |\hat{a}|, 2 \cdot (\hat{t}_{mix} + |\mathcal{S}| \cdot |\hat{a}| - 1) \cdot n_{samp.} + 1 \right)$$
$$+ \max(N \cdot |\hat{a}| - |\hat{a}| - \hat{t}_{mix}, 0) + 1\ actions, \tag{16}$$

*where $n_{samp.}$ is defined in Equation 11 and $n_{samp.\ cyc.}$ is defined in Equation 18.*

*Proof.* **Determining $n_{\mathbf{cyc}}$:**

CycleFind initially takes $c_{\text{init}}$ actions, where, in terms of the upper bounds $N \geq |S|$ and $\hat{t}_{\text{mix}} \geq t_{\text{mix}}$,

$$c_{\text{init}} := (2\hat{t}_{\text{mix}} + 3N \cdot |\hat{a}| - 2) \cdot n_{\text{samp. cyc.}} - \hat{t}_{\text{mix}} - N \cdot |\hat{a}| + 1 + \max((N-1) \cdot |\hat{a}|, \hat{t}_{\text{mix}}), \tag{17}$$

where

$$n_{\text{samp. cyc.}} := \left\lceil \ln\left( \frac{\delta}{4|\mathcal{A}| \cdot N \cdot (N-1) \cdot |\mathcal{F}|} \right) \Big/ \ln\left( \frac{9}{16} \right) \right\rceil. \tag{18}$$

CycleFind first takes action $\hat{a}_0$, then $\hat{a}_1$, then $\hat{a}_2$, etc, until taking action $\hat{a}_{|\hat{a}|-1}$, at which point it repeats the process starting at $\hat{a}_0$, for a total of $c_{\text{init}}$ steps. The observation after each of these actions is recorded as $x_{CF} := [x_1, ..., x_{c_{\text{init}}}]$. Let $s_{CF} := [s_1, ..., s_{c_{\text{init}}}]$ be the (initially unknown) latent states corresponding to these observations; that is, $\phi^*(x)$ for each $x$ in $x_{CF}$. (For indexing purposes, $x_0$ and $s_0$ will refer to the observation and latent state, respectively, of the Ex-BMDP *before* the first action was taken by CycleFind. However, these will not be used by the algorithm.)

First, we show that $s_{CF}$ must in fact end in a cycle of period $n_{\text{cyc}} \cdot |\hat{a}|$, for some $n_{\text{cyc}} \leq N$. Let $s_{per.}$ consist of every $|\hat{a}|$'th element in $s_{CF}$ starting at an offset $m := \max(0, \hat{t}_{\text{mix}} - (N-1) \cdot |\hat{a}|)$; that is, $s_{per.} := [s_m, s_{m+|\hat{a}|}, s_{m+2|\hat{a}|}, ..., s_{m+\lfloor (c_{\text{init}}-m)/|\hat{a}| \rfloor |\hat{a}|}]$. Note that the evolution from one state to the next in $s_{per.}$ is deterministic, because it is caused by the same sequence of actions, $[\hat{a}_{m\%|\hat{a}|}, \hat{a}_{(m+1)\%|\hat{a}|}, ..., \hat{a}_{(m+|\hat{a}|-1)\%|\hat{a}|}]$, being taken after each state. That is, if $s_{m+i\cdot|\hat{a}|} = s$ and $s_{m+j\cdot|\hat{a}|} = s$ and $s_{m+(i+1)\cdot|\hat{a}|} = s'$, then $s_{m+(j+1)\cdot|\hat{a}|} = s'$. As a consequence, if $s_{m+i\cdot|\hat{a}|} = s$, and the next occurrence of the latent state $s$ in the sequence $s_{per.}$ is $s_{m+(i+t)\cdot|\hat{a}|} = s$, then all subsequent states in the sequence $s_{per.}$ will consist of repetitions of the sequence of $s_{m+(i+1)\cdot|\hat{a}|}$ through $s_{m+(i+t)\cdot|\hat{a}|}$.

Then $s_{per.}$ must consist of some sequence of 'transient' latent states which occur only once at the beginning of the sequence, followed by a repeated cycle. Because these states never re-occur, the transient part lasts length $n_{trn} \leq N - 1$.

Then, the period of the cycle is $n_{cyc}$, where $n_{trn} + n_{cyc} \leq N$. Note that the cycle in $s_{per.}$ contains no repeated states. (Otherwise, the span between the first two repetitions of a state in the cyclic sequence in $s_{per.}$ will itself repeat indefinitely, so we can analyse this smaller cycle as the cycle of length $n_{cyc}$.)

There is a corresponding cycle in $s_{CF}$, of length $n_{cyc} \cdot |\hat{a}|$. To see this, note that for all $i \geq n_{trn}$, we have that $s_{m+i|\hat{a}|} = s_{m+i|\hat{a}|+n_{cyc}\cdot|\hat{a}|}$. Furthermore, for all $j$ in $\{0, ...|\hat{a}| - 1\}$, the sequence of actions taken between $s_{m+i|\hat{a}|}$ and $s_{m+i|\hat{a}|+j}$ is the same as the sequence of actions taken between $s_{m+i|\hat{a}|+n_{cyc}\cdot|\hat{a}|}$ and $s_{m+i|\hat{a}|+j+n_{cyc}\cdot|\hat{a}|}$. Therefore $s_{m+i|\hat{a}|+j} = s_{m+i|\hat{a}|+j+n_{cyc}\cdot|\hat{a}|}$. Thus, for any general $i' \geq n_{trn}|\hat{a}|$ (which always can be written as $i' = i|\hat{a}| + j$) we have that $s_{m+i'} = s_{m+i'+n_{cyc}\cdot|\hat{a}|}$. However, this cycle *may* contain repeated states.

In order to avoid the transient part of $s_{CF}$, and to prevent sampling observations with exogenous noise that is correlated to samples taken in previous iterations of CycleFind, we skip the first $\max((N - 1) \cdot |\hat{a}|, \hat{t}_{mix})$ transitions in $s_{CF}$. For convenience, we will let $\bar{s}_i := s_{i\cdot|\hat{a}|+\max((N-1)\cdot|\hat{a}|,\hat{t}_{mix})}$, and similarly $\bar{x}_i := x_{i\cdot|\hat{a}|+\max((N-1)\cdot|\hat{a}|,\hat{t}_{mix})}$. Note that the sequence $[\bar{s}_0, \bar{s}_1, ...]$ is equivalent to $s_{per.}$ after skipping the first $N - 1 \geq n_{trn.}$ elements of the sequence.

Because the cycle in $s_{per.}$ contains no repeated states, we have that

$$\forall i, j \in \mathbb{N}, \ \ \bar{s}_i = \bar{s}_j \Leftrightarrow i \equiv j \pmod{n_{cyc}} \tag{19}$$

In order to find $n_{cyc}$, we test the hypothesis that $n_{cyc} = n'_{cyc}$, for each $n'_{cyc} \in \{N, ..., 2\}$, in order, until we identify $n_{cyc}$. If none of the tests pass, then we know that $n_{cyc} = 1$. The test for each hypothesis $n_{cyc} = n'_{cyc}$ has a zero false-negative rate. Consequently, the loop will always end before $n_{cyc} > n'_{cyc}$, so at each iteration, it must always be the case that $n_{cyc} \leq n'_{cyc}$. A failure can only occur if the test that $n_{cyc} = n'_{cyc}$ has a false positive, when in fact $n_{cyc} < n'_{cyc}$.

Each test proceeds as follows:

- Let $q := \lceil \hat{t}_{mix}/(n'_{cyc} \cdot |\hat{a}|) \rceil$ and $r := q \cdot n'_{cyc}$.

- Let $k := \lfloor \frac{c_{init}+r\cdot|\hat{a}|-\max((N-1)\cdot|\hat{a}|,\hat{t}_{mix})}{2r\cdot|\hat{a}|+n'_{cyc}\cdot|\hat{a}|} \rfloor$

- Let $D_0 := \{\bar{x}_{r+(2r+n'_{cyc})i+j} | i \in \{0, ..., k - 1\}, \ j \in \{1, ..., n'_{cyc} - 1\}\}$.

- Let $D_1 := \{\bar{x}_{(2r+n'_{cyc})i} | i \in \{0, ..., k - 1\}\}$.

- Use the training oracle to try to learn to distinguish $D_0$ from $D_1$, yielding $f \in \mathcal{F}$.

- If $n_{cyc} = n'_{cyc}$, then,

  - Note that, because $n_{cyc}|r$

  $$\forall i \in \mathbb{N}, \ \ (2r + n'_{cyc})i \equiv 0 \pmod{n_{cyc}}, \tag{20}$$

  but

  $$\forall i \in \mathbb{N}, j \in \{1, ..., n'_{cyc} - 1\},$$
  $$r + (2r + n'_{cyc})i + j \equiv j \not\equiv 0 \pmod{n_{cyc}}, \tag{21}$$

  - Consequently, by Equation 19, all elements of $D_1$ will have the same latent state, and none of the elements of $D_0$ have this latent state. By realizability, $f$ will have 100% accuracy on the training set. (This is the "true positive" case of the test.)

- Conversely, if $n_{cyc} < n'_{cyc}$, there is only a small chance that any classifier $f$ will have 100% accuracy on the training set.

- Define $j'$ as the (unknown) value $j' := (-r - 1) \% n_{\text{cyc}} + 1$. Noting, by assumption, that $n_{\text{cyc}} < n'_{\text{cyc}}$, we have that $j' \in \{1, .., n'_{\text{cyc}} - 1\}$. Then, $\forall i \in \{0, ..., k - 1\}$, we have that $\bar{x}_{r+(2r+n'_{\text{cyc}})i+j'} \in D_0$, while $\bar{x}_{(2r+n'_{\text{cyc}})i} \in D_1$. However, we also have that:

$$r + (2r + n'_{\text{cyc}})i + j' \equiv (2r + n'_{\text{cyc}})i \pmod{n_{\text{cyc}}} \tag{22}$$

  which by Equation 19 implies that

$$\bar{s}_{r+(2r+n'_{\text{cyc}})i+j'} = \bar{s}_{(2r+n'_{\text{cyc}})i}. \tag{23}$$

- Now, we can define $D_0^{(j')} \subseteq D_0$ as

$$D_0^{(j')} := \{\bar{x}_{r+(2r+n'_{\text{cyc}})i+j'} \mid i \in \{0, ..., k - 1\}\}. \tag{24}$$

- Fix any arbitrary classifier $f' \in \mathcal{F}$.
- In order for $f'$ to have 100% accuracy on the training set, we must have $f'(x) = 1$ for all $x \in D_1$, and $f'(x) = 0$ for all $x \in D_0^{(j')}$.
- Note that all observations in $D_1 \uplus D_0^{(j')}$ are collected at least $t_{\text{mix}}$ steps apart from one another. (Specifically, $\bar{x}_{r+(2r+n'_{\text{cyc}})i+j'}$ is collected $(r + j') \cdot |\hat{a}| \geq r \cdot |\hat{a}| \geq t_{mix}$ steps after $\bar{x}_{(2r+n'_{\text{cyc}})i}$, and $(r + n'_{\text{cyc}} - j') \cdot |\hat{a}| \geq r \cdot |\hat{a}| \geq t_{mix}$ steps before $\bar{x}_{(2r+n'_{\text{cyc}})(i+1)}$.)
- Because, additionally, $D_0^{(j')}$ and $D_1$ are defined independently of $\Omega$, by Lemma 1 we have:

$$\forall t \in \{t' \mid \bar{x}_{t'} \in D_1 \uplus D_0^{(j')}\},$$
$$p_s - 1/4 \leq \Pr(f'(\bar{x}_t) = 1 \mid (D_1 \uplus D_0^{(j')})_{<t}, \phi^*(\bar{x}_t) = s) \leq p_s + 1/4 \tag{25}$$

  where $(D_1 \uplus D_0^{(j')})_{<t}$ refers to the samples in $D_1 \uplus D_0^{(j')}$ collected before $\bar{x}_t$ and:

$$\forall s \in \mathcal{S}, \ p_s := \Pr(f'(x) = 1 \mid x \sim \mathcal{Q}(s, e), e \sim \pi_{\mathcal{E}}). \tag{26}$$

  Then, by Equation 26, the probability that $f'$ returns 1 on all samples in $D_1$, and 0 on all samples in $D_0^{(j')}$ is at most:

$$\Pi_{i=0}^{k-1}(p_{\bar{s}_{(2r+n'_{\text{cyc}})i}} + 1/4) \cdot \Pi_{i=0}^{k-1}(1 - (p_{\bar{s}_{r+(2r+n'_{\text{cyc}})i+j'}} - 1/4)). \tag{27}$$

  By Equation 23, this is:

$$\Pi_{i=0}^{k-1}(p_{\bar{s}_{(2r+n'_{\text{cyc}})i}} + 1/4) \cdot \Pi_{i=0}^{k-1}(1 - (p_{\bar{s}_{(2r+n'_{\text{cyc}})i}} - 1/4)). \tag{28}$$

  Rearranging gives us:

$$\text{FPR}(f') \leq \Pi_{i=0}^{k-1}(-p_{\bar{s}_{(2r+n'_{\text{cyc}})i}}^2 + p_{\bar{s}_{(2r+n'_{\text{cyc}})i}} + 5/16). \tag{29}$$

  Because $\forall p, \ -p^2 + p + 5/16 \leq 9/16$, we can upper-bound this as:

$$\text{FPR}(f') \leq \Pi_{i=0}^{k-1}(-p_{\bar{s}_{(2r+n'_{\text{cyc}})i}}^2 + p_{\bar{s}_{(2r+n'_{\text{cyc}})i}} + 5/16) \leq \left(\frac{9}{16}\right)^k \tag{30}$$

- As a uniform convergence bound:

$$FPR(f) \leq |\mathcal{F}| \left(\frac{9}{16}\right)^k \tag{31}$$

- Finally, we take a union bound over all values of $n'_{\text{cyc}}$. To do this, we must lower bound $k$ for all values of $n'_{\text{cyc}}$. First, note that

$$\hat{t}_{\text{mix}} + N|\hat{a}| - 1 \geq \hat{t}_{\text{mix}} + n'_{\text{cyc}}|\hat{a}| - 1 \geq r|\hat{a}| \tag{32}$$

Then:

$$
\begin{aligned}
k &= \left\lfloor \frac{(\hat{t}_{\text{mix}} + N \cdot |\hat{a}| - 1) \cdot (2 \cdot n_{\text{samp. cyc.}} - 1) + N \cdot |\hat{a}| \cdot n_{\text{samp. cyc.}} + r \cdot |\hat{a}|}{2r \cdot |\hat{a}| + n'_{\text{cyc}} \cdot |\hat{a}|} \right\rfloor \\
&\geq \left\lfloor \frac{r \cdot |\hat{a}| \cdot (2 \cdot n_{\text{samp. cyc.}} - 1) + N \cdot |\hat{a}| \cdot n_{\text{samp. cyc.}} + r \cdot |\hat{a}|}{2r \cdot |\hat{a}| + N \cdot |\hat{a}|} \right\rfloor \\
&\geq \lfloor n_{\text{samp. cyc.}} \rfloor \\
&\geq n_{\text{samp. cyc.}}.
\end{aligned}
\tag{33}
$$

So we have that, by union bound over all values of $n'_{\text{cyc}}$:

$$
FPR(f) \leq (N-1)|\mathcal{F}| \left( \frac{9}{16} \right)^{n_{\text{samp. cyc.}}}
\tag{34}
$$

**Collecting $\mathcal{D}'_i$:**

We now know that $s_{CF}$ eventually enters a cycle of length $n_{\text{cyc}} \cdot |\hat{a}|$, where $n_{\text{cyc}}$ is known. This is the latent-state cycle $[s_0^{cyc*}, s_1^{cyc*}, ..., s_{|\hat{a}| \cdot n_{\text{cyc}} - 1}^{cyc*}]$ mentioned in Proposition 1.

Depending on the value of $n_{\text{cyc}}$, we might now need to extend $x_{CF}$ (and, respectively $s_{CF}$) by making additional loops through $\hat{a}$, until the length of $x_{CF}$ is at least $c$, where:

$$
c := 2 \cdot n_{\text{cyc}} \cdot |\hat{a}| \cdot \left( (n_{\text{samp.}} - 1) \cdot \left\lceil \frac{\hat{t}_{\text{mix}}}{|\hat{a}| \cdot n_{\text{cyc}}} \right\rceil + 1 \right) + \hat{t}_{\text{mix}} + \max((N - n_{\text{cyc}}) \cdot |\hat{a}|, \hat{t}_{\text{mix}}).
\tag{35}
$$

This will entail taking an additional $\max(c - c_{\text{init}}, 0)$ steps on $M$. Note that in the worst case, this means that CycleFind takes a total of at most:

$$
\begin{aligned}
\max(c_{\text{init}}, c) \leq \max \Big( &(2\hat{t}_{\text{mix}} + 3N \cdot |\hat{a}| - 2) \cdot n_{\text{samp. cyc.}} - N \cdot |\hat{a}|, \\
&2 \cdot (\hat{t}_{\text{mix}} + |\mathcal{S}| \cdot |\hat{a}| - 1) \cdot n_{\text{samp.}} + 1 \Big) \\
&+ \max(N \cdot |\hat{a}| - |\hat{a}| - \hat{t}_{\text{mix}}, 0) + 1 \text{ actions.}
\end{aligned}
\tag{36}
$$

We now define how to collect two datasets for each position in the cycle in $s_{CF}$, $\mathcal{D}_i^A$ and $\mathcal{D}_i^B$ for each $i \in \{0, ..., n_{\text{cyc}} \cdot |\hat{a}| - 1\}$. Specifically we take:

$$
\begin{aligned}
\mathcal{D}_i^A = \Big\{ x_j | \exists k \in \{0, ..., n_{\text{samp.}} - 1\} : \\
j = k \cdot \left( |\hat{a}| \cdot n_{\text{cyc}} \cdot \left\lceil \frac{\hat{t}_{\text{mix}}}{|\hat{a}| \cdot n_{\text{cyc}}} \right\rceil \right) + n_0 + (i - n_0)\% \left( n_{\text{cyc}} \cdot |\hat{a}| \right) \Big\}
\end{aligned}
\tag{37}
$$

where we let

$$
n_0 := \max((N - n_{\text{cyc}}) \cdot |\hat{a}|, \hat{t}_{\text{mix}}),
\tag{38}
$$

and

$$
\begin{aligned}
\mathcal{D}_i^B = \Big\{ x_j | \exists k \in \{0, ..., n_{\text{samp.}} - 1\} : \\
j = k \cdot \left( |\hat{a}| \cdot n_{\text{cyc}} \cdot \left\lceil \frac{\hat{t}_{\text{mix}}}{|\hat{a}| \cdot n_{\text{cyc}}} \right\rceil \right) + n'_0 + (i - n'_0)\% \left( n_{\text{cyc}} \cdot |\hat{a}| \right) \Big\}
\end{aligned}
\tag{39}
$$

where

$$
n'_0 := n_0 + (n_{\text{samp.}} - 1) \cdot \left( |\hat{a}| \cdot n_{\text{cyc}} \cdot \left\lceil \frac{\hat{t}_{\text{mix}}}{|\hat{a}| \cdot n_{\text{cyc}}} \right\rceil \right) + |\hat{a}| \cdot n_{\text{cyc}} + \hat{t}_{\text{mix}}.
\tag{40}
$$

Note that, because we know that $s_{CF}$ enters a cycle of length $n_{\text{cyc}} \cdot |\hat{a}|$ after at most $(N - n_{\text{cyc}}) \cdot |\hat{a}| \leq n_0$ transitions, we have that,

$$
\forall i, j \geq n_0, \ i \equiv j \pmod{n_{\text{cyc}} \cdot |\hat{a}|} \to s_i = s_j.
\tag{41}
$$

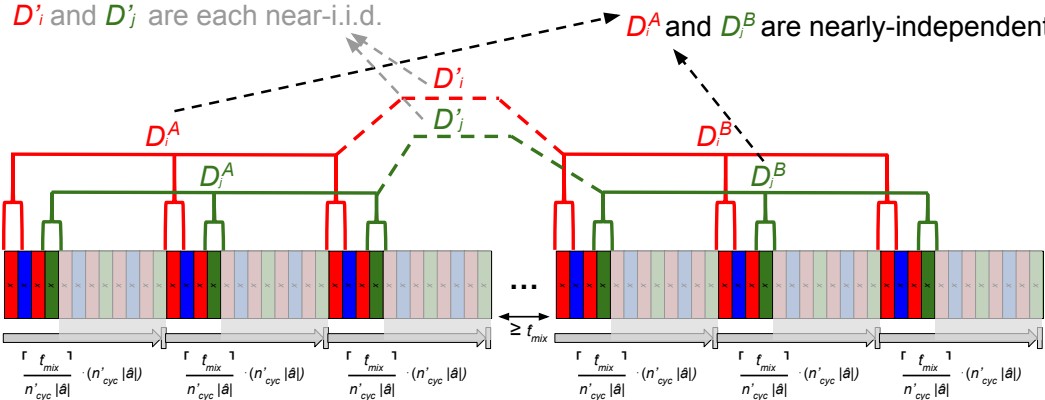

Figure 4: Illustration of the sampling procedure for datasets $\mathcal{D}'_i$. The first goal is to ensure that for *each* cycle position $i$, the samples in $\mathcal{D}'_i$ are sampled $t_{\text{mix}}$ steps apart from each other, and are therefore nearly i.i.d. We *also* want to ensure that for any *pair* of cycle positions $i, j$, there is a large subset of $\mathcal{D}'_i$ that only contains samples retrieved at least $t_{\text{mix}}$ steps apart from some large subset of $\mathcal{D}'_j$. (This second goal is meant to guarantee that if $\mathcal{D}'_i$ and $\mathcal{D}'_j$ represent the same latent state, then it is unlikely that any classifier exists than can separate the two subsets perfectly, which would be strictly necessary to perfectly separate $\mathcal{D}'_i$ and $\mathcal{D}'_j$). However, it is *not* necessary for all samples in $\cup_i \mathcal{D}_i$ to be collected $t_{\text{mix}}$ steps apart. Therefore, we collect an observation of *each* cycle position $i \in \{0, n_{\text{cyc}} \cdot |\hat{a}|\}$, all together, every $\hat{t}_{\text{mix}}$ steps (rounded up to the cycle period). We continue until $n_{\text{samp}}$ observations of each position are collected; then wait $\hat{t}_{\text{mix}}$ steps and collect $n_{\text{samp}}$ additional observations of each cycle position. This process ensures that $\mathcal{D}'_i$ and $\mathcal{D}'_j$ contain complementary subsets $\mathcal{D}^A_i$ and $\mathcal{D}^B_j$, each with at least $n_{\text{samp}}$ samples, such all samples in $\mathcal{D}^A_i \cup \mathcal{D}^B_j$ are near-i.i.d.

Therefore, because:

$$k \cdot \left( |\hat{a}| \cdot n_{\text{cyc}} \cdot \left\lceil \frac{\hat{t}_{\text{mix}}}{|\hat{a}| \cdot n_{\text{cyc}}} \right\rceil \right) + n_0 + (i - n_0)\%\Big( n_{\text{cyc}} \cdot |\hat{a}| \Big) \equiv i \pmod{n_{\text{cyc}} \cdot |\hat{a}|} \qquad (42)$$

(and a similar equivalence holds for $n'_0$), we have that, for any fixed $i$, all observations in $\mathcal{D}^A_i \uplus \mathcal{D}^B_i$ must share the same latent state. Also, for any fixed $i$, all observations in $\mathcal{D}^A_i$ and $\mathcal{D}^B_i$ are collected at least $\hat{t}$ steps apart. Additionally, $\mathcal{D}^A_i$ and $\mathcal{D}^B_i$ are defined solely in terms of $n_{\text{cyc}}$ and $\hat{a}$, and so the selection of samples to put in these sets only depends on the sequence of *latent* states $s$ that the Ex-BMDP traverses, and is therefore defined deterministically and independently of of $\Omega$ (assuming $n_{\text{cyc}}$ is correctly determined). We therefore define

$$\mathcal{D}'_i := \mathcal{D}^A_i \uplus \mathcal{D}^B_i, \qquad (43)$$

and note that all elements in this set both share the same latent state and were collected at least $\hat{t}$ steps apart from one another.

Additionally, for any fixed **pair** $i, j$, all observations in $\mathcal{D}^A_i \uplus \mathcal{D}^B_j$ are collected at least $\hat{t}$ steps apart from one another.

Using the $c$ samples in $x_{CF}$, this allows us to construct $\mathcal{D}^A_i$ and $\mathcal{D}^B_i$, for each $i \in \{0, n_{\text{cyc}} \cdot |\hat{a}| - 1\}$, where $|\mathcal{D}^A_i| = |\mathcal{D}^B_i| = n_{\text{samp}}$. See Figure 4 for an illustration of the sampling procedure.

**Identifying new latent states from $\mathcal{D}'$:**

At this point, each set $\mathcal{D}'_i$ consists of observations of a single latent state $s$, but two such sets $\mathcal{D}'_i$ and $\mathcal{D}'_j$ may represent the same latent state, and $\mathcal{D}'_i$ may contain the same latent state as some previously-collected $\mathcal{D}(s)$ for some $s \in \mathcal{S}'$.

In order to identify the newly-discovered latent states to add to $\mathcal{S}'$, and appropriately update $\mathcal{D}(\cdot)$ and $T'$, we proceed as follows:

- For $i \in \{0, ..., n_{\text{cyc}} \cdot |\hat{a}| - 1\}$:

- For each $s \in \mathcal{S}'$, use the training oracle to learn a classifier $f \in \mathcal{F}$, with $D_0 := \mathcal{D}(s)$ and $D_1 := \mathcal{D}'_i$. If $f$ can distinguish $D_0$ from $D_1$ with 100% training set accuracy, then we conclude (with high probability) that $\mathcal{D}(s)$ and $\mathcal{D}'_i$ represent two different latent states. Otherwise, we conclude that $\mathcal{D}(s)$ and $\mathcal{D}'_i$ both represent the same latent state.

- If $\mathcal{D}'_i$ is identified as representing some already-discovered latent state $s \in \mathcal{S}'$ then discard $\mathcal{D}'_i$. (Or, we can update $\mathcal{D}(s)$ by merging the samples in $\mathcal{D}'_i$ into it; this choice does not affect our analysis – however, we should avoid doing this if $\mathcal{D}(s)$ was either defined, or already updated, during this call of CycleFind: this is because two datasets $\mathcal{D}'_i$ and $\mathcal{D}'_j$ from the same iteration of CycleFind may contain observations that were sampled fewer than $\hat{t}_{\mathrm{mix}}$ steps apart from each other, which would break the inductive assumption on $\mathcal{D}(s)$ if they are both merged into $\mathcal{D}(s)$.) Record this latent state $s$ as:

$$s_i^{\mathrm{cyc}} := s \tag{44}$$

- Otherwise, if $\mathcal{D}'_i$ does not represent the any latent state $s \in \mathcal{S}'$, then $\mathcal{D}'_i$ (and $\mathcal{D}''_i$) represents a newly-discovered state. We update $\mathcal{S}'$ by inserting a new state $s'$ into it, and update $\mathcal{D}(s)$ by associating $s'$ with $\mathcal{D}'_i$:

$$\begin{aligned} \mathcal{S}' &\leftarrow \mathcal{S}' \cup \{s'\} \\ \mathcal{D}(s') &:= \mathcal{D}'_i \end{aligned} \tag{45}$$

Finally, we also record this new latent state as :

$$s_i^{\mathrm{cyc}} := s' \tag{46}$$

- To analyse the success rate of using the training oracle to determine if a given $\mathcal{D}(s)$ and $\mathcal{D}'_i$ represent the same latent state, consider the following:

  - If $\mathcal{D}(s)$ and $\mathcal{D}'_i$ contain different latent states, then $f$ will be able to distinguish $D_0$ from $D_1$, deterministically, with 100% accuracy on the training set (due to our realizability assumption.)
  - Otherwise, $\mathcal{D}(s)$ and $\mathcal{D}'_i$ both contain samples entirely of the *same* latent state, $s$. Then, either:
    * The latent state $s$ was identified before the current run of the CycleFind subroutine. Therefore, some subset of samples $D'_0 \subseteq D_0 = \mathcal{D}(s)$ were added to $\mathcal{D}(s)$ before the current run of CycleFind, such that $|D'_0| \geq n_{\mathrm{samp.}}$. Let $D'_1 := D_1 = \mathcal{D}'_i$, and note also that all samples in $D'_0 \uplus D'_1$ were collected at least $\hat{t}_{\mathrm{mix}}$ steps apart from one another. (This is by inductive hypothesis for $\mathcal{D}(s)$, by construction for $\mathcal{D}'_i$, and by the fact that each run of CycleFind starts by "wasting" at least $\hat{t}_{\mathrm{mix}}$ steps.)
    * The latent state $s$ was identified during the current run of CycleFind, such that $\mathcal{D}(s) = \mathcal{D}'_j$ for some $j < i$. Then let $D'_0 := \mathcal{D}_j^A \subseteq D_0$ and $D'_1 := \mathcal{D}_i^B \subseteq D_1$. Note that $|D'_0|, |D'_1| \geq n_{\mathrm{samp.}}$, and all observations in $D'_0 \uplus D'_1$ were collected at least $\hat{t}_{\mathrm{mix}}$ steps apart from one another.
    * In either case, the choice of samples to include in $D'_0 \uplus D'_1$ was made deterministically and independently of $\Omega$ (by construction and/or assumption).

  We define $p_s$ as in Equation 72. Note that the samples in $D'_0$ and $D'_1$ were observed at least $\hat{t}_{\mathrm{mix}}$ steps apart, at deterministically-chosen timesteps. Then Lemma 1 is applicable, and we have that the probability that an arbitrary $f' \in \mathcal{F}$ returns 1 on all samples from $D_1 \supseteq D'_1$; and also returns 0 on all samples from $D_0 \supseteq D'_0$ is at most:

$$(p_s + 1/4)^{n_{\mathrm{samp.}}} \cdot (1 - (p_s - 1/4))^{n_{\mathrm{samp.}}} =$$
$$(-p_s^2 + p_s + 5/16)^{n_{\mathrm{samp.}}} \leq \left(\frac{9}{16}\right)^{n_{\mathrm{samp.}}} \tag{47}$$

As a uniform convergence bound, we then have that:

$$FPR(f) \leq |\mathcal{F}| \left(\frac{9}{16}\right)^{n_{\mathrm{samp.}}} \tag{48}$$

- Note that at all iterations, $|\mathcal{S}'| \leq |\mathcal{S}|$, so we train at most $|\mathcal{S}| \cdot n_{\text{cyc}} \cdot |\hat{a}|$ classifiers. Therefore, by union bound, the total failure rate is bounded by:

$$\Pr(\text{fail}) \leq |\mathcal{S}| \cdot n_{\text{cyc}} \cdot |\hat{a}| \cdot |\mathcal{F}| \left(\frac{9}{16}\right)^{n_{\text{samp.}}} \tag{49}$$

- Note that the states $s_i^{\text{cyc}}$ now represent the latent states associated with the cyclic part of $x_{CF}$. Because we know the actions in the cycle, we can use this information to update $T'$. Specifically, $\forall i \in \{0, 1, ..., |\hat{a}| \cdot n_{\text{cyc}} - 1\}$, the action taken after $s_i^{\text{cyc}}$ and before $s_{(i+1)\%(|\hat{a}| \cdot n_{\text{cyc}})}^{\text{cyc}}$ is $\hat{a}_{i\%|\hat{a}|}$. We can then update:

$$T'(s_i^{\text{cyc}}, a_{i\%|\hat{a}|}) \leftarrow s_{(i+1)\%(|\hat{a}| \cdot n_{\text{cyc}})}^{\text{cyc}} \tag{50}$$

**Return the updated $\mathcal{S}',\mathcal{D}$, and $T'$, as well as $s_{\text{curr.}}$:**

Returning the updated $\mathcal{S}',\mathcal{D}$, and $T'$ is straightforward. Note that the choice to assign or merge a given $\mathcal{D}'_i$ in to a given $\mathcal{D}(s)$ depends only on the latent states $s$ in the datasets, and so is independent of $\Omega$.

We have then shown that, if CycleFind succeeds, then states $[s_0^{\text{cyc}}, .., s_{n_{\text{cyc}} \cdot |\hat{a}|-1}^{\text{cyc}}]$, have been added to $\mathcal{S}'$, if they were not present already. These states correspond to the states in the cycle $[s_0^{cyc*}, .., s_{n_{\text{cyc}} \cdot |\hat{a}|-1}^{cyc*}]$, and the corresponding transitions have been added to $T'$; furthermore, the datasets $\mathcal{D}(s)$ have been updated appropriately.

To determine the learned latent state of the Ex-BMDP $M$ after CycleFind is run, simply note that this is equivalently the state corresponding to the observation $x_c$, which we know belongs to dataset $\mathcal{D}'_{c\%(n_{\text{cyc}}|\hat{a}|)}$. We then know that this observation must have the same latent state as the rest of $\mathcal{D}'_{c\%(n_{\text{cyc}}|\hat{a}|)}$; that is, the observation $s_{c\%(n_{\text{cyc}} \cdot |\hat{a}|)}^{\text{cyc}}$.

The total failure rate for the CycleFind algorithm can be bounded by union bound from the failure rates of Parts 1 and 3 of the algorithm; that is, Equations 34 and 49. That is:

$$\begin{aligned}
\Pr(\text{fail}) &\leq (N-1) \cdot |\mathcal{F}| \left(\frac{9}{16}\right)^{n_{\text{samp. cyc.}}} + |\mathcal{S}| \cdot n_{\text{cyc}} \cdot |\hat{a}| \cdot |\mathcal{F}| \left(\frac{9}{16}\right)^{n_{\text{samp.}}} \\
&\leq (N-1) \cdot |\mathcal{F}| \frac{\delta}{4|\mathcal{A}|N(N-1)|\mathcal{F}|} + N^3 \cdot (D+1) \cdot |\mathcal{F}| \frac{\delta}{4|\mathcal{A}|N^4(\hat{D}+1)|\mathcal{F}|} \\
&\leq \frac{\delta}{2|\mathcal{A}| \cdot N}.
\end{aligned} \tag{51}$$

All claims of Proposition 1 have therefore been proven. $\qquad\square$

### B.1.2  STEEL PHASE 1

Note that given a fixed $\hat{a}$, there might be multiple different state cycles that could be discovered by CycleFind. However, only one will *actually* be discovered, depending on the state that the Ex-BMDP starts in as well as the *not-yet-discovered parts of the state dynamics*. For example, consider an Ex-BMDP $\mathcal{A} := \{L, R\}, \mathcal{S} := \{\hat{1}, \hat{2}\}$ with the following latent dynamics:

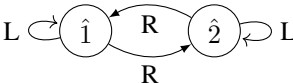

If we set $\hat{a} := [L]$, then, depending on the initial state, CycleFind will *either* collect samples of $\hat{1}$ and discover its self-loop transition, *or* collect samples of $\hat{2}$ and discover its self-loop transition.

In order to learn the complete latent dynamics of the Ex-BMDP, we maintain a representation $T'$ of the partial transition graph that has been discovered so far, and iteratively apply CycleFind using, at each step, an action sequence $\hat{a}$ that is guaranteed to produce a cycle that is *not* entirely contained in the partial graph discovered so far.

Note that this is *not* as simple as choosing a sequence of actions that leads to an unknown state transition from the final latent state of the Ex-BMDP reached in the previous iteration of CycleFind. For example, consider the following partially-learned latent state dynamics (with $\mathcal{A} := \{L, R\}$):

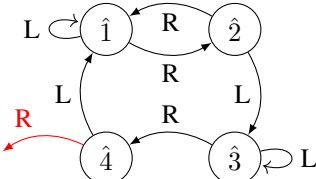

Here, the only unknown transition from an known state is the effect of the 'R' action from $\hat{4}$. Suppose we know from the previous iteration of CycleFind that the current latent state of the Ex-BMDP is $s_{\text{curr.}} = \hat{3}$. Naively, it might seem as if running CycleFind with $\hat{a} = [R, R]$ would learn some new transition dynamics or states, because it would navigate through the unknown transition. However, this might not be the case in fact. In particular, the 'R'-transition from $\hat{4}$ might only be visited *transiently*. For example, suppose the *full* latent dynamics of the Ex-BMDP are as follows (with the currently unknown parts shown in gray):

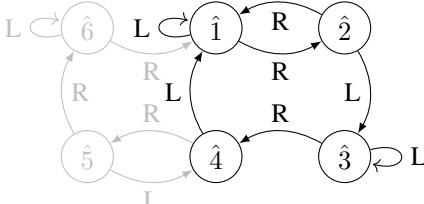

Then, if we run CycleFind with $\hat{a} = [R, R]$, it will converge to a cycle between the nodes $\hat{1}$ and $\hat{2}$:

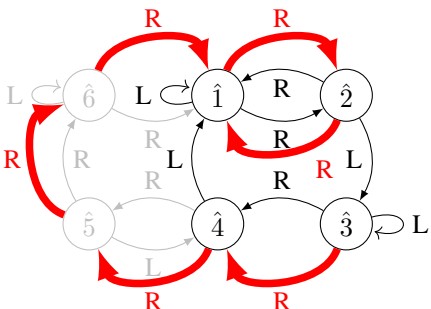

Note that the states ($\hat{1}$ and $\hat{2}$) and associated transitions that CycleFind converges on were *already* explored, so we learn no new information from this application of CycleFind.

Instead, at each iteration, we design $\hat{a}$ so that *no cycle of the actions $\hat{a}$ can be entirely contained within the currently-known partial transition graph.* We show that the length of the resulting $\hat{a}$ is at most $(D + 1)|\mathcal{S}|$.

We proceed as follows. Note that in the first iteration, before any latent states are known, we can simply use $\hat{a} = [a]$ for some arbitrary $a \in \mathcal{A}$. Otherwise, we use the following algorithm:

- Initialize $\mathcal{B}$ with all of the previously-learned latent states (that is, $\mathcal{B} \leftarrow \mathcal{S}'$.)
- While $\mathcal{B}$ is non-empty:
    - Remove some latent state $s$ from $\mathcal{B}$.
    - Use Dijkstra's algorithm to compute a shortest path in the partial transition graph that starts at $s$ and ends at any not-yet-defined transition. (that, is, any transition for which $T'(\cdot, \cdot) = \bot$). ( Note this must be possible. Otherwise, because all states can reach each other in the full latent dynamics, if there were no such undiscovered edge in the

same connected component as $s$, then we would know that we have *already* found the complete latent dynamics.) Also note that the shortest path through such an edge can have length at most $D + 1$, simply because:

* The length of the shortest path from $s$ to the state with the undefined edge in the *full* transition graph $T$ is at most $D$.
* Suppose that some transition on this shortest path is missing in the partial transition graph $T'$. Concretely, let $d$ be the first state along this path such that the transition out of it is missing. Then, we have a path from $s$ to $d$ of length less than $D$, and we know that $d$ is itself missing a transition. Then $d$ can be used in place of the original state with the undefined edge: it has a missing transition, and it is at most $D$ steps, in $T'$, from $s$.

- Let $\hat{a}'$ be the list of actions on the path we have found from $s$ through an undefined edge. Note that taking actions $\hat{a}'$ from $s$ will result in taking an unknown transition, and that $|\hat{a}'| \leq D + 1$.

- Replace $\mathcal{B}$ with the set of states that can result from starting at any state $s' \in \mathcal{B}$ and then taking actions $\hat{a}'$, according to the learned partial transition graph $T'$. For a given $s' \in \mathcal{B}$, if this path leads to an unknown transition, do not insert any state corresponding to $s'$ into the new $\mathcal{B}$.

- Concatenate $\hat{a}'$ to the end of $\hat{a}$.

Note that at every iteration, $|\mathcal{B}|$ decreases by at least 1, so the algorithm runs for at most $|\mathcal{S}'|$ iterations, so the final length of $\hat{a}$ is at most $(D + 1)|\mathcal{S}'|$. Also note that, by construction, taking all actions in $\hat{a}$ will traverse an unknown transition at some point, starting at *any* latent state that has been learned so far. As a consequence, any cycle traversed by taking $\hat{a}$ repeatedly must involve at least one transition (and possibly some states) that are not yet included in the partial transition graph. Therefore, applying CycleFind using an $\hat{a}$ constructed in this way is guaranteed to learn at least one new transition. Therefore, to fully learn the transition dynamics, we must apply CycleFind at most $|\mathcal{A}| \cdot |\mathcal{S}|$ times.

Also, note that the process of constructing $\hat{a}$ at each iteration depends only on the partial latent dynamics model $T'$, which in turn depends only on the choices of $\hat{a}$ in previous invocations of CycleFind, and ultimately these depend only on the starting latent state $s_{\text{init}}$ and the ground-truth dynamics $T$. Therefore $\hat{a}$ is at every iteration independent of $\Omega$, as required by Proposition 1.

Then, assuming CycleFind succeeds at each invocation, by the end of Phase 1, STEEL will have discovered the complete state set $\mathcal{S}$ and transition function $T$, up to permutation.

### B.1.3 STEEL PHASE 2

In the next phase, once we have completely learned $T'$ (that is, once there are no state-action pairs $s \in \mathcal{S}', a \in \mathcal{A}$ for which $T'(s, a)$ is undefined), we collect additional samples of each latent state, until the total number of samples collected for each is at least $d$, where:

$$d := \lceil \frac{3|\mathcal{S}'| \ln(16|\mathcal{S}'|^2|\mathcal{F}|/\delta)}{\epsilon} \rceil. \tag{52}$$

We can leverage the fact that we now have a complete latent transition graph as well as knowledge of the current latent state $s_{\text{curr.}}$ from the last iteration of CycleFind.

To do this, we proceed as follows:

- Use $T'$ to plan a sequence of actions $\bar{a}$ such that:
  - $\hat{t}_{\text{mix}} \leq |\bar{a}| \leq \max(|\mathcal{S}| \cdot D, \hat{t}_{\text{mix}} + D)$, and
  - Taking the actions in $\bar{a}$ starting at $s_{\text{curr.}}$ traverses a cycle. That is,

$$T'(T'(T'(...T'(s_{\text{curr.}}, \bar{a}_0), \bar{a}_1), \bar{a}_2), ..., \bar{a}_{|\bar{a}|-1})) = s_{\text{curr.}} \tag{53}$$

  and,
  - Taking the actions in $\bar{a}$ starting at $s_{\text{curr.}}$ visits all latent states in $s \in \mathcal{S}' \setminus \{s_{\text{curr.}}\}$ such that $|\mathcal{D}(s)| < d$ at least once.

Note that planning such a sequence $\bar{a}$ always must be possible. For example, starting at $s_{\text{curr.}}$, we can greedily plan a route to the nearest as-of-yet-unvisited latent state $s \in \mathcal{S}' \setminus \{s_{\text{curr.}}\}$ such that $|\mathcal{D}(s)| < d$ and repeat until all such states have been visited, and then navigate back to $s_{\text{curr.}}$. This takes at most $|\mathcal{S}'| \cdot D$ steps. If this sequence has length less than $\hat{t}_{\text{mix}}$, then we can insert a self-loop at any state in the sequence (such as the state with the shortest self-loop) and repeat this self-loop as many times as necessary until $|\bar{a}| \geq \hat{t}_{\text{mix}}$. Because all self-loops are of length at most $D + 1$, this can "overshoot" by at most $D$, so we have that $|\bar{a}| \leq \max(|\mathcal{S}| \cdot D, \hat{t}_{\text{mix}} + D)$.

- Execute the actions in $\bar{a}$ on $M$ once without collecting data, in order to ensure that within each set $\mathcal{D}(s)$, the newly-collected observations are collected at least $\hat{t}_{\text{mix}}$ steps after observations added in previous phases of STEEL.

- Repeatedly take the actions $\bar{a}$ on $M$, collecting the observation of each latent state $s$ the first time in the cycle that it is visited and inserting the observation into $\mathcal{D}(s)$, until $\forall s \in \mathcal{S}'$, $|\mathcal{D}(s)| \geq d$. Note that for a given latent state $s$, we collect observations of $s$ exactly $|\hat{a}|$ steps apart. Because $|\hat{a}| \geq \hat{t}_{\text{mix}}$, this ensures that the observation added to $\mathcal{D}(s)$ are collected at least $\hat{t}_{\text{mix}}$ steps apart. Because each $\mathcal{D}(s)$ will already contain at least one sample (from CycleFind), this process will take at most $d - 1$ iterations.

- Note that if for some state $s \in \mathcal{S}'$, $|\mathcal{D}(s)|$ reaches $d$ during some iteration of taking the actions $\bar{a}$, then for the next iteration, we can re-plan a shorter $\bar{a}$ that does not necessarily visit $s$. However, when we do this, we must execute the newly-planned cycle $\bar{a}$ once without collecting data, in order to ensure that all observation added to any particular $\mathcal{D}(s)$ are collected at least $\hat{t}_{\text{mix}}$ steps apart. This could require at most $|\mathcal{S}|$ additional iterations through some $\bar{a}$.

This process will ensure that $\forall s \in \mathcal{S}'$, $|\mathcal{D}(s)| \geq d$, in at most

$$(d - 1 + |\mathcal{S}|) \cdot \max(D + \hat{t}_{\text{mix}}, |\mathcal{S}| \cdot D) \text{ steps.} \tag{54}$$

Also, note that all samples collected during this phase are sorted into the appropriate dataset $\mathcal{D}(s)$ entirely by open-loop planning on $T'$, so the choice of samples in each $\mathcal{D}(s)$ remains independent of $\Omega$, and, in principle, can be a deterministic function of $s_{\text{init}}$.

### B.1.4 STEEL Phase 3

Finally, for each learned latent state $s \in \mathcal{S}'$, we train a classifier $f_s$ to distinguish $D_0 := \biguplus_{s' \in \mathcal{S}' \setminus \{s\}} \mathcal{D}(s')$ from $D_1 := \mathcal{D}(s)$. This set of classifiers allows us to perform one-versus-rest classification to identify the latent state of any observation $x$, by defining:

$$\phi'(x) := \arg\max_s f_s(x). \tag{55}$$

Along with the learned transition dynamics $T'$, this should be a sufficient representation of the latent dynamics.

We want to guarantee that when the *exogenous* state $e$ of the Ex-BMDP is at equilibrium (that is, is sampled from its stationary distribution), for any latent state $s$, if $x \sim \mathcal{Q}(s, e)$, then the probability that $f_s(x) = 1$ and, $\forall s' \neq s$, $f(s') = 0$ is at least $1 - \epsilon$. By union bound, we can do this by ensuring that the accuracy of *each* classifier $f_s$, on each latent state $s' \in \mathcal{S}$, is at least $1 - \epsilon/|\mathcal{S}|$. By realizability, we know that $\forall s$, there exists some classifier $f_s^* \in \mathcal{F}$ for which $f_s^*(s) = 1$ iff $\phi^*(x) = s$. Therefore, we need to upper-bound the probability that $\exists f' \in \mathcal{F}$, for which $\forall x \in D_1, f(x) = 1$ and $\forall x \in D_0, f(x) = 0$, but for which either

$$\Pr_{x \sim \mathcal{Q}(s,e); e \sim \pi} (f'(x) = 0) \geq \frac{\epsilon}{|\mathcal{S}|} \tag{56}$$

Or, for any $s' \neq s$,

$$\Pr_{x \sim \mathcal{Q}(s',e); e \sim \pi} (f'(x) = 1) \geq \frac{\epsilon}{|\mathcal{S}|}. \tag{57}$$

For all $s$, all samples in $\mathcal{D}(s)$ are collected at least $t_{\text{mix}}$ samples apart at timesteps chosen deterministically and independently of $\Omega$. Therefore, for any single fixed classifier $f$, we can use Lemma 2 and the fact that $\forall s, |\mathcal{D}(s)| \geq d$ to bound the false-positive rates in Equations 56 and 57 as:

$$\Pr \left( \forall x \in \mathcal{D}(s), f'(x) = 1 \bigwedge \Pr_{x \sim \mathcal{Q}(s,e); e \sim \pi} f'(x) = 0 \geq \frac{\epsilon}{|\mathcal{S}|} \right) \leq 8e^{-\frac{\epsilon \cdot d}{3|\mathcal{S}|}}. \tag{58}$$

and, $\forall s' \in \mathcal{S}' \setminus \{s\}$,

$$\Pr\left(\forall x \in \mathcal{D}(s'), f'(x) = 0 \bigwedge_{x \sim \mathcal{Q}(s',e); e \sim \pi} f'(x) = 1 \geq \frac{\epsilon}{|\mathcal{S}|}\right) \leq 8e^{-\frac{\epsilon \cdot d}{3|\mathcal{S}|}}. \tag{59}$$

Taking the union bound bound over $s$ and all latent states $s' \in \mathcal{S}' \setminus \{s\}$ gives a total false positive rate for learning $f'$ as $f_s$ as:

$$\text{FPR}(f', s) \leq 8|\mathcal{S}|e^{-\frac{\epsilon \cdot d}{3|\mathcal{S}|}}. \tag{60}$$

Taking the union bound over all $f \in \mathcal{F}$ gives:

$$\text{FPR}(f_s) \leq 8|\mathcal{S}||\mathcal{F}|e^{-\frac{\epsilon \cdot d}{3|\mathcal{S}|}}. \tag{61}$$

Finally, taking the union bound over each classifier $f_s$ gives:

$$\text{FPR} \leq 8|\mathcal{S}|^2|\mathcal{F}|e^{-\frac{\epsilon \cdot d}{3|\mathcal{S}|}}. \tag{62}$$

### B.1.5 BOUNDING THE OVERALL FAILURE RATE AND SAMPLE COMPLEXITY

Here, we bound the overall failure rate of the STEEL algorithm. We do this by separately bounding the failure rate of the first phase of the algorithm (the repeated applications of CycleFind) and the final phase of the algorithm, the learning of classifiers $f_s$. We let each of these failure rates be at most $\delta/2$. Therefore, we must have, over the at most $|\mathcal{S}| \cdot |\mathcal{A}|$ iterations of CycleFind, a failure rate of at most

$$\frac{\delta}{2} \geq |\mathcal{S}| \cdot |\mathcal{A}| \cdot \Pr(\text{CycleFind Fails}). \tag{63}$$

This is satisfied by Proposition 1 (noting that $N \geq |\mathcal{S}|$). The number of samples needed for these $|\mathcal{S}| \cdot |\mathcal{A}|$ iterations of CycleFind, each with $|\hat{a}| \leq |\mathcal{S}| \cdot (D + 1)$, is (by Equation 36) at most:

$$|\mathcal{S}| \cdot |\mathcal{A}| \cdot \Bigg( \max\Big( (2\hat{t}_{\text{mix}} + 3N \cdot |\mathcal{S}| \cdot (D + 1) - 2) \cdot n_{\text{samp. cyc.}} - N \cdot |\mathcal{S}| \cdot (D + 1),$$
$$2 \cdot (\hat{t}_{\text{mix}} + |\mathcal{S}|^2 \cdot (D + 1) - 1) \cdot n_{\text{samp.}} + 1 \Big) \tag{64}$$
$$+ \max((N - 1) \cdot |\mathcal{S}| \cdot (D + 1) - \hat{t}_{\text{mix}}, 0) + 1 \Bigg)$$

Which is upper-bounded by:

$$|\mathcal{S}| \cdot |\mathcal{A}| \cdot \Bigg( \max\Big( (2\hat{t}_{\text{mix}} + 3N \cdot |\mathcal{S}| \cdot (D + 1) - 2) \cdot n_{\text{samp. cyc.}},$$
$$2 \cdot (\hat{t}_{\text{mix}} + |\mathcal{S}|^2 \cdot (D + 1) - 1) \cdot n_{\text{samp.}} \tag{65}$$
$$+ 2 + \max((N - 1) \cdot |\mathcal{S}| \cdot (D + 1) - \hat{t}_{\text{mix}}, 0) \Big) \Bigg)$$

where $n_{\text{samp.}}$ is given by Equation 11 and $n_{\text{samp. cyc.}}$ is given by Equation 18. Meanwhile, the overall failure rate of the second phase is at most

$$\frac{\delta}{2} \geq 8|\mathcal{S}|^2|\mathcal{F}|e^{-\frac{\epsilon \cdot d}{3|\mathcal{S}|}}. \tag{66}$$

Solving for $d$ in Equation 66 gives:

$$\frac{3|\mathcal{S}|\ln(16|\mathcal{S}|^2|\mathcal{F}|/\delta)}{\epsilon} \leq d. \tag{67}$$

Which is indeed satisfied by Equation 52, given that the structure of the latent dynamics were correctly learned using CycleFind in the first phase of the algorithm, so that $|\mathcal{S}'| = |\mathcal{S}|$.

By Equation 54, we then know that the number of samples needed for this phase is at most:

$$\max(D + \hat{t}_{\text{mix}}, |\mathcal{S}| \cdot D) \cdot \left( \lceil \frac{3|\mathcal{S}| \ln(16|\mathcal{S}|^2|\mathcal{F}|/\delta)}{\epsilon} \rceil - 1 + |\mathcal{S}| \right). \tag{68}$$

Combining the number of samples over both phases and simplifying gives us an overall upper-bound of the number of required samples of:

$$\max(D + \hat{t}_{\text{mix}}, |\mathcal{S}| \cdot D) \cdot \left( \frac{3|\mathcal{S}| \ln(16|\mathcal{S}|^2|\mathcal{F}|/\delta)}{\epsilon} + |\mathcal{S}| \right) +$$

$$|\mathcal{S}| \cdot |\mathcal{A}| \cdot \Bigg( \max \Big( (2\hat{t}_{\text{mix}} + 3N \cdot |\mathcal{S}| \cdot (D+1) - 2) \cdot$$

$$(\ln \left( 4|\mathcal{A}| \cdot N \cdot (N-1) \cdot |\mathcal{F}|/\delta \right) / \ln(16/9) + 1), \tag{69}$$

$$2 \cdot (\hat{t}_{\text{mix}} + |\mathcal{S}|^2 \cdot (D+1) - 1) \cdot$$

$$(\ln \left( 4|\mathcal{A}| \cdot N^4 \cdot (\hat{D}+1) \cdot |\mathcal{F}|/\delta \right) / \ln(16/9) + 1)$$

$$+ 2 + \max((N-1) \cdot |\mathcal{S}| \cdot (D+1) - \hat{t}_{\text{mix}}, 0) \Big) \Bigg)$$

This gives us a big-O sample complexity of (using that $\hat{D} \leq N$ and $|\mathcal{S}| \leq N$ ):

$$\mathcal{O}\Big( |\mathcal{S}|^2 \cdot N \cdot D \cdot |\mathcal{A}| \cdot (\log |\mathcal{A}| + \log(N) + \log |\mathcal{F}| + \log(1/\delta)) +$$

$$|\mathcal{S}| \cdot |\mathcal{A}| \cdot \hat{t}_{\text{mix}} \cdot (\log |\mathcal{A}| + \log(N) + \log |\mathcal{F}| + \log(1/\delta)) +$$

$$|\mathcal{S}|^2 \cdot D \cdot (1/\epsilon) \cdot (\log(|\mathcal{S}|) + \log |\mathcal{F}| + \log(1/\delta)) + \tag{70}$$

$$|\mathcal{S}| \cdot \hat{t}_{\text{mix}} \cdot (1/\epsilon) \cdot (\log(|\mathcal{S}|) + \log |\mathcal{F}| + \log(1/\delta)) \Big)$$

Using the notation $\mathcal{O}^*(f(x)) := \mathcal{O}(f(x) \log(f(x)))$, we can write this as:

$$\mathcal{O}^* \left( ND|\mathcal{S}|^2|\mathcal{A}| \cdot \log \frac{|\mathcal{F}|}{\delta} + |\mathcal{S}||\mathcal{A}|\hat{t}_{\text{mix}} \cdot \log \frac{N|\mathcal{F}|}{\delta} + \frac{|\mathcal{S}|^2 D}{\epsilon} \cdot \log \frac{|\mathcal{F}|}{\delta} + \frac{|\mathcal{S}|\hat{t}_{\text{mix}}}{\epsilon} \cdot \log \frac{|\mathcal{F}|}{\delta} \right). \tag{71}$$

Therefore, we have shown that, with high probability, STEEL returns (up to permutation) the correct latent dynamics for the Ex-BMDP, and a high-accuracy latent-state encoder $\phi'$, within the sample-complexity bound stated in Theorem 1. This completes the proof.

## C  USEFUL LEMMATA

**Lemma 1.** *Consider an Ex-BMDP $M = \langle \mathcal{X}, \mathcal{A}, \mathcal{S}, \mathcal{E}, \mathcal{Q}, T, \mathcal{T}_e, \pi_{\mathcal{E}}^{init} \rangle$ starting at an arbitrary latent endogenous state $s_{init} \in \mathcal{S}$. Let $\Omega$ represent the sample space of the three sources of randomness in $M$: that is, $\mathcal{T}_e$, $\mathcal{Q}$, and the initial exogenous latent state $e_{init}$. Assume that all actions on $M$ are taken deterministically and independently of $\Omega$. Let $f \in \mathcal{X} \to \{0, 1\}$ be a fixed arbitrary function, and for each $s \in \mathcal{S}$ let*

$$p_s := \Pr(f(x) = 1 | x \sim \mathcal{Q}(s, e), e \sim \pi_{\mathcal{E}}). \tag{72}$$

*where $\pi_{\mathcal{E}}$ is the stationary distribution of $\mathcal{T}_e$. Consider a trajectory sampled from this Ex-BMDP denoted as $x_{traj} := x'_0, x'_1, x'_2...$, with endogenous latent states $s_{traj} := s'_0, s'_1, s'_2...$ (so that $s'_0 = s_{init}$), and exogenous states $e_{traj} := e'_0, e'_1, e'_2.... $. Then, for any fixed $t_1, t_2 \in \mathbb{N}$, selected independently of $\Omega$, where $t_2 - t_1 \geq t_{mix}$, we have that:*

$$p_s - 1/4 \leq \Pr(f(x'_{t_2}) = 1 | x'_{\leq t_1}, s'_{t_2} = s) \leq p_s + 1/4, \tag{73}$$

*where $x'_{\leq t_1}$ denotes the observations in the trajectory $x_{traj}$ up to and including $x'_{t_1}$.*

*Note that this does not necessarily hold if $t_1$, $t_2$ depend on $\Omega$.*

*Proof.* From the definition of mixing time:

$$\forall e \in \mathcal{E}, \| \Pr(e'_{t_2} = \cdot | e'_{t_1} = e) - \pi_{\mathcal{E}} \|_{\text{TV}} \leq \frac{1}{4}. \tag{74}$$

Then,

$$\forall e \in \mathcal{E}, \left| \Pr(f(x) = 1 | x \sim \mathcal{Q}(s'_{t_2}, e'), e' \sim \pi_{\mathcal{E}}) - \Pr_{x \sim \mathcal{Q}(s'_{t_2}, e'_{t_2})} (f(x) = 1 | e'_{t_1} = e) \right| \leq \frac{1}{4}. \tag{75}$$

Because $e'_{t_2}$ depends on $x'_{\leq t_1}$ only through $e'_{t_1}$, we have:

$$\left| \Pr(f(x) = 1 | x \sim \mathcal{Q}(s'_{t_2}, e'), e' \sim \pi_{\mathcal{E}}) - \Pr_{x \sim \mathcal{Q}(s'_{t_2}, e'_{t_2})} (f(x) = 1 | x'_{\leq t_1}) \right| \leq \frac{1}{4}. \tag{76}$$

Then by Equation 72,

$$\left| p_{s'_{t_2}} - \Pr_{x \sim \mathcal{Q}(s'_{t_2}, e'_{t_2})} (f(x) = 1 | x'_{\leq t_1}) \right| \leq \frac{1}{4}, \tag{77}$$

which directly implies Equation 73. Note that this is does not hold if $t_1$ and $t_2$ can depend on $\Omega$. For example, if we define $t_2 := (\min t \text{ such that } f(x'_t) = 0 \text{ and } t \geq t_1 + t_{\text{mix}})$, then trivially Equation 73 may not apply. $\qquad \square$

**Lemma 2.** *Consider an irreducible, aperiodic Markov chain $\mathcal{T}_e$ on states $\mathcal{E}$ with mixing time $t_{mix}$ and stationary distribution $\pi_{\mathcal{E}}$, and an arbitrary function $f : \mathcal{E} \to \{0, 1\}$. Suppose $\Pr_{e \sim \pi_{\mathcal{E}}}(f(e) = 1) \leq 1 - \epsilon$. Consider a fixed sequence of $N$ timesteps $t_1, ..., t_N$, where $\forall i, t_i - t_{i-1} \geq t_{mix}$. Now, for a trajectory $e_0, ..., e_{t_N}$ sampled from the Markov chain, starting at an arbitrary $e_0$, we have that:*

$$\Pr(\bigcap_{i=1}^{N} f(e_{t_i}) = 1) \leq 8 e^{-\frac{\epsilon \cdot N}{3}}. \tag{78}$$

*Proof.* Define $\epsilon'$ as:

$$\epsilon' := \Pr_{e \sim \pi_{\mathcal{E}}} (f(e) = 0). \tag{79}$$

Note that we know that $\epsilon' \geq \epsilon$. Now, fix any $i \geq t_{\text{mix}}$. Let $M^i$ denote the linear operator on state distributions corresponding to taking $i$ steps of the Markov chain: that is, $M^i \pi$ gives the distribution after $i$ time steps. Also, let $\Pi$ be the linear operator defined as:

$$\Pi \pi := \left( \int_{e \in \mathcal{E}} \pi(e) de \right) \pi_e. \tag{80}$$

Define the linear operator $\Delta^i$ as:

$$\Delta^i := M^i - \Pi. \tag{81}$$

By linearity and noting that both $M^i$ and $\Pi$ are stochastic operators, we have that, for any $\pi$,

$$\int_{e \in \mathcal{E}} (\Delta^i \pi)(e) de = 0. \tag{82}$$

Also, from the definition of mixing time, we have, for any function $\pi$:

$$\|\Delta^i \pi\|_1 \leq \frac{1}{2} \|\pi\|_1. \tag{83}$$

(To see this, for any $e \in \mathcal{E}$ consider the unit vector $\vec{e}$. Then, note that:

$$\|\Delta^i \vec{e}\|_1 = 2 \cdot \|\pi_e - M^i \vec{e}\|_{TV} \leq \frac{1}{2}. \tag{84}$$

Then, for any $\pi$, we have:

$$\|\Delta^i \pi\|_1 = \| \int_{e \in \mathcal{E}} \pi(e) \Delta^i \vec{e} de \|_1 \leq \int_{e \in \mathcal{E}} |\pi(e)| \|\Delta^i \vec{e}\|_1 de \leq \frac{1}{2} \|\pi\|_1.) \tag{85}$$

Because $\pi_e$ is a stationary distribution of $M^i$, we also have that:

$$\Delta^i \pi_e = (|M^i - \Pi)\pi_e = \pi_e - \pi_e = 0. \tag{86}$$

Additionally, let $\Gamma$ be the linear operator defined as:

$$\Gamma\pi := \int_{e \in \mathcal{E}} \pi(e) f(e) \vec{e} \, de \tag{87}$$

In other words, $(\Gamma\pi)(e) := f(e) \cdot \pi(e)$. One useful fact about this operator is that, for any function $\pi_0$ such that $\int_{e \in \mathcal{E}} \pi_0(e) de = 0$, we have that

$$\int_{e \in \mathcal{E}} (\Gamma\pi_0)(e) de \leq \frac{1}{2}\|\pi_0\|_1. \tag{88}$$

(To see this, note that we have:

$$\int_{e \in \mathcal{E}} (\Gamma\pi_0)(e) de + \int_{e \in \mathcal{E}} ((I - \Gamma)\pi_0)(e) de = 0, \tag{89}$$

and also that, because $\Gamma\pi_0$ and $(I - \Gamma)\pi_0$ are nonzero for disjoint $e$'s:

$$\|\Gamma\pi_0\|_1 + \|(I - \Gamma)\pi_0\|_1 = \|\pi_0\|_1. \tag{90}$$

Then, we also have:

$$\left| \int_{e \in \mathcal{E}} (\Gamma\pi_0)(e) de \right| \leq \|\Gamma\pi_0\|_1, \tag{91}$$

and

$$\left| \int_{e \in \mathcal{E}} (\Gamma\pi_0)(e) de \right| = \left| \int_{e \in \mathcal{E}} ((I - \Gamma)\pi_0)(e) de \right| \leq \|(I - \Gamma)\pi_0\|_1. \tag{92}$$

Combining these equations and inequalities yields Equation 88.)

Now, consider the operator $M^i\Gamma$. This operator, when applied to a probability distribution $\pi$, yields the *(unnormalized)* probability density that results from applying $\mathcal{T}_e$ to $e'$ $i$ times, where $e'$ is sampled from $\pi$ conditioned on $f(e') = 1$. More precisely, it is the density of $e \sim \mathcal{T}_e^i(e')$, scaled down by the probability that $f(e') = 1$. In other words, it is given by:

$$(M^i\Gamma\pi)(e) = p(e_i = e | e \sim \mathcal{T}_e^i(e') \wedge f(e') = 1 \wedge e' \sim \pi) \cdot \Pr_{e' \sim \pi}(f(e') = 1). \tag{93}$$

Now, consider any vector $v$. Note that $v$ always can be uniquely decomposed as follows:

$$v := a\pi_e + b\bar{v} \text{ where } \int_{e \in \mathcal{E}} \bar{v}(e) de = 0 \text{ and } \|\bar{v}\|_1 = 1 \text{ and } b \geq 0. \tag{94}$$

(Specifically, we must set $a := \int_{e \in \mathcal{E}} v(e) de$ and $b := \|v - a\pi_e\|_1$ and $\bar{v} := (v - a\pi_e)/b$.) Assume that $v$ is such that $a \geq 0$. Now, consider the equation:

$$v' = M^i\Gamma v \tag{95}$$

If we consider the above decomposition, we have:

$$a'\pi_e + b'\bar{v}' = M^i\Gamma(a\pi_e + b\bar{v}) \tag{96}$$

Note that this can be re-written as:

$$a'\pi_e + b'\bar{v}' = \Pi\Gamma(a\pi_e + b\bar{v}) + \Delta^i\Gamma(a\pi_e + b\bar{v}). \tag{97}$$

Note that by Equation 80, the image of $\Pi$ can be written in the form $a'\pi_e$, while, by Equation 82, the image of $\Delta^i$ can be written as $b'\bar{v}$, where $b'$ and $\bar{v}$ are constrained as in Equation 97. Then, we have (using Equation 80):

$$a' = \int_{e \in \mathcal{E}} (\Gamma(a\pi_e + b\bar{v}))(e) de = a \int_{e \in \mathcal{E}} (\Gamma\pi_e))(e) de + b \int_{e \in \mathcal{E}} (\Gamma\bar{v}))(e) de \tag{98}$$

and:

$$b' = \|\Delta^i\Gamma(a\pi_e + b\bar{v})\|_1 \leq a\|\Delta^i\Gamma\pi_e\|_1 + b\|\Delta^i\Gamma\bar{v}\|_1. \tag{99}$$

Now, from Equation 98, we have:

$$
\begin{aligned}
a' &= a \int_{e \in \mathcal{E}} (\Gamma \pi_e)(e) de + b \int_{e \in \mathcal{E}} (\Gamma \bar{v})(e) de \\
&\leq a \int_{e \in \mathcal{E}} (\Gamma \pi_e)(e) de + \frac{b}{2} \|\bar{v}\|_1 \qquad \text{(by Equation 88)} \\
&\leq a(1 - \epsilon') + \frac{b}{2} \|\bar{v}\|_1 \qquad \text{(by Equation 79)} \\
&\leq a(1 - \epsilon') + \frac{b}{2} \qquad \text{(by definition of } \bar{v})
\end{aligned}
\tag{100}
$$

And, from Equation 99, we have:

$$
\begin{aligned}
b' &\leq a\|\Delta^i \Gamma \pi_e\|_1 + b\|\Delta^i \Gamma \bar{v}\|_1 \\
&\leq a\|\Delta^i (I - (I - \Gamma))\pi_e\|_1 + b\|\Delta^i \Gamma \bar{v}\|_1 \\
&\leq a\|\Delta^i \pi_e\|_1 + a\|\Delta^i (I - \Gamma)\pi_e\|_1 + b\|\Delta^i \Gamma \bar{v}\|_1 \\
&\leq a\|\Delta^i (I - \Gamma)\pi_e\|_1 + b\|\Delta^i \Gamma \bar{v}\|_1 \qquad \text{(by Equation 86)} \\
&\leq \frac{a}{2}\|(I - \Gamma)\pi_e\|_1 + \frac{b}{2}\|\Gamma \bar{v}\|_1 \qquad \text{(by Equation 83)} \\
&\leq \frac{a\epsilon'}{2} + \frac{b}{2}\|\Gamma \bar{v}\|_1 \qquad \text{(by Equation 79)} \\
&\leq \frac{a\epsilon'}{2} + \frac{b}{2}\|\bar{v}\|_1 \qquad \text{(by Equation 90)} \\
&\leq \frac{a\epsilon'}{2} + \frac{b}{2} \qquad \text{(by definition of } \bar{v}.)
\end{aligned}
\tag{101}
$$

We can summarize these results as:

$$
\begin{bmatrix} a' \\ b' \end{bmatrix} \leq \begin{bmatrix} 1 - \epsilon' & \frac{1}{2} \\ \frac{\epsilon'}{2} & \frac{1}{2} \end{bmatrix} \begin{bmatrix} a \\ b \end{bmatrix}.
\tag{102}
$$

where there "$\leq$" sign applies elementwise. Also, because the elements of this matrix are all non-negative, we have:

$$
\begin{bmatrix} x' \\ y' \end{bmatrix} \leq \begin{bmatrix} x \\ y \end{bmatrix} \implies \begin{bmatrix} 1 - \epsilon' & \frac{1}{2} \\ \frac{\epsilon'}{2} & \frac{1}{2} \end{bmatrix} \begin{bmatrix} x' \\ y' \end{bmatrix} \leq \begin{bmatrix} 1 - \epsilon' & \frac{1}{2} \\ \frac{\epsilon'}{2} & \frac{1}{2} \end{bmatrix} \begin{bmatrix} x \\ y \end{bmatrix}.
\tag{103}
$$

Now, let $\pi_{t_1}$ represent the probability distribution of the Markov chain at timestep $t_1$, and $\pi_{t_N + t_{\text{mix}}}$ be the probability distribution at timestep $t_N + t_{\text{mix}}$. Applying Equation 93 repeatedly gives us that:

$$
\begin{aligned}
M^{t_{\text{mix}}} \Gamma M^{t_N - t_{N-1}} \Gamma M^{t_{N-1} - t_{N-2}} \Gamma ... M^{t_2 - t_1} \Gamma \pi_{t_1} &= \\
\pi_{t_N + t_{\text{mix}}} \cdot \Pr(f(e_{t_1}) = 1) \cdot \Pr(f(e_{t_2}) = 1 | f(e_{t_1}) = 1) ... \\
\Pr(f(e_{t_{N-1}}) = 1 | \cap_{i=1}^{N-2} f(e_{t_i}) = 1) \cdot \Pr(f(e_{t_N}) = 1 | \cap_{i=1}^{N-1} f(e_{t_i}) = 1)
\end{aligned}
\tag{104}
$$

This gives us that:

$$
\int_{e \in \mathcal{E}} (M^{t_{\text{mix}}} \Gamma M^{t_N - t_{N-1}} \Gamma ... M^{t_2 - t_1} \Gamma \pi_{t_1})(e) de = \Pr(\cap_{i=1}^{N} f(e_{t_i}) = 1)
\tag{105}
$$

where the right-hand side is the probability that we are ultimately trying to bound.

Now, let $v_0 := \pi_{t_1}$; for $1 \leq i \leq N-1$, let $v_i := M^{t_{i+1} - t_i} \Gamma M^{t_i - t_{i-1}} \Gamma ... \Gamma \pi_{t_1}$; and finally let $v_N := M^{t_{\text{mix}}} \Gamma M^{t_N - t_{N-1}} \Gamma ... \Gamma \pi_{t_1}$. Let $a_i$ and $b_i$ represent the components $a$ and $b$ in the decomposition given in Equation 94 of $v_i$. Note that:

$$
a_N = \int_{e \in \mathcal{E}} (M^{t_{\text{mix}}} \Gamma M^{t_N - t_{N-1}} \Gamma ... M^{t_2 - t_1} \Gamma \pi_{t_1})(e) de = \Pr(\cap_{i=1}^{N} f(e_{t_i}) = 1).
\tag{106}
$$

Also, note that $\forall j \in [1, N]$, we have that $v_j = M^i \Gamma v_{j-1}$, for some $i \geq t_{\text{mix}}$. Therefore, by Equation 102,

$$
\begin{bmatrix} a_i \\ b_i \end{bmatrix} \leq \begin{bmatrix} 1 - \epsilon' & \frac{1}{2} \\ \frac{\epsilon'}{2} & \frac{1}{2} \end{bmatrix} \begin{bmatrix} a_{i-1} \\ b_{i-1} \end{bmatrix}.
\tag{107}
$$

(Additionally, each $a_i$ represents a probability, and so $a_i \geq 0$, so Equation 102 is applicable.) Now, due to the relation shown in Equation 103, we can apply this inequality recursively:

$$
\begin{bmatrix} a_i \\ b_i \end{bmatrix} \leq \begin{bmatrix} 1 - \epsilon' & \frac{1}{2} \\ \frac{\epsilon'}{2} & \frac{1}{2} \end{bmatrix} \begin{bmatrix} a_{i-1} \\ b_{i-1} \end{bmatrix} \bigwedge \begin{bmatrix} a_{i-1} \\ b_{i-1} \end{bmatrix} \leq \begin{bmatrix} 1 - \epsilon' & \frac{1}{2} \\ \frac{\epsilon'}{2} & \frac{1}{2} \end{bmatrix} \begin{bmatrix} a_{i-2} \\ b_{i-2} \end{bmatrix} \implies
$$

$$
\begin{bmatrix} a_i \\ b_i \end{bmatrix} \leq \begin{bmatrix} 1 - \epsilon' & \frac{1}{2} \\ \frac{\epsilon'}{2} & \frac{1}{2} \end{bmatrix} \begin{bmatrix} a_{i-1} \\ b_{i-1} \end{bmatrix} \bigwedge
$$

$$
\begin{bmatrix} 1 - \epsilon' & \frac{1}{2} \\ \frac{\epsilon'}{2} & \frac{1}{2} \end{bmatrix} \begin{bmatrix} a_{i-1} \\ b_{i-1} \end{bmatrix} \leq \begin{bmatrix} 1 - \epsilon' & \frac{1}{2} \\ \frac{\epsilon'}{2} & \frac{1}{2} \end{bmatrix} \begin{bmatrix} 1 - \epsilon' & \frac{1}{2} \\ \frac{\epsilon'}{2} & \frac{1}{2} \end{bmatrix} \begin{bmatrix} a_{i-2} \\ b_{i-2} \end{bmatrix} \implies
$$

$$
\begin{bmatrix} a_i \\ b_i \end{bmatrix} \leq \begin{bmatrix} 1 - \epsilon' & \frac{1}{2} \\ \frac{\epsilon'}{2} & \frac{1}{2} \end{bmatrix} \begin{bmatrix} a_{i-1} \\ b_{i-1} \end{bmatrix} \leq \begin{bmatrix} 1 - \epsilon' & \frac{1}{2} \\ \frac{\epsilon'}{2} & \frac{1}{2} \end{bmatrix} \begin{bmatrix} 1 - \epsilon' & \frac{1}{2} \\ \frac{\epsilon'}{2} & \frac{1}{2} \end{bmatrix} \begin{bmatrix} a_{i-2} \\ b_{i-2} \end{bmatrix} \tag{108}
$$

So that we have:

$$
\begin{bmatrix} a_N \\ b_N \end{bmatrix} \leq \left( \begin{bmatrix} 1 - \epsilon' & \frac{1}{2} \\ \frac{\epsilon'}{2} & \frac{1}{2} \end{bmatrix} \right)^N \begin{bmatrix} a_0 \\ b_0 \end{bmatrix}. \tag{109}
$$

The matrix $\begin{bmatrix} 1 - \epsilon' & \frac{1}{2} \\ \frac{\epsilon'}{2} & \frac{1}{2} \end{bmatrix}$ has eigenvalues $\frac{(\frac{3}{2} - \epsilon') \pm \sqrt{\epsilon'^2 + \frac{1}{4}}}{2}$; we can use the closed-form solution to the $N$th power of an arbitrary $2 \times 2$ matrix given by Williams (1992) to exactly write this upper-bound on $a_N$:

$$
a_N \leq \left[ \left( \frac{(\frac{3}{2} - \epsilon') + \sqrt{\epsilon'^2 + \frac{1}{4}}}{2} \right)^N \frac{1}{\sqrt{\epsilon'^2 + \frac{1}{4}}} \begin{bmatrix} 1 - \epsilon' - \frac{(\frac{3}{2} - \epsilon') - \sqrt{\epsilon'^2 + \frac{1}{4}}}{2} \\ \frac{1}{2} \end{bmatrix}^T \right.
$$

$$
\left. - \left( \frac{(\frac{3}{2} - \epsilon') - \sqrt{\epsilon'^2 + \frac{1}{4}}}{2} \right)^N \frac{1}{\sqrt{\epsilon'^2 + \frac{1}{4}}} \begin{bmatrix} 1 - \epsilon' - \frac{(\frac{3}{2} - \epsilon') + \sqrt{\epsilon'^2 + \frac{1}{4}}}{2} \\ \frac{1}{2} \end{bmatrix}^T \right] \begin{bmatrix} 1 \\ b_0 \end{bmatrix}.
$$

Where we are also using that $\pi_{t_1}$ is a normalized probability distribution, so $a_0 = 1$. Simplifying:

$$
a_N \leq \left[ \left( \frac{(\frac{3}{2} - \epsilon') + \sqrt{\epsilon'^2 + \frac{1}{4}}}{2} \right)^N \frac{1}{\sqrt{\epsilon'^2 + \frac{1}{4}}} \left( \frac{1}{4} - \frac{3\epsilon'}{2} + \frac{\sqrt{\epsilon'^2 + \frac{1}{4}}}{2} + \frac{b_0}{2} \right) \right.
$$

$$
\left. - \left( \frac{(\frac{3}{2} - \epsilon') - \sqrt{\epsilon'^2 + \frac{1}{4}}}{2} \right)^N \frac{1}{\sqrt{\epsilon'^2 + \frac{1}{4}}} \left( \frac{1}{4} - \frac{3\epsilon'}{2} - \frac{\sqrt{\epsilon'^2 + \frac{1}{4}}}{2} + \frac{b_0}{2} \right) \right]
$$

This gives us:

$$
a_N \leq \left[ \left| \frac{(\frac{3}{2} - \epsilon') + \sqrt{\epsilon'^2 + \frac{1}{4}}}{2} \right|^N \frac{1}{\sqrt{\epsilon'^2 + \frac{1}{4}}} \left| \frac{1}{4} - \frac{3\epsilon'}{2} + \frac{\sqrt{\epsilon'^2 + \frac{1}{4}}}{2} + \frac{b_0}{2} \right| \right.
$$

$$
\left. + \left| \frac{(\frac{3}{2} - \epsilon') - \sqrt{\epsilon'^2 + \frac{1}{4}}}{2} \right|^N \frac{1}{\sqrt{\epsilon'^2 + \frac{1}{4}}} \left| \frac{1}{4} - \frac{3\epsilon'}{2} - \frac{\sqrt{\epsilon'^2 + \frac{1}{4}}}{2} + \frac{b_0}{2} \right| \right] \tag{110}
$$

Note that because $\pi_{t_1}$ is a normalized probability distribution, we have that $b_0$ is at most 2 (with this maximum achieved when $\pi_{t_1}$ has disjoint support from $\pi_e$.) Numerically one can see that, for $\epsilon' \in [0, 1]$,

$$
-.7 < \frac{1}{4} - \frac{3\epsilon'}{2} + \frac{\sqrt{\epsilon'^2 + \frac{1}{4}}}{2} \leq .5 \tag{111}
$$

$$
-1.9 < \frac{1}{4} - \frac{3\epsilon'}{2} - \frac{\sqrt{\epsilon'^2 + \frac{1}{4}}}{2} \leq 0 \tag{112}
$$

Additionally, we can bound $\frac{1}{\sqrt{\epsilon'^2 + \frac{1}{4}}} \leq 2$. Then this gives us:

$$a_N \leq \left[ 4 \left| \frac{(\frac{3}{2} - \epsilon') + \sqrt{\epsilon'^2 + \frac{1}{4}}}{2} \right|^N + 4 \left| \frac{(\frac{3}{2} - \epsilon') - \sqrt{\epsilon'^2 + \frac{1}{4}}}{2} \right|^N \right] \tag{113}$$

Because, for $\epsilon' \in [0, 1]$, $\left| \frac{(\frac{3}{2} - \epsilon') - \sqrt{\epsilon'^2 + \frac{1}{4}}}{2} \right| < \left| \frac{(\frac{3}{2} - \epsilon') + \sqrt{\epsilon'^2 + \frac{1}{4}}}{2} \right|$, and using Equation 106, we can bound:

$$\Pr(\bigcap_{i=1}^{N} f(e_{t_i}) = 1) \leq 8 \left( \frac{(\frac{3}{2} - \epsilon') + \sqrt{\epsilon'^2 + \frac{1}{4}}}{2} \right)^N \tag{114}$$

This bound is clearly somewhat unwieldy; to obtain a more manageable bound, it is helpful to consider the asymptotic behavior near $\epsilon' = 0$. Letting $\delta := \Pr(\bigcap_{i=1}^{N} f(e_{t_i}) = 1)$, we have:

$$\ln \delta \leq \ln 8 + N \ln \frac{(\frac{3}{2} - \epsilon') + \sqrt{\epsilon'^2 + \frac{1}{4}}}{2} \tag{115}$$

For $\epsilon'$ small, we have $\epsilon'^2 \ll \frac{1}{4}$, so $\sqrt{\epsilon'^2 + \frac{1}{4}} \approx \frac{1}{2}$. Then:

$$\ln \delta \lesssim \ln 8 + N \ln \left( 1 - \frac{\epsilon'}{2} \right) \tag{116}$$

Then, using the standard approximation $\ln(1 - x) \approx -x$ gives us:

$$\ln \delta \lesssim \ln 8 - \frac{N\epsilon'}{2}. \tag{117}$$

This approximation would give us $\delta \lesssim 8 e^{-\frac{N\epsilon'}{2}}$. However, while this holds approximately for small $\epsilon'$, it does not hold exactly. Despite this, it does suggest a form for our final bound. If we try $8 e^{-\frac{N\epsilon'}{3}}$, we find that it holds that:

$$\frac{(\frac{3}{2} - \epsilon') + \sqrt{\epsilon'^2 + \frac{1}{4}}}{2} \leq e^{-\frac{\epsilon'}{3}} \text{ on the interval } 0 \leq \epsilon' \leq 0.44. \tag{118}$$

Combining with Equation 114, this implies that

$$\delta \leq 8 e^{-\frac{N\epsilon'}{3}} \text{ on the interval } 0 \leq \epsilon' \leq 0.44. \tag{119}$$

For very large $\epsilon'$, we can use a much simpler bound on $\Pr(\bigcap_{i=1}^{N} f(e_{t_i}) = 1)$. Recall that:

$$\delta = \Pr\left( \bigcap_{i=1}^{N} f(e_{t_i}) = 1 \right) = \Pr\left( f(e_{t_N}) = 1 \middle| \bigcap_{i=1}^{N-1} f(e_{t_i}) = 1 \right)$$

$$\cdot \Pr\left( f(e_{t_{N-1}}) = 1 \middle| \bigcap_{i=1}^{N-2} f(e_{t_i}) = 1 \right) \tag{120}$$

$$\cdot \ldots$$

$$\cdot \Pr(f(e_{t_2}) = 1 | f(e_{t_1}) = 1) \cdot \Pr(f(e_{t_1}) = 1)$$

However, because $\forall i,\ t_i - t_{i-1} \geq t_{\text{mix}}$, we have that:

$$\forall i > 1, \ \Pr\left( f(e_{t_i}) = 1 \middle| \bigcap_{j=1}^{i-1} f(e_{t_j}) = 1 \right) \leq \Pr_{e \sim \pi_e} (f(e) = 1) + \frac{1}{4} = \frac{5}{4} - \epsilon' \tag{121}$$

Combining these equations gives us:

$$\delta \leq \left( \frac{5}{4} - \epsilon' \right)^{N-1} \cdot \Pr(f(e_{t_1}) = 1)$$

$$\leq \left( \frac{5}{4} - \epsilon' \right)^{N-1} = \left( \frac{5}{4} - \epsilon' \right)^{-1} \left( \frac{5}{4} - \epsilon' \right)^N \leq 8 \left( \frac{5}{4} - \epsilon' \right)^N \tag{122}$$

Where we used that $\epsilon' \leq 1$ in the last step. Finally, it holds that:

$$\left(\frac{5}{4} - \epsilon'\right) \leq e^{-\frac{\epsilon'}{3}} \text{ on the interval } 0.37 \leq \epsilon' \leq 1. \tag{123}$$

This then implies that:

$$\delta \leq 8e^{-\frac{N\epsilon'}{3}} \text{ on the interval } 0.37 \leq \epsilon' \leq 1. \tag{124}$$

Combining with Equation 119, we have that

$$\forall \epsilon' \in [0, 1], \quad \Pr\left(\bigcap_{i=1}^{N} f(e_{t_i})\right) \leq 8e^{-\frac{N\epsilon'}{3}} \tag{125}$$

Because $\epsilon' \geq \epsilon$, we have:

$$\Pr\left(\bigcap_{i=1}^{N} f(e_{t_i})\right) \leq 8e^{-\frac{N\epsilon'}{3}} \leq 8e^{-\frac{N\epsilon}{3}}. \tag{126}$$

which was to be proven. $\qquad\square$

## D    UPPER-BOUNDING MIXING TIMES FOR EXAMPLES

Here, we prove that the values of $\hat{t}_{\text{mix}}$ used in the simulation experiments are in fact (somewhat loose) upper bounds on the true mixing times of $\mathcal{T}_e$ for these environments. While in practice, the true mixing times would not be known *a priori*, it is important for the validity of our examples that the true $t_{\text{mix}}$ is in fact $\leq \hat{t}_{\text{mix}}$.

We use the following following well-known fact:

For distributions $\mathcal{A} := \mathcal{A}_1 \otimes \mathcal{A}_2 \otimes ... \otimes \mathcal{A}_n$ and $\mathcal{B} := \mathcal{B}_1 \otimes \mathcal{B}_2 \otimes ... \otimes \mathcal{B}_n$:

$$\|\mathcal{A} - \mathcal{B}\|_{TV} \leq \sum_{i=1}^{n} \|\mathcal{A}_i - \mathcal{B}_i\|_{TV} \tag{127}$$

First, we deal with the combination lock experiment. We can write the transition matrix for any arbitrary two-state Markov chain as

$$\begin{bmatrix} 1 - \epsilon_0 & \epsilon_1 \\ \epsilon_0 & 1 - \epsilon_1 \end{bmatrix} \tag{128}$$

where $0 \leq \{\epsilon_0, \epsilon_1\} \leq 1$. Note that in our particular example, we have $0.1 \leq \{\epsilon_0, \epsilon_1\} \leq 0.9$.

This matrix has eigenvalues $1$ and $1 - \epsilon_0 - \epsilon_1$, and the stationary distribution (the eigenvector corresponding to the eigenvalue 1) is

$$\pi_\infty := [\epsilon_1/(\epsilon_0 + \epsilon_1), \epsilon_0/(\epsilon_0 + \epsilon_1)]^T. \tag{129}$$

Using the closed-form formula for the $n$'th power of a two-state Markov Chain given by Williams (1992), we have:

$$\left(\begin{bmatrix} 1 - \epsilon_0 & \epsilon_1 \\ \epsilon_0 & 1 - \epsilon_1 \end{bmatrix}\right)^n = \frac{1}{\epsilon_0 + \epsilon_1}\left[\begin{bmatrix} \epsilon_1 & \epsilon_1 \\ \epsilon_0 & \epsilon_0 \end{bmatrix} - (1 - \epsilon_0 - \epsilon_1)^n \begin{bmatrix} -\epsilon_0 & \epsilon_1 \\ \epsilon_0 & -\epsilon_1 \end{bmatrix}\right] \tag{130}$$

To compute the mixing time, we compute the state distribution $\pi_n$, $n$ timesteps after each starting state:

$$\left(\begin{bmatrix} 1 - \epsilon_0 & \epsilon_1 \\ \epsilon_0 & 1 - \epsilon_1 \end{bmatrix}\right)^n \begin{bmatrix} 1 \\ 0 \end{bmatrix} = \frac{1}{\epsilon_0 + \epsilon_1}\begin{bmatrix} \epsilon_1 + (1 - \epsilon_0 - \epsilon_1)^n \epsilon_0 \\ \epsilon_0 - (1 - \epsilon_0 - \epsilon_1)^n \epsilon_0 \end{bmatrix} \tag{131}$$

and,

$$\left(\begin{bmatrix} 1 - \epsilon_0 & \epsilon_1 \\ \epsilon_0 & 1 - \epsilon_1 \end{bmatrix}\right)^n \begin{bmatrix} 0 \\ 1 \end{bmatrix} = \frac{1}{\epsilon_0 + \epsilon_1}\begin{bmatrix} \epsilon_1 - (1 - \epsilon_0 - \epsilon_1)^n \epsilon_1 \\ \epsilon_0 + (1 - \epsilon_0 - \epsilon_1)^n \epsilon_1 \end{bmatrix} \tag{132}$$

Note that the TV distance between either of these distributions and the stationary distribution $\pi$ is at most

$$\|\pi_n - \pi_\infty\|_{TV} \leq \frac{|(1 - \epsilon_0 - \epsilon_1)|^n \max(\epsilon_0, \epsilon_1)}{(\epsilon_0 + \epsilon_1)} \leq |(1 - \epsilon_0 - \epsilon_1)|^n. \tag{133}$$

The parameters $\{\epsilon_0, \epsilon_1\}$ for each two-state Markov chain are chosen uniformly at random, such that $0.1 \leq \{\epsilon_0, \epsilon_1\} \leq 0.9$. Therefore, $0 \leq |(1 - \epsilon_0 - \epsilon_1)| \leq 0.8$. Then, for any individual chain, we have:

$$\|\pi_n - \pi_\infty\|_{TV} \leq |(1 - \epsilon_0 - \epsilon_1)|^n \leq 0.8^n \tag{134}$$

In the combination lock experiments, there are up to $L = 512$ of these noise Markov chains; the probability distribution over the exogenous noise $\mathcal{E}$ is the *product distribution* over these chains. Then we use Equation 127 to bound the *total* TV distance for the chain $\mathcal{T}_e$ to its stationary distribution $\pi_{\mathcal{E}}$; that is:

$$\|\pi_n^{\text{total}} - \pi_{\mathcal{E}}\|_{TV} \leq 512 \cdot 0.8^n. \tag{135}$$

Then, by the definition of mixing time, we have:

$$t_{\text{mix}} \leq \min n, \text{ such that } 512 \cdot 0.8^n \leq \frac{1}{4} \tag{136}$$

Which gives us:

$$t_{\text{mix}} \leq \left\lceil \frac{-\log(2048)}{\log(0.8)} \right\rceil = 35. \tag{137}$$

So the value that we use in the experiment, $\hat{t}_{\text{mix}} = 40$, is a valid upper bound.

For the multi-maze experiment, the exogenous noise state consists of eight identical mazes, with agents moving uniformly at random in each of them. Unlike the "combination lock" example, where the individual components of the exogenous noise are conditioned on parameters $\epsilon_0, \epsilon_1$, which can vary, in the multi-maze example the individual mazes always represent instances of *exactly the same, specific Markov chain.* Let the transition matrix of this chain be $M$, with stationary distribution $\pi_M$. Then, by Equation 127, for the whole exogenous state $\mathcal{T}_e$, we have:

$$
\begin{aligned}
t_{\text{mix}} &= \min n, \text{ such that } \forall s_0, \|(\mathcal{T}_e)^n s_0 - \pi_{\mathcal{E}}\| \leq \frac{1}{4} \\
&= \min n, \text{ such that } \forall s_0^{(1)}, s_0^{(2)}, ...s_0^{(8)}, \left\| M^n s_0^{(1)} \otimes M^n s_0^{(2)} \otimes ... \otimes M^n s_0^{(8)} - \right. \\
&\qquad\qquad\qquad\qquad\qquad\qquad \left. \pi_M \otimes \pi_M \otimes ... \otimes \pi_M \right\| \leq \frac{1}{4} \\
&\leq \min n, \text{ such that } \forall s_0, \sum_{i=1}^{8} \|M^n s_0 - \pi_M\| \leq \frac{1}{4} \\
&= \min n, \text{ such that } \forall s_0, \|M^n s_0 - \pi_M\| \leq \frac{1}{32}
\end{aligned} \tag{138}
$$

Which is to say that $t_{mix}$ for the entire exogenous noise chain is upper-bounded by $t_{mix}(1/32)$ for the individual maze chain $M$. Furthermore, while the state space for the entire chain is of size $|\mathcal{E}| = 68^8$, the individual maze chain $M$ operates on a state of size 68. This is small enough that it is tractable to exactly compute $t_{mix}(1/32)$ for $M$ using numerical techniques. We performed the computation (source code is provided with the supplementary materials), and found that

$$\min n, \text{ such that } \forall s_0, \|M^n s_0 - \pi_M\| \leq \frac{1}{32} = 293. \tag{139}$$

Therefore, for the full exogenous noise chain $\mathcal{T}_e$, we have that $t_{mix} \leq 293$. Then the value that we use in the experiment, $\hat{t}_{\text{mix}} = 300$, is a valid upper bound.

# E  Discussion of assumptions

In this section, we argue that two major assumptions of the STEEL algorithm are *necessary* assumptions. In other words, if we remove these assumptions, then, we argue, achieving similar guarantees to our algorithm would become effectively impossible for *any* algorithm to accomplish in the single-trajectory, no-resets Ex-BMDP setting (at least without making *additional* assumptions to compensate). These assumptions are (1) reachability: the assumption that all endogenous states $s \in \mathcal{S}$ are reachable from one another in $T$; and (2) the availability of a known upper bound on the exogenous state mixing time $t_{\mathrm{mix}}$.

## E.1  Reachability

Here, we argue that *any* algorithm for single-trajectory, no-resets representation learning of Ex-BMDPs must *necessarily* make a reachability assumption: that all endogenous latent states must be reachable from one another. At a high level, we argue that, if all latent states are *not* reachable from one another, then an algorithm must visit states in a particular order in order to explore the entire dynamics with a single trajectory. However, because the latent dynamics are not known *a priori*, it is impossible for an algorithm to guarantee that it visits states in the appropriate order.

More formally, recall that the reachability assumption is that for any pair of latent states $s_1, s_2$, there exists both a path from $s_1$ to $s_2$ and a path from $s_2$ to $s_1$. Consider any Ex-BMDP $M$ where this condition does not hold, such that $|\mathcal{A}| \geq 2$, and any learning algorithm $A$. Firstly, if there is any pair of states $s_1$ and $s_2$ where *neither* can reach the other, then clearly a single trajectory is insufficient to learn the Ex-BMDP, because it cannot visit both $s_1$ and $s_2$. Therefore, we restrict to the case where, for each pair of states, either both are reachable from each other, or (without loss of generality) $s_2$ is reachable from $s_1$ but $s_1$ is not reachable from $s_2$.

Consider every edge $(s, a, s')$ of the state transition graph defined by the state transition function $T$. If, for all such edges, $s$ is reachable from $s'$, then the reachability assumption holds on the entire dynamics (because all single "steps" are invertible by some sequence of actions, so for any pair $s_1, s_2$, the path from $s_1$ to $s_2$ implies the existence of a path from $s_2$ to $s_1$). Therefore, if reachability does not hold, then there exists some edge $(s, a, s')$ such that $s$ is not reachable from $s'$. Now, consider the first time that the algorithm $A$ encounters the state $s$. Regardless of the details of $A$, there must exist some action $a'$ such that, on this first encounter, the probability of $A$ taking action $a'$ is at least $1/|\mathcal{A}|$.

Therefore, if $a' = a$, then probability that the algorithm $A$ never revisits $s$, and so never takes any other action from $s$ apart from $a'$, is at least $1/|\mathcal{A}|$. Then, with substantial probability, the algorithm $A$ never explores the $|\mathcal{A}| - 1$ other possible transitions from $s$, and cannot possibly learn the full dynamics of the Ex-BMDP. (To be more precise, the algorithm's output will not depend *at all* on the ground-truth value of $T(s, a'')$ for $a'' \in \mathcal{A} \setminus \{a\}$, and so is highly unlikely to return the correct values for these transitions on arbitrary Ex-BMDPs.)

Alternatively, if $a' \neq a$, then consider the alternative Ex-BMDP $M'$, which is identical to $M$ in every way (in terms of dynamics, exogenous state, emission function, etc.), except that the effects of actions $a$ and $a'$ on the latent state $s$ are swapped. Note that before first encountering the latent state $s$, the MDPs $M$ and $M'$ will produce identically-distributed sequences of observations, so the algorithm $A$ will behave identically on them, and have identical internal memory/state. Then, when first encountering $s$, the algorithm $M$ on $A$ will take action $a'$ with substantial probability, and then transition to $s'$ and be unable to revisit $s$.

Therefore, for any Ex-BMDP $M$ that violates reachability, any algorithm $A$ is either likely to fail on $M$, or to fail on a slightly-modified version of $M$. In any case, no such algorithm will be able to succeed with high probability on any general class of Ex-BMDPs that does not require reachability.

## E.2  Known upper bound on the mixing time $t_{\mathrm{MIX}}$

Here, we argue that the assumption that an upper bound on the mixing time is provided is also necessary. We argue that:

1. Any general, provably sample-efficient algorithm for learning, with high probability, an Ex-BMDP from a single trajectory must necessarily require an upper bound on the mixing time of the exogenous noise to be provided *a priori* (unless some other information about the exogenous noise is provided.)

2. The runtime of any algorithm for learning an Ex-BMDP must, in the worst case, include some term that scales at least linearly with the mixing time of the exogenous noise.

At a high level, we argue that, with a single trajectory, it can be impossible to distinguish between a static exogenous dynamics (i.e., where only one exogenous state exists) and an extremely slow-mixing exogenous dynamics, in time sublinear in the mixing time. For any proposed algorithm that does not have access to a bound on the mixing time, and any Ex-BMDP with a single exogenous state (that is, a Block MDP), we can construct an extremely slow-mixing Ex-BMDP that behaves identically to the Block MDP with substantial probability during the entire duration of the runtime of the learning algorithm, but which at equilibrium has a quite different distribution of observations. This will make the learned encoder fail on the Ex-BMDP at equilibrium.

More precisely, to show (1): suppose the converse is true: that there exists some algorithm $A$ that learns Ex-BMDP latent dynamics and state encoders, which is not provided any upper-bound on the mixing time of the exogenous noise of the Ex-BMDP, or any other information about the exogenous noise distribution. We assume that $A$ can learn the correct endogenous dynamics, as well as an encoder with accuracy (for every endogenous state, under the stationary exogenous distribution) of at least $1 - \epsilon$, with probability at least $1 - \delta$, for small values of $\epsilon$ and $\delta$. Then, consider any Ex-BMDP $M_1$ and related parameterized family of Ex-BMDPs $M_2(\gamma), M_3(\gamma)$ with the following properties:

- $M_1$ has N endogenous latent states $s_1, ..., s_N$ with some transition function $T$, a single exogenous latent state $e_1$, and an emission function $\mathcal{Q}(s, e_1)$. (In other words, $M_1$ is any Block MDP).

- $M_2(\gamma)$ has the same endogenous states and transition probabilities as $M_1$, but has two exogenous latent states $e_1, e_2$. The state $e_1$ transitions to $e_2$, and $e_2$ transitions to $e_1$, each with probability $\gamma$. Note that the stationary distribution of the exogenous state is uniform over $e_1$ and $e_2$. We also assume that the initial exogenous state distribution is uniform over $e_1$ and $e_2$. Regardless of $\gamma$, the emission function of $M_2(\gamma)$ is defined such that $\forall s_i, \mathcal{Q}_{M_2}(s_i, e_1) = \mathcal{Q}_{M_1}(s_i, e_1)$.

- $M_3(\gamma)$ is identical to $M_2(\gamma)$, except that its emission function is defined as $\forall s_i, \mathcal{Q}_{M_3}(s_i, e_1) := \mathcal{Q}_{M_1}(s_i, e_1)$, but $\forall s_i, \mathcal{Q}_{M_3}(s_i, e_2) := \mathcal{Q}_{M_2}(s_{(i+1)\%N}, e_2)$. In other words, when the exogenous state is equal to $e_2$, the emission distributions for the endogenous states are permuted in $M_3$ compared to $M_2$.

We also assume that the encoder hypothesis class can represent the inverses of $\mathcal{Q}_{M_1}$, $\mathcal{Q}_{M_2}$, and $\mathcal{Q}_{M_3}$. However, note that by construction, any fixed encoder $\phi(\cdot)$ which has accuracy at least $1 - \epsilon$ on $M_2(\gamma)$ (for every endogenous state, under the stationary exogenous distribution) can only have accuracy at most $0.5 + \epsilon$ on $M_3(\gamma)$, because, when the exogenous state is $e_2$, any time that the encoder returns the correct latent state for $M_2(\gamma)$, it will return the incorrect latent state for $M_3(\gamma)$.[8]

Now, consider what happens when we run the algorithm $A$ on $M_1$. The number of environment steps that $A$ takes on $M_1$ forms some distribution; let $t$ be the 90'th percentile of this distribution, so that with probability 0.9, $A$ stops sampling before step $t$.

Now, we can set $\gamma := 1 - 0.9^{1/t}$, so that, with probability 0.9, the endogenous state of $M_2(\gamma)$ or $M_3(\gamma)$ do not change before step $t$. Therefore, by union bound, on $M_2(\gamma)$ or $M_3(\gamma)$, if the initial exogenous state is $e_1$, then with probability at least 0.8, the exogenous state will stay constant at $e_1$

---

[8]There is some nuance involved here: note that for an encoder $\phi$ to achieve a high accuracy, it only needs to approximate $\sigma(\phi^*(\cdot))$ under *some choice of* permutation $\sigma$. However, we know that for any $s_i, s_{(i+1)\%N} \in \mathcal{S}$, the $\phi$ which is highly accurate on $M_2(\gamma)$ must map all observations in the support of $\mathcal{Q}_{M_2}(s_i, e_1)$ and $\mathcal{Q}_{M_2}(s_{(i+1)\%N}, e_2)$ to two *distinct* codes, each with marginal probability at least $1 - 2\epsilon$. Therefore, this $\phi$ will map at least $0.5 - \epsilon$ of the observations sampled from $\mathcal{Q}_{M_3}(s_i, e)$ for $e \sim \pi_{\mathcal{E}}$ to one latent state, and at least another $0.5 - \epsilon$ of the observations sampled from this distribution to a *different* latent state. Then under *any* choice of perturbation $\sigma$, the accuracy of $\phi(\cdot)$ on $M_3$ can be at most $0.5 + \epsilon$.

up to timestep $t$, and the algorithm $A$, *seeing exactly the same distribution of observations as if the MDP was $M_1$*, will halt before timestep $t$.

Considering the 50% probability that the exogenous state starts at $e_1$, this gives at least a 40% chance that $A$ applied to $M_2(\gamma)$ or $M_3(\gamma)$ never encounter the exogenous state $e_2$. In this case, the distribution of encoders output by $A$ should be the same for the two Ex-BMDPs: let this distribution be $\mathcal{G}$. (That is, $\mathcal{G}$ is the conditional distribution of encoders output by $A$ when applied to $M_2(\gamma)$ or $M_3(\gamma)$, given that $A$ never encounters $e_2$.) However, because $A$ fails on $M_2(\gamma)$ with probability at most $\delta$, we can conclude that at least $(1 - 0.4^{-1}\delta)$ of the encoders from the distribution $\mathcal{G}$ represent *successes* of $A$ on $M_2(\gamma)$.

Because $A$ on $M_3(\gamma)$ also draws from $\mathcal{G}$ with probability at least 0.4, we can conclude that at least $0.4 - \delta$ of the time, $A$ on $M_3(\gamma)$ produces an encoder that is highly accurate on $M_2(\gamma)$. As discussed above, any encoder that is highly-accurate on $M_2(\gamma)$ cannot be highly accurate on $M_3(\gamma)$, so $A$ must fail on $M_3$ with substantial probability.

Therefore, we can conclude that for any algorithm $A$ that has no "hint" about the exogenous dynamics (such as knowing the mixing time), there exists some $\gamma$ such that $A$ cannot possibly succeed with high probability on both $M_2(\gamma)$ and $M_3(\gamma)$.

To show point (2), simply note that in this construction, in order to ensure that $e_2$ is observed with probability $(1 - \delta)$, an algorithm must observe at least the first $\lceil \log(2\delta)/\log(1 - \gamma) \rceil$ timesteps. Meanwhile, the mixing time of the two-state Markov chain exogenous state is given by $\lceil -1/\log_2(1 - 2\gamma) \rceil$. For a fixed $\delta$ and as $\gamma$ approaches 0, the ratio between these quantities approaches a constant: therefore, the number of steps required to ensure that $e_2$ is observed with high probability is linear in the mixing time.

## F  DOUBLE-PRIME LOOP EXPERIMENTS

In this section, we present additional experiments which demonstrate that, on some types on Ex-BMDPs, the STEEL algorithm can empirically outperform the algorithms proposed in Lamb et al. (2023) and Levine et al. (2024), which do not have sample-complexity guarantees. We show that there are certain structures of Ex-BMDP latent dynamics which are difficult for these prior methods to learn efficiently, but on which STEEL performs well. Specifically, we look at a family of "double-prime loop" tabular Ex-BMDPs which are discussed by Levine et al. (2024). First, though, we describe the algorithms proposed by Lamb et al. (2023) and Levine et al. (2024), and motivate why certain dynamics structures such as "double-prime loops" present a challenge to these methods.

### F.1  BACKGROUND

#### F.1.1  AC-STATE AND ACDF ALGORITHMS

Lamb et al. (2023) and Levine et al. (2024) both propose algorithms to learn endogenous state encoders in the Ex-BMDP framework.

Lamb et al. (2023) first proposed the AC-State algorithm, which aims to learn an encoder $\phi : \mathcal{X} \to \mathbb{N}$, such that, under some one-to-one permutation $\sigma$, $\sigma(\phi(x)) = \phi^*(x)$. To accomplish this task, AC-State optimizes $\phi$ using a *multistep inverse dynamics* objective.

Specifically, in the theoretical treatment, AC-State tries to find the encoder $\phi$ which optimizes the following objective:

$$
\begin{aligned}
\mathcal{L}(\phi) &:= \min_{g \in \mathbb{N} \times \mathbb{N} \times \mathbb{N} \to \mathcal{P}(\mathcal{A})} \; \mathbb{E}_{k \sim \{1,\dots,K\}} \; \mathbb{E}_{(x_t, a_t, x_{t+k}) \sim \mathcal{D}} - \log \left( g\big(\phi(x_t), \phi(x_{t+k}), k\big)\big[a_t\big] \right) \\
\phi_{\text{AC-State}} &:= \argmin_{\mathcal{L}(\phi) = \min_{\phi'} \mathcal{L}(\phi')} |\text{Range}(\phi)|
\end{aligned}
\tag{140}
$$

In other words, the loss $\mathcal{L}(\phi)$ is minimized on tuples $(x_t, a_t, x_{t+k})$ sampled from a trajectory, consisting of an observation $x_t$, the following action $a_t$, and the observation $k$ *steps in the future*, $x_{t+k}$.

The encoder $\phi(x)$ is trained to retain any information about the observations $\phi(x_t)$ and $\phi(x_{t+k})$ that is useful for predicting $a_t$. Specifically, an inverse-dynamics model $g$ is simultaneously trained with

$\phi$ to predict $a_t$ given $\phi(x_t)$, $\phi(x_{t+k})$, and $k$. The loss $\mathcal{L}(\phi)$ is then taken as the minimum loss for $\phi$ over all such functions $g$.

However, note that, based on this loss function alone, the identity encoder $\phi(x) := x$ achieves the minimum possible value of $\mathcal{L}(\phi)$.[9] Therefore, in order to filter exogenous information, the final encoder returned by AC-State is the *minimal range* encoder that minimizes the loss function.

Lamb et al. (2023) claims that, if the data $\mathcal{D}$ is collected by a policy $\pi(x)$ that *only* depends on $x$ through $\phi^*(x)$ (that is, the policy ignores exogenous noise); *and* the maximum segment length $K$ is at least the endogenous dynamics diameter $D$; *and* the endogenous dynamics are deterministic, then the final encoder $\phi_{\text{AC-State}}$ returned will correctly encode the endogenous latent state (in the limit of infinite data; and with sufficient function approximation).

When applied in practice, $\phi$ and $g$ are learned neural networks, and a vector quantization bottleneck is used to restrict the range of $\phi$ (so the output of $\phi$ is a quantized vector, rather than an integer.)

Levine et al. (2024) subsequently show that, in some cases, the AC-State objective in Equation 140 is not sufficient to learn a correct latent state encoder of an Ex-BMDP, even with unbounded amounts of data. Levine et al. (2024) identify specific flaws in the proofs of Lamb et al. (2023), and propose a modified loss function to address these flaws. In particular,

1. The maximum segment length $K$ required to correctly learn the endogenous encoder must in some cases be larger than $D$; a corrected upper bound on the necessary segment length of $2D^2 + D$ is given.

2. If the endogenous dynamics are *periodic*, then the multistep inverse objective can be insufficient, for any choice of $K$. Levine et al. (2024) then propose to add an auxiliary loss function to the loss in Equation 140, namely a *latent forward dynamics loss*:

$$\mathcal{L}_{\text{forward}}(\phi) := \min_{h \in \mathbb{N} \times \mathcal{A} \to \mathcal{P}(\mathbb{N})} \mathbb{E}_{(x_t, a_t, x_{t+1}) \sim \mathcal{D}} - \log\left(h\big(\phi(x_t), a_t\big)\big[\phi(x_{t+1})\big]\right). \quad (141)$$

This loss term enforces that the transitions in the learned endogenous latent state space are deterministic; Levine et al. (2024) prove that enforcing this constraint is sufficient, when combined with the multistep inverse loss, to ensure that a correct endogenous representations are learned even when the endogenous latent dynamics are periodic. Levine et al. (2024) name the AC-State algorithm with this modified loss function ACDF.

Note that because $D$ is in general unknown *a priori*, $K$ must be treated as a hyperparameter of the algorithm, so point (1) above is primarily of theoretical interest, but may aid in setting this hyperparameter if there is some prior knowledge of $D$.

So far, we have only described the learning objectives of the two proposed algorithms: now, we address how these algorithms collect the trajectory from which the tuples $(x_t, a_t, x_{t+k})$ are sampled. Recall that a condition on the correctness of AC-State (and ACDF) is that the data-collection policy does not depend on the noise in the observation, but only on the endogenous latent state. This condition naturally raises the question of *how exactly* an algorithm intended to discover $\phi^*$ can take actions that depend only on $\phi^*(x)$, without already knowing $\phi^*$ in the first place.

Lamb et al. (2023) performs experiments using two data-collection policies: (1). a *uniformly random* policy, which meets the condition simply by ignoring the observation $x$ entirely; and (2) a goal-seeking policy, which aims to maximize latent state coverage. For the latter policy, the data is collected simultaneously with training $\phi$, and the partially-trained encoder $\phi_t$ is used to (imperfectly) filter out exogenous noise and plan in (an approximation of) the endogenous latent space. Lamb et al. (2023) explicitly acknowledge that this bootstrapping approach *breaks* the condition that the action $a_t$ depends only on the true endogenous state $\phi^*(x_t)$, which is necessary in their proof of the correctness of their algorithm. Despite this, they observe that the approach seems to work well empirically. Meanwhile, Levine et al. (2024) *only* use uniformly random actions in their experiments.

For now, we will assume that data is collected under a uniformly-random policy. We will return to discussing the latent-state-coverage maximizing approach proposed by Lamb et al. (2023) later, in Section F.3.

---

[9]Assuming $\mathcal{X}$ is countable.

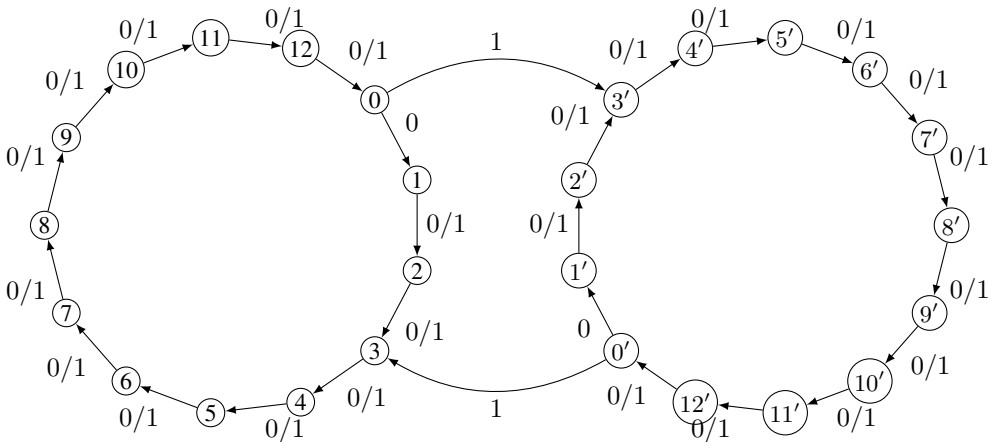

Figure 5: Latent dynamics of DoublePrime(11, 13).

To specifically demonstrate that $K \approx \mathcal{O}(D^2)$ can be necessary to learn a correct encoder, even when a forward dynamics loss is *also* being used, Levine et al. (2024) give a concrete example. This example is the family of tabular "double-prime loop" Ex-BMDPs. For any two primes $p, q$, with $p < q$, let the Ex-BMDP DoublePrime$(p, q)$ be defined as follows:

- $\mathcal{S} = \{0, 1, ..., (q-1), 0', 1', ..., (q-1)'\}$; $\mathcal{A} = \{0, 1\}$; $\mathcal{E} = \{e_0\}$.
- $\mathcal{X} = \mathcal{S}$; $\mathcal{Q}(s, e_0) = s$; $\mathcal{T}_e(e_0) = e_0$.
- The endogenous latent state transition function $T$ is defined as:

$$T(s, a) = \begin{cases} (s+1)\% \, q & \text{if } s \in \{1, ..., q-1\} \text{ or } (s = 0 \text{ and } a = 0) \\ ((s+1)\% \, q)' & \text{if } s \in \{1', ..., (q-1)'\} \text{ or } (s = 0' \text{ and } a = 0) \\ (q-p+1)' & \text{if } (s = 0 \text{ and } a = 1) \\ (q-p+1) & \text{if } (s = 0' \text{ and } a = 1). \end{cases} \quad (142)$$

In other words, DoublePrime$(p, q)$ is a noise-free Ex-BMDP, where all of the dynamics are deterministic, and the latent state $s$ is directly observed as $x$. The dynamics consist of two connected loops, each of $q$ states. The actions are generally ignored and the agent continues to progress through a loop, except for in states $0$ and $0'$, where taking the action $1$ transports the agent to the other loop, at position $q - p + 1$. As an example, the dynamics of DoublePrime$(11, 13)$ are shown in Figure 5.

There is also a related family of "single-prime loop" Ex-BMDPs. Let SinglePrime$(p, q)$ be defined as (borrowing notation from Levine et al. (2024)):

- $\mathcal{S} = \{0^*, 1^*, ..., (q-1)^*\}$; $\mathcal{A} = \{0, 1\}$; $\mathcal{E} = \{e_0\}$; $\mathcal{T}_e(e_0) = e_0$.
- $\mathcal{X} = \{0, 1, ..., (q-1), 0', 1', ..., (q-1)'\}$.
- $\mathcal{Q}(s^*, e_0) = \begin{cases} s & \text{with probability } 1/2 \\ s' & \text{with probability } 1/2. \end{cases}$
- The endogenous latent state transition function $T$ is defined as:

$$T(s^*, a) = \begin{cases} ((s+1)\% \, q)^* & \text{if } s \in \{1^*, ..., (q-1)^*\} \text{ or } (s = 0^* \text{ and } a = 0) \\ (q-p+1)^* & \text{if } (s = 0^* \text{ and } a = 1). \end{cases} \quad (143)$$

In other words, SinglePrime$(p, q)$ has the same observation space $\mathcal{X}$ and action space $\mathcal{A}$ as DoublePrime$(p, q)$, but fewer controllable latent states. In SinglePrime$(p, q)$, when the agent is at latent state $s_t = i^*$, either the observation $x_t = i$ or the observation $x_t = i'$ is emitted, each with probability 0.5. The agent progresses through the loop of latent states regardless of actions, except

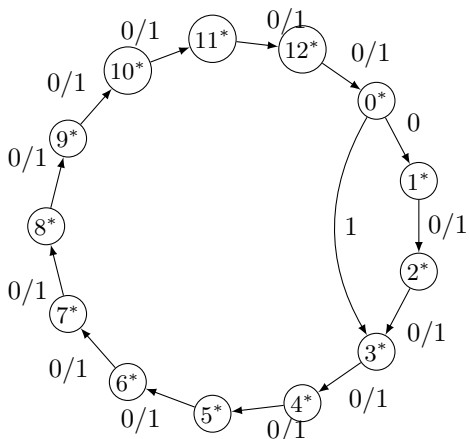

Figure 6: Latent dynamics of SinglePrime$(11, 13)$.

in latent state $0^*$, where taking action 1 transitions the latent state to $(q - p + 1)^*$. (See Figure 6 for an example.) Note that if we ignore the distinction between the observations $i$ and $i'$, then DoublePrime$(p, q)$ and SinglePrime$(p, q)$ appear to have identical observed dynamics.

Let $\phi_{DP}(x) := x$ be the optimal encoder for DoublePrime$(p, q)$, and

$$\phi_{SP}(x) := i^* \text{ if } x = i \text{ or } x = i' \tag{144}$$

be the optimal encoder for SinglePrime$(p, q)$.

Levine et al. (2024) show that, on the Ex-BMDP DoublePrime$(p, q)$ under a uniformly random behavioral policy, *in the limit of infinite training data*, the encoders $\phi_{DP}$ and $\phi_{SP}$ have *exactly the same loss* under the loss function in Equation 140, with $K \leq (q - 1)p$. Note that if $p$ and $q$ are close, then $(q - 1)p \approx D^2/4$. Because Equation 140 prefers the smallest-range encoder among encoders with identical losses, this means that AC-State will incorrectly return $\phi_{SP}$ (which only has $q$ distinct outputs, rather than $2q$.). Futhermore, the forward-dynamics loss suggested by Levine et al. (2024) does not help in this case: it will be zero for both encoders under data generated by DoublePrime$(p, q)$.

It is important to note that SinglePrime$(p, q)$ and DoublePrime$(p, q)$ have *truly distinct* controllable latent dynamics: in DoublePrime$(p, q)$, given sufficient lead-time, the agent can (eventually) control whether it is in state $1'$ or $1$ at a future time step, while an agent in SinglePrime$(p, q)$ can *never* control this. Therefore, AC-State is making an *error* if it returns $\phi_{SP}$ rather than $\phi_{DP}$.

Levine et al. (2024) only considers the case of unlimited data, and only introduces the double-prime loop Ex-BMDPs to make the point that the hyperparameter $K$ must be set very high in some cases. Here, however, we will *also* show empirically that, even when using a carefully-tuned and very large value of $K$, AC-State and ACDF can take many samples to correctly learn these dynamics.

### F.1.3 TABULAR EXPERIMENTS IN LEVINE ET AL. (2024)

Levine et al. (2024) empirically compare AC-State (Lamb et al., 2023) to their proposed ACDF method, and empirically explore the effect of the hyperparameter $K$. To do so from a purely statistical perspective (i.e., controlling for differences in optimization), in one set of experiments, Levine et al. (2024) perform tests on small tabular Ex-BMDPs, including SinglePrime$(3, 5)$ and DoublePrime$(3, 5)$.[10] For these experiments, the final learned $\phi_{\text{AC-State}}$ (or $\phi_{\text{ACDF}}$) is found by exhaustively computing the losses in Equations 140 and 141 on *all possible* encoders $\phi \in \mathcal{X} \to \{0, ..., |\mathcal{X}| - 1\}$ (up to a relabeling perturbation of the output). For each encoder $\phi$, the multistep inverse model $g$ (and, for ACDF, the forward model $h$) is estimated as a tabular function based on

---

[10]The experiment in Levine et al. (2024) on SinglePrime$(3, 5)$ is on a slightly different variant of the MDP from what is presented here: the controllable latent dynamics are the same, but the noise is time-correlated.

the collected data:

$$g(s, s', k)[a] := \frac{\text{\# of instances of } (x_t, a_t, x_{t+k}) \text{ such that } \phi(x_t){=}s; a_t{=}a; \text{ and } \phi(x_{t+k}){=}s'}{\text{\# of instances of } (x_t, a_t, x_{t+k}) \text{ such that } \phi(x_t){=}s \text{ and } \phi(x_{t+k}){=}s'}. \quad (145)$$

However, due to overfitting, this tabular definition of $g$, if used naively, would always lead to the identity encoder $\phi(x) := x$ having the lowest empirical loss, regardless of the true latent encoder. Therefore, Levine et al. (2024) collect *two* trajectories for each experiment, and uses one to *fit* $g$ and $h$ for each possible encoder $\phi$, and the other to *evaluate* the loss on each $\phi$ in order to determine the final output encoder $\phi_{\text{AC-State}}$ (or $\phi_{\text{ACDF}}$). We will call these sets of tuples the *fitting set* and *optimization set*, repectively.

Because the number of possible encoders grows very quickly with $|\mathcal{X}|$, Levine et al. (2024) limit these experiments to very small tabular Ex-BMDPs, with $|\mathcal{X}| \leq 10$.

With this background out of the way, we describe *our* experiments:

## F.2 EXPERIMENTS

### F.2.1 SETUP

Given that Levine et al. (2024) describe how learning the correct latent state encoder for DoublePrime$(p, q)$ using a multistep-inverse loss requires taking into account specific *long-duration* dependencies between states, we hypothesized that learning such an encoder using multistep-inverse methods may be considerably less sample-efficient that doing so using STEEL, particularly for large $p$ and $q$. We therefore adapted the tabular Ex-BMDP experimental setup from the released code of Levine et al. (2024) and tested the algorithms head-to-head.

However, for large $p$ and $q$, the process of exhaustively searching all possible encoders (that is, all partitions of $2q$ elements) quickly becomes intractable. Therefore, we restricted the hypothesis space of encoders to *only* include the *two* hypotheses $\phi_{SP}$ and $\phi_{DP}$. Note that this modification makes the learning task strictly "easier." For a fair comparison, we also restricted the hypothesis class for STEEL to the minimum-possible set of "one-versus-rest" classifiers that would ensure realizability for both DoublePrime$(p, q)$ and SinglePrime$(p, q)$. In particular, this is $3q$ classifiers $f \in \mathcal{F}$: for each $i \in \{0, ..., 1 - q\}$, $\mathcal{F}$ includes a hypothesis which distinguishes $i$ from all other observations, a hypothesis which distinguishes $i'$ from all other observations, and a hypothesis which distinguishes $i$ *or* $i'$ from all other observations.

We make the following further modifications to the experimental protocol from Levine et al. (2024):

- When assessing the "minimum-range" minimal loss encoder as in Equation 140, Levine et al. (2024) include an empirical "fudge factor": their protocol returns the minimum-range encoder that achieves a loss within 0.1% of the true minimum loss over all possible encoders. We found that *even without this fudge factor*, the multistep-inverse methods were still heavily biased towards returning $\phi_{SR}$, when applied either to SinglePrime$(p, q)$; *or to* DoublePrime$(p, q)$ with too-small $K$ or too-few samples. This observation has a simple statistical explanation: if, in the "infinite sample" limit, $\Pr(a_t|s_t = i, s_{t+1} = j) = \Pr(a_t|s_t = i', s_{t+1} = j')$, then fitting $g(i, j, k)$ to both samples of $(x_t = i, a_t, x_{t+k} = j)$ *and* samples of $(x_t = i', a_t, x_{t+k} = j')$ from the "fitting set" will lead to a higher-quality multistep inverse model $g$, and therefore lower loss on the "optimization set", compared to using only the (smaller number of) samples of $(x_t = i, a_t, x_{t+k} = j)$ alone. We therefore remove the fudge factor entirely, and in fact return $\phi_{GR}$ in the case of exact numerical ties.

- Because the forward-model loss in Equation 141 is zero for both $\phi_{SR}$ and $\phi_{GR}$ for *any* dataset collected from DoublePrime$(p, q)$, we do not use it in our experiments: that is, we regard AC-State and ACDF as equivalent in this setting, and refer to AC-State alone from here onward. (Technically, the forward model loss *would* help identify the correct encoder $\phi_{SR}$ in data generated from SinglePrime$(p, q)$. However, as mentioned above, $\phi_{SR}$ is *already* returned essentially "by default" by AC-State, so this loss term turns out to be unnecessary.)

- To better match the spirit of "learning an Ex-BMDP from a single trajectory", we collect the "fitting" and "optimization" sets from one single trajectory, one after the other, with each corresponding to half of the trajectory: we regard the total sample complexfity as the

length of the *entire* trajectory. Within of these two sub-trajectories, we collect all available tuples $(x_t, a_t, x_{t+k})$ for all $k \leq K$. There are therefore slightly fewer tuples with $k = K$ than with $k = 1$: we weight all *tuples* equally in the loss (rather than weighting all values of $k$ equally; Lamb et al. (2023) is unclear about the "correct" behavior here, and it is not theoretically important).

- Based on Equation 145, $g(s, s', k)[a]$ can be zero, which, by Equation 140, can lead to an infinite loss on a sample in the "optimization" set. In order to avoid this, Levine et al. (2024) set $g(s, s', k)[a]$ to an arbitrary floor value of $10^{-7}$. As a more principled and scale-invariant solution, we use standard Laplace ("add-one") smoothing when fitting $g$.

We empirically compare the sample-efficiency of STEEL to AC-State on these problems with $(p, q) = (3, 5), (5, 7), (11, 13), (17, 19), (29, 31),$ and $(41, 43)$.

In our comparison, we run each algorithm under the "best" choice of hyperparameters. For AC-State, the conduct a large hyperparameter search to find the optimal $K$, while for STEEL, we simply choose a single set of hyperparameters that will succeed on both DoublePrime$(p, q)$ and SinglePrime$(p, q)$ based on our prior knowledge of the problems. Specifically:

- For AC-State, for each tested value of $(p, q)$, we first collect 20 validation trajectories of DoublePrime$(p, q)$ (under a uniformly random policy), for increasing dataset sizes starting at $T = 100$ steps. At each tested dataset size $T$, we compute the optimal $\phi_{\text{AC-State}}$ for each of the 20 trajectories, with each possible value of $K$ from $K = 1$ to $K = 50,000$. We first repeatedly increase the dataset size $T$ by factors of 10. Once we first identify a $T = 10^m$ such that, for at least some $K$, AC-State correctly returns $\phi_{\text{AC-State}} = \phi_{DP}$ for all 20 trajectories, we then test with $T = \{2 \cdot 10^{m-1}, 3 \cdot 10^{m-1}, ..., 9 \cdot 10^{m-1}\}$. At the earliest of *these* timesteps at which, for some $K$, a correct encoder is learned for all 20 trajectories, we record the median such $K$ as the "tuned" value $K_{\text{opt}}$ of this hyperparameter. (See Figure 7 for the results of the hyperparameter search.) At this point, we perform the actual test: we evaluate the success rate of AC-State on DoublePrime$(p, q)$, using only $K = K_{\text{opt}}$, for values of $T$ starting at $T = 10^{m-1}$ and increasing by increments of $10^{m-1}$, on 20 new trajectories for each tested dataset size. We stop when this test-time accuracy reaches 20/20 on trajectories from DoublePrime$(p, q)$, at some $T_{\text{max}}$. Finally, we verify that AC-State can *also* consistently, correctly learn $\phi_{SP}$ on data from SinglePrime$(p, q)$ with $K = K_{\text{opt}}$ and a trajectory length of $T_{\text{max}}$, for 20 trajectories. (This always held in practice.[11])

- For STEEL, we set $N = \hat{D} = 2q$ (to match the maximum number of states in DoublePrime$(p, q)$ or SinglePrime$(p, q)$); $\hat{t}_{mix} = 1$ (because neither DoublePrime$(p, q)$ nor SinglePrime$(p, q)$ have time-correlated noise); $\epsilon = 0.49$ (because, due to the simple emission distributions $\mathcal{Q}$ of SinglePrime and DoublePrime, a $51\%$ encoder accuracy on each latent state implies a $100\%$ encoder accuracy); and $\delta = 0.05$. We test for 20 trials each of SinglePrime$(p, q)$ and DoublePrime$(p, q)$. **All of these tests returned the correct encoders**, so the only metric we needed to consider was the number of environment steps taken for each run.

### F.2.2 RESULTS

Our top-line results are reported in Table 3.Our first reported statistic is the "Max Steps": the number of environment steps at which all 20 out of 20 tested trajectories for each of DoublePrime$(p, q)$ and SinglePrime$(p, q)$ returned the correct encoder. (For STEEL, this is simply the maximum steps taken over all 40 trajectories; for AC-State, this is the $T_{\text{max}}$ discovered in the test-time search described in the previous section.) We see that, while for small instances ($|S| \leq 14$) of DoublePrime$(p, q)$, AC-State is more sample-efficient than STEEL, STEEL scales efficiently to larger instances of the problem.

---

[11]We avoided more extensive testing with SinglePrime$(p, q)$, both because, as discussed above, AC-State tends to "default" to returning the encoder $\phi_{SP}$, which is correct for SinglePrime$(p, q)$; and also because the deterministic dynamics of DoublePrime$(p, q)$ allowed us to take some computational "shortcuts" when evaluating the loss function in Equation 140 on data from DoublePrime$(p, q)$ for very large $K$, which were not possible for SinglePrime$(p, q)$. This made more exhaustive experimentation on DoublePrime$(p, q)$ significantly more tractable than it would be on SinglePrime$(p, q)$.

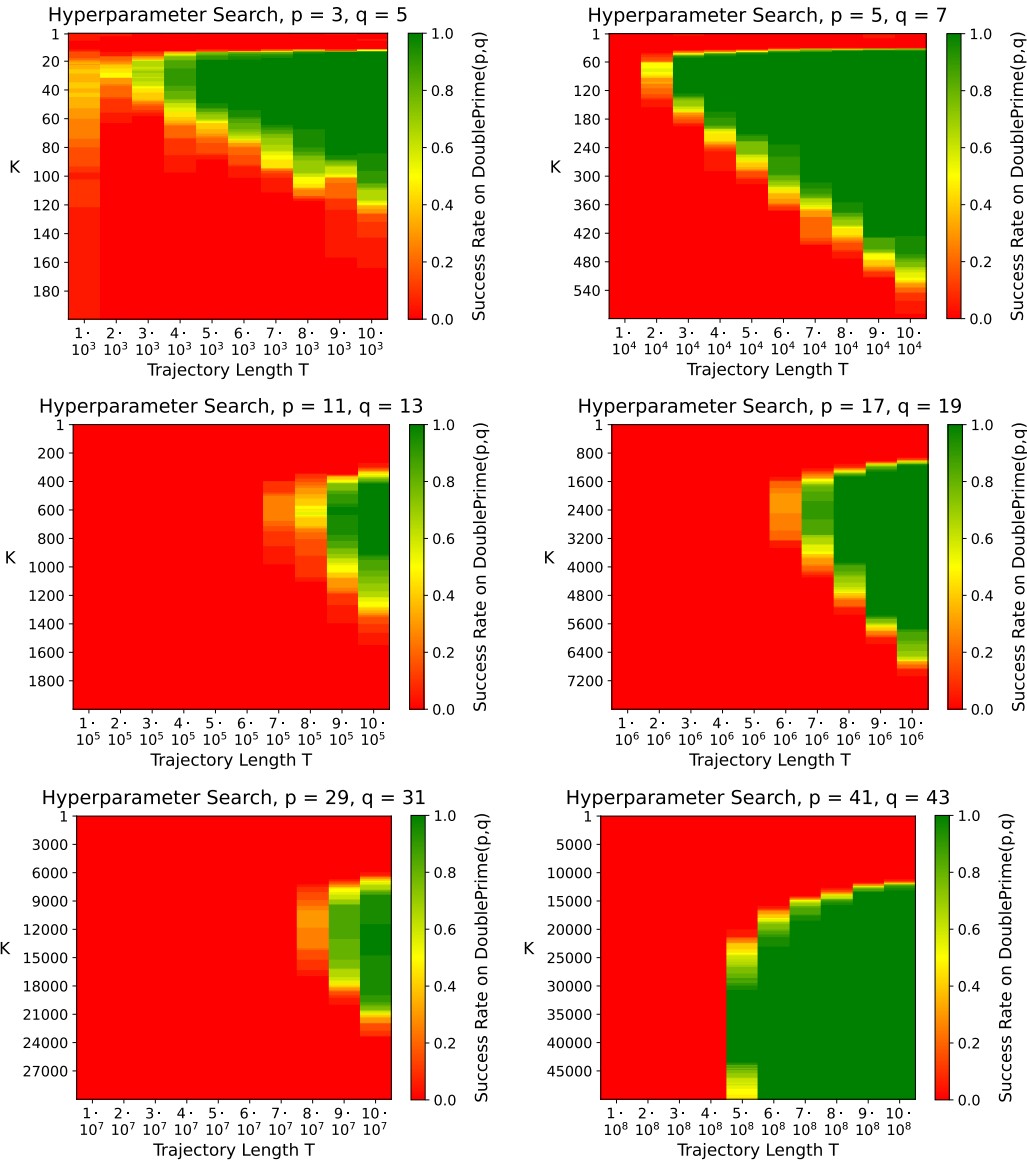

Figure 7: Results from the hyperparameter search for the maximum-step-count parameter $K$ in Equation 140. The objective is to find a $K$ which leads to a high success rate in learning the correct latent state encoder for DoublePrime$(p, q)$, for the lowest possible number of environment steps.

| $(p, q)$ | (3,5) | (5,7) | (11,13) | (17,19) | (29,31) | (41,43) |
|---|---|---|---|---|---|---|
| Max Steps for STEEL | $5.97 \cdot 10^4$ | $1.49 \cdot 10^5$ | $9.07 \cdot 10^5$ | $\mathbf{2.60 \cdot 10^6}$ | $\mathbf{1.10 \cdot 10^7}$ | $\mathbf{2.83 \cdot 10^7}$ |
| Max Steps for AC-State | $\mathbf{4 \cdot 10^3}$ | $\mathbf{4 \cdot 10^4}$ | $1 \cdot 10^6$ | $8 \cdot 10^6$ | $1 \cdot 10^8$ | $5 \cdot 10^8$ |
| Max Steps: (AC-State / STEEL) | $\approx 0.07$ | $\approx 0.3$ | $\approx 1$ | $\approx 3$ | $\approx 9$ | $\approx 18$ |
| Median Steps (STEEL, DoublePrime) | $2.50 \cdot 10^4$ | $7.48 \cdot 10^4$ | $5.23 \cdot 10^5$ | $\mathbf{1.08 \cdot 10^6}$ | $\mathbf{1.74 \cdot 10^6}$ | $\mathbf{3.29 \cdot 10^6}$ |
| Median Steps (AC-State, DoublePrime) | $\mathbf{3 \cdot 10^3}$ | $\mathbf{3 \cdot 10^4}$ | $8 \cdot 10^5$ | $7 \cdot 10^6$ | $9 \cdot 10^7$ | $5 \cdot 10^8$ |
| Median Steps: (AC-State / STEEL) | $\approx 0.1$ | $\approx 0.4$ | $\approx 1.5$ | $\approx 6$ | $\approx 50$ | $\approx 150$ |

Table 3: Comparison of the empirical sample complexity of STEEL and AC-State on prime-loop MDPs. See text of Section F.2.2.

Further, we noticed an interesting trend in the data. AC-State tended to transition very abruptly from learning the wrong encoder on all tests of DoublePrime to learning the correct encoder on all tests as $T$ increased (as can be observed in the hyperparameter search in Figure 7). By contrast, the distribution of steps taken for STEEL on DoublePrime$(p, q)$ was highly skewed: most trials took significantly fewer than the maximum observed number of steps. We therefore also compare the *median* number of steps taken to learn a correct encoder for DoublePrime$(p, q)$. For STEEL, this is computed as simply the median length of the 20 DoublePrime$(p, q)$ trajectories tested. For AC-State, it is the first timestep $T$ in the test-time search where AC-State with $K = K_{\text{opt}}$ learns a correct encoder for at least 10 of the 20 trajectories. Here, we find that, especially for large $(p, q)$, STEEL's advantage over AC-State becomes even more pronounced.

## F.3 EXPLORATION POLICIES WITH AC-STATE

So far, we have only compared STEEL to AC-State, where AC-State uses a uniformly random exploration policy. This may seem unfair: STEEL takes decidedly nonrandom actions, and Lamb et al. (2023) proposes an active exploration method for AC-State (albeit, as mentioned in Section F.1.1 above, a method without strict theoretical grounding).

However, there is reason to believe that exploration policies similar to the one proposed by Lamb et al. (2023) would be *unlikely* to help correctly identify the dynamics of DoublePrime$(p, q)$.

To start with, the stated goal of the the exploration policy in Lamb et al. (2023) is to "achieve high coverage of the control-endogenous state space." A natural question to ask is: how poorly do uniformly random policies perform at achieving this goal? To quantify this, we examined the state coverage for a single $10^6$-step trajectory on DoublePrime$(41, 43)$, and computed the ratio of the state visitation of the *most-visited state* to the state visitation of the *least-visited state*. Surprisingly, this was $\approx 2$. In other words, DoublePrime is **not** a hard exploration problem, and so techniques designed to improve state coverage would seem to be unlikely to provide much of a benefit over a uniformly random policy.

Still, we will now examine the specific exploration technique proposed by Lamb et al. (2023). At a high level, the technique proceeds as follows:

- Throughout data collection, the agent maintains an approximate version of the encoder $\phi'$ obtained by optimizing the AC-State loss (Equation 140), and an approximate transition function $T'$, obtained through counting. The algorithm also maintains state visitation counts for each of the learned latent states encoded by $\phi'$.

- At the start of each round of exploration (time $t$), the agent selects a goal learned-latent state $s_g$, with probability inversely proportional to the visitation count. The agent then takes a single random action $a_t$, and then, starting at time $t + 1$, plans shortest-path to $s_g$, and computes the number of steps $k'$ it will take to get to $s_g$. Then, the agent proceeds to navigate to $s_g$ in a closed-loop manner, using both $\phi'$ and $T'$. After $k'$ steps, *regardless of whether $s_g$ is reached*, a new goal is set, and the process starts over.

- The encoder $\phi$ is *only* trained on the tuples $(x_t, a_t, x_{t+k'+1})$ which begin with a random action and end with a goal state.

We performed experiments of a version of this algorithm adapted for our tabular setting, on the DoublePrime environments. Here, we explain the modifications we made to this exploration algorithm from Lamb et al. (2023) for our tests. Examining the original algorithm, we immediately

| $(p,q)$ | (3,5) | (5,7) | (11,13) | (17,19) | (29,31) |
|---|---|---|---|---|---|
| Max Steps (AC-State, DoublePrime, Uniform Policy) | $4 \cdot 10^3$ | $4 \cdot 10^4$ | $1 \cdot 10^6$ | $8 \cdot 10^6$ | $1 \cdot 10^8$ |
| Max Steps (AC-State, DoublePrime, Exploration) | $8 \cdot 10^3$ | $6 \cdot 10^4$ | $2 \cdot 10^6$ | $2 \cdot 10^7$ | $2 \cdot 10^8$ |
| Median Steps (AC-State, DoublePrime, Uniform Policy) | $3 \cdot 10^3$ | $3 \cdot 10^4$ | $8 \cdot 10^5$ | $7 \cdot 10^6$ | $9 \cdot 10^7$ |
| Median Steps (AC-State, DoublePrime, Exploration) | $6 \cdot 10^3$ | $4 \cdot 10^4$ | $2 \cdot 10^6$ | $2 \cdot 10^7$ | $2 \cdot 10^8$ |

Table 4: Comparison of the empirical sample complexity of AC-State on the DoublePrime$(p,q)$ environments, with and without active exploration. (Note that we did not perform a final comparison for $p = 41, q = 43$, because the hyperparameter search for the active exploration policy (see Figure 8) with a maximum $K$ of 50000 did not rule out a $K_{\text{opt}} > 50000$, so we could not fairly tune this hyperparameter for the active exploration policy. We were unable to perform a hyperparameter search with $K > 50000$ due to technical limitations.)

notice an issue: the encoder is only trained with segments of length $k = k' + 1$, which (assuming $\phi'$ and $T'$ are anywhere close to accurate) will only scale linearly with $D$. However, properly learning the encoder for the DoublePrime environments *requires* examining longer segments of trajectories than the diameter of the dynamics graph. Therefore, in order adapt this exploration method to the setting we consider, we eliminate this feature of the algorithm and train $\phi$ on the entire collected trajectory, as in Equation 140.

Furthermore, we give the exploration method what should be a large, unnatural advantage, by planning, choosing goals, and assessing state visitation on the *true ground truth dynamics $T$, with the ground-truth state encoder $\phi$*. This represents what should be the "best possible" case for the algorithm. It also enables us to test in this setting without having to optimize $\phi'$ continuously during data collection, and to assess many values of the hyperparameter $K$ (which would impact the intermediate encoder $\phi'$) simultaneously on a single collected trajectory.

In our implementation, we reset the state visitation counts between collecting the "fitting" and "optimization" portions of the trajectory, in order to ensure that they are distributed in the same way. We also take uniformly random actions in cases where both actions produce a path of the same length to the goal. Otherwise, we simply take the shortest available path to the goal state, and, when it is reached, pick a new goal and take one random action.

Hyperparameter search results are shown in Figure 8, and top-line results are shown in Table 4. We see in Table 4 that AC-State with the active exploration method seems to slightly *underperform* random exploration on the DoublePrime$(p,q)$ environments, requiring slightly *more* samples to find the correct encoder, for all values of $(p,q)$ that were tested.[12] While it is not immediately obvious why we see this gap, these results support our initial hypothesis that, because the DoublePrime environments are well-explored by uniformly random policies, techniques designed to improve latent state coverage are not helpful in improving AC-State's performance on these environments.

### F.4 DISCUSSION

It is important to note that the DoublePrime environments are *not* hard-exploration environments, nor do they have *any* time-correlated noise, nor do they have rich observations. In fact, adding time-correlated noise or rich observations would likely only scale the sample-complexity of STEEL on these environments only modestly (linearly in $\hat{t}_{mix}$ and $\log(|\mathcal{F}|)$). Rather, STEEL outperforms AC-State on large instances of these environments because the environments' transition dynamics are arranged in a way that is in a sense "adversarial" to multistep-inverse methods. By contrast, because STEEL has well-understood sample-complexity that applies for *any* transition graph structure (as long as the reachability assumption holds), we do not expect that any such "adversarial" environments to exist for STEEL.

---

[12]However, note that because we only tested trajectory lengths $T$ at particular values as described in Section F.2.1, some of these performance differences may appear exaggerated: for example, the difference between a trajectory length of $1 \cdot 10^6$ and $2 \cdot 10^6$ must be interpreted with the understanding that no trajectory lengths *between* $1 \cdot 10^6$ and $2 \cdot 10^6$ were tested.

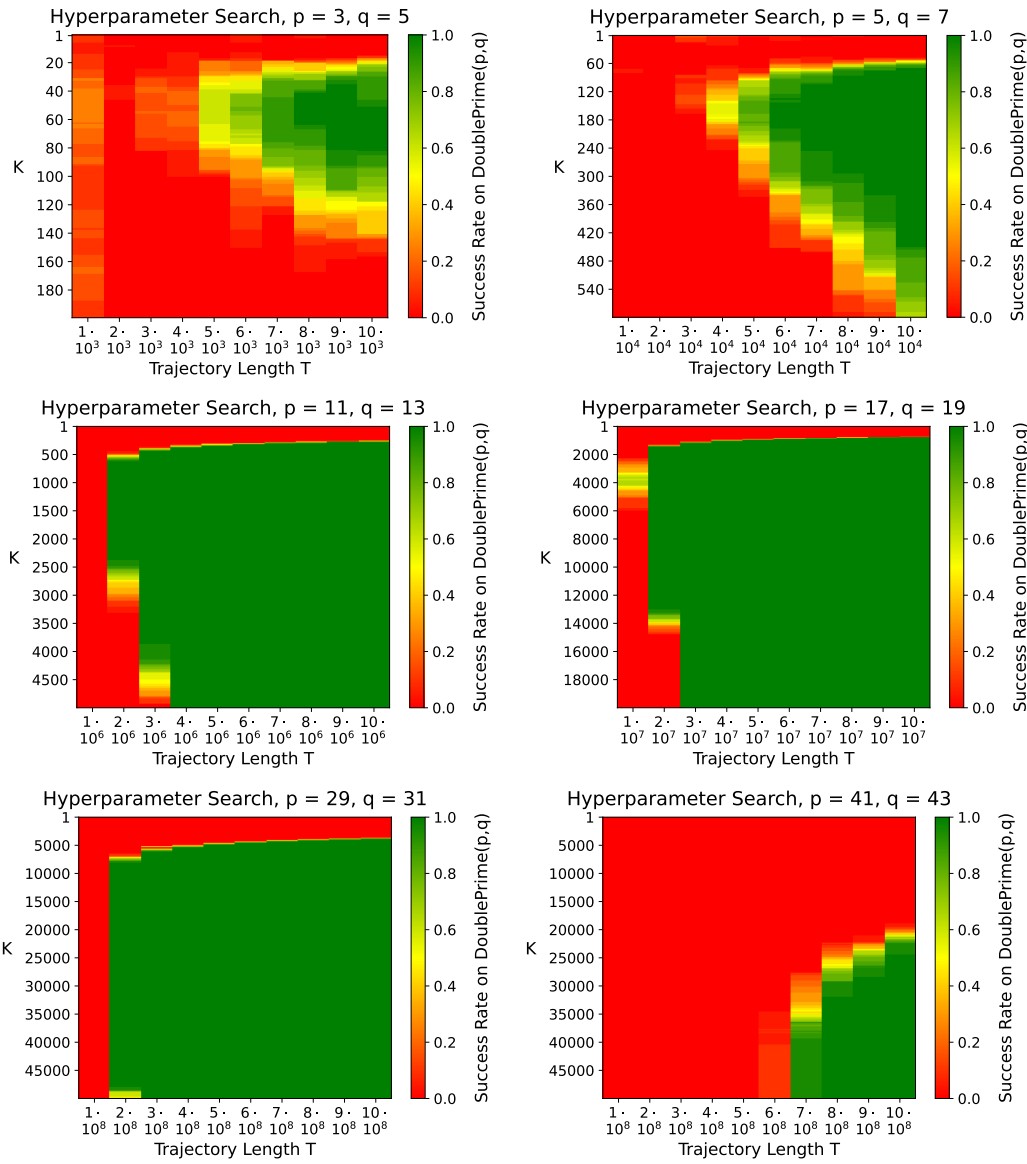

Figure 8: Results from the hyperparameter search for the maximum-step-count parameter $K$ for DoublePrime$(p, q)$, using the active exploration policy described in Section F.3.

