# OpenReview forum: "Learning a Fast Mixing Exogenous Block MDP using a Single Trajectory"
_ICLR.cc/2025/Conference — ICLR 2025 Poster_

### Official Review · Reviewer_1LWB · 2024-10-31

**Soundness:** 3
**Presentation:** 3
**Contribution:** 2
**Rating:** 8
**Confidence:** 4

**Summary:**

This paper studies a structured class of MDPs called an Ex-BMDP, where the latent factors of the observations decompose into a lower-dimensional controllable factor (which evolves deterministically according to the agent's action) and high-dimensional exogenous factor (which evolves independent of actions). This paper focuses on the single-episodic setting and proposes sample-efficient algorithms for learning controllable dynamics of an Ex-BMDP with sample complexity that depends only on the sizes of the low-dimensional controllable state, the encoder function class, and the mixing time of the exogenous noise factor. The paper also empirically tests the proposed STEEL algorithm on the infinite-horizon variations of the "combination lock" and "multi-maze" environments.

**Strengths:**

- The class of Ex-BMDP studied in this paper is a general class of structured POMDPs. It captures problems where, despite having high dimensional observation, the majority of the states are exogenous and only a small controllable state matters for learning. It therefore allows more sample-efficient learning by filtering out the exogenous factors and reducing to a smaller MDP depending only on the controllable states. Such setting fits many applications and gives insight to how to best exploit these hidden structures to optimize learning.
- The main novelty of this paper compared with prior work in Ex-BMDP is that instead of the episodic setting in Efroni et al. (2022) where one gets to reset to starting state, it assumes the agent interacts with the environment in a single episode. This setting is more challenging given it is more difficult to collect samples of a given latent state without the episodic resets.
- The paper also assumes a more general assumption on the state and emission function, where only the partial inverse with respect to the controllable state exists, but places no such assumption on the exogenous states. This is more general than assumption a block structure in prior works which allows a full inverse from observation to state.

**Weaknesses:**

- The proposed algorithm is highly dependent on the assumptions that (1) the dynamics of the latent controllable states is deterministic; (2) the mixing time of the exogenous dynamics. Intuitively, assumption (1) leads to to a cycle of latent states of bounded length that is repeatedly visited and allows repeated collection of the same latent state, which on a high-level is similar to "resetting" the environment; assumption (2), given the looping behavior, can wait out the mixing time of the exogenous dynamics and collect near i.i.d. samples of each latent state. However, assuming deterministic dynamics and bounded mixing time seems restrictive, and possibly does not capture many practical setting. How sensitive is the algorithm to the violation of both assumptions? Does non-deterministic dynamics of the controllable latent states break the proposed algorithm?

**Questions:**

- The paper presumes the availability of an encoder hypothesis class $\mathcal{F}$, where the true decoder $f(x)$ is included, and the final complexity depends on the size $\log\mathcal{F}$. However, it does not seem to specify how to choose this hypothesis class. The simulation section gives an example of $\mathcal{F}$ that is specific to the examples. Is there any general procedure for selecting $\mathcal{F}$ with a reasonable size that also guarantees to include the correct decoder? In general settings beyond the specific examples given in the paper, can you provide guidelines or heuristics for selecting an appropriate hypothesis class?
- Given $\mathcal{F}$, the paper also assumes access to a training oracle that optimally distinguishes two sets of observations (e.g., similar to minimizing 0-1 loss). What would be an example of such an oracle without prior knowledge of the true classifier? And what is the sample/computation cost of constructing such an oracle?
- The STEEL algorithm assumes access to an upper bound on the mixing time $t_{mix}$ for the exogenous dynamics. For a general setting with unknown exogenous latent factors and dynamics, how do you get such an upper bound? Can you discuss potential methods or heuristics for estimating or bounding the mixing time in settings where the exogenous dynamics are not fully known?

---

> ### Author Response · Authors · 2024-11-20
> **Response to Review (#1)**
>
> Reviewer 1LWB,
>
> We thank the reviewer for a providing a substantial and constructive review.
>
> The review mentions two substantial limitations: the assumptions of __determinism__ and __a known upper bound on the mixing time__.
>
> __Determinism__:
> We agree that this is a substantial limitation: we address this concern thoroughly in our Response #1 to Reviewer kqTs.
>
> __Known Upper bound on the mixing time__:
>
>  The review mentions that “ assuming [...] bounded mixing time seems restrictive, and possibly does not capture many practical settings,” and asks how such an upper bound may be obtained by an algorithm. Here, we argue that the assumption that an upper bound on the mixing time is provided  is in fact absolutely necessary. We argue that:
>
> 1.  __Any__ general, provably sample-efficient algorithm for learning, with high probability, an Ex-BMDP from a single trajectory __must necessarily__ require an upper bound on the mixing time of the exogenous noise to be provided _a priori_ (unless some other information about the exogenous noise is provided.)
>
> 2. The runtime of any algorithm for learning an Ex-BMDP must, in the worst case, scale  linearly with the mixing time of the exogenous noise.
>
> At a high level, we argue that, with a single trajectory, it can be impossible to distinguish between a static exogenous dynamics (i.e., where only one exogenous state exists) and an extremely slow-mixing exogenous dynamics, in time sublinear in the mixing time. For any proposed algorithm that does not have access to a bound on the mixing time, and any Ex-BMDP with a single exogenous state (that is, a Block MDP), we can construct an extremely slow-mixing Ex-BMDP that behaves identically to the Block MDP with substantial probability during the entire duration of the runtime of the learning algorithm, but which at equilibrium has a quite different distribution of observations. This will make the learned encoder fail on the Ex-BMDP at equilibrium.
>
> More precisely, to show (1): suppose the converse is true: that there exists some algorithm $A$ that learns Ex-BMDP latent dynamics and state encoders, which is not provided any upper-bound on the mixing time of the exogenous noise of the Ex-BMDP, or any other information about the exogenous noise distribution. We assume that $A$ can learn the correct endogenous dynamics, as well as an encoder with accuracy  (for every endogenous state, under the stationary exogenous distribution) of at least $1-\epsilon$, with probability at least $1-\delta$, for small values of $\epsilon$ and $\delta$.
>
>  Then, consider any Ex-BMDP $M_1$  and related parameterized family of Ex-BMDPs $M_2(\gamma), M_3(\gamma)$ with the following properties:
>
> - $M_1$ has N endogenous latent states $s_1,...,s_N$ with some transition function $T$, a single exogenous latent state $e_1$, and an emission function $Q(s,e_1)$. (In other words, $M_1$ is any Block MDP).
>
> - $M_2(\gamma)$ has the same endogenous states and transition probabilities as $M_1$, but has two exogenous latent states $e_1, e_2$. The state $e_1$ transitions to $e_2$, and $e_2$ transitions to $e_1$, each with probability $\gamma$. Note that the stationary distribution of the exogenous state is uniform over $e_1$ and $e_2$.  We also assume that the initial exogenous state distribution is uniform over $e_1$ and $e_2$. Regardless of $\gamma$, the emission function of $M_2(\gamma)$ is defined such that $\forall s_i, Q_{M_2}(s_i,e_1) = Q_{M_1}(s_i,e_1)$.
>
> - $M_3(\gamma)$ is identical to $M_3(\gamma)$, except that its emission function is defined as  $\forall s_i, Q_{M_3}(s_i,e_1) := Q_{M_1}(s_i,e_1)$, but $\forall s_i, Q_{M_3}(s_i,e_2) := Q_{M_2}(s_{(i+1) \text{ mod } N},e_2)$. In other words, when the exogenous state is equal to $e_2$, the emission distributions for the endogenous states are permuted in $M_3$ compared to $M_2$.
>
> We also assume that the encoder hypothesis class can represent the inverses of $Q_{M_1}$, $Q_{M_2}$, and $Q_{M_3}$. However, note that by construction, any fixed encoder $\phi(\cdot)$ which has accuracy at least $1- \epsilon$ on $M_2(\gamma)$  (for every endogenous state, under the stationary exogenous distribution) can only have accuracy at most $0.5 + \epsilon$ on  $M_3(\gamma)$, because, when the exogenous state is $e_2$, any time that the  encoder returns the correct latent state for $M_2(\gamma)$, it will return the incorrect latent state for $M_3(\gamma)$.
>
> Now, consider what happens when we run the algorithm $A$ on $M_1$. The number of environment steps that $A$ takes on $M_1$ forms some distribution; let $t$ be the 90’th percentile of this distribution, so that with probability 0.9, $A$ stops sampling before step $t$.
>
> (continued)

---

> > ### Author Response · Authors · 2024-11-20
> > **Response to Reviewer #2**
> >
> > Now, we can set $\gamma := 1 - 0.9^{1/t}$, so that, with probability 0.9, the endogenous state of $M_2(\gamma)$ or $M_3(\gamma)$ do not change before step $t$. Therefore, by union bound,  on  $M_2(\gamma)$ or  $M_3(\gamma)$, if the initial exogenous state is $e_1$, then with probability at least 0.8, the exogenous state will stay constant at $e_1$ up to  timestep $t$, and the algorithm $A$, *seeing exactly the same distribution of observations as if the MDP was $M_1$*, will halt before timestep $t$.
> >
> >  Considering the 50% probability that the exogenous state starts at $e_1$, this gives at least a 40% chance that $A$ applied to  $M_2(\gamma)$ or $M_3(\gamma)$ never encounter the exogenous state $e_2$. In this case, the distribution of encoders output by $A$ should be the same for the two Ex-BMDPs: let this distribution be $\mathcal{G}$. (That is, $\mathcal{G}$ is the conditional distribution of encoders output by $A$ when applied to $M_2(\gamma)$ or $M_3(\gamma)$, given that $A$ never encounters $e_2$.)  However, because  $A$ fails on $M_2(\gamma)$ with probability at most $\delta$, we can conclude that at least $(1 - 0.4^{-1} \delta)$ of the encoders from the distribution $\mathcal{G}$ represent *successes* of $A$ on $M_2(\gamma)$.
> >
> > Because $A$ on $M_3(\gamma)$ also draws from  $\mathcal{G}$ with probability at least 0.4, we can conclude that at least $0.4 - \delta$ of the time, $A$ on $M_3(\gamma)$ produces an encoder that is highly accurate on $M_2(\gamma)$. As discussed above, any encoder that is highly-accurate on $M_2(\gamma)$ cannot be highly accurate on  $M_3(\gamma)$, so $A$ must fail on $M_3$ with substantial probability.
> >
> > Therefore, we can conclude that for any algorithm $A$ that has no “hint” about the exogenous dynamics (such as knowing the mixing time), there exists some $\gamma$ such that $A$ cannot possibly succeed with high probability on both $M_2(\gamma)$ and $M_3(\gamma)$.
> >
> > To show point (2), simply note that in this construction, in order to ensure that $e_2$ is observed with probability $(1-\delta)$, an algorithm must observe at least the first $\lceil log(2\delta)/log(1-\gamma) \rceil$ timesteps.  Meanwhile, the mixing time of the two-state Markov chain exogenous state is given by $\lceil -1/log_2(1-2\gamma) \rceil$. For a fixed $\delta$ and as  $\gamma$ approaches 0, the ratio between these quantities approaches a constant: therefore, the number of steps required to ensure that $e_2$ is observed with high probability is linear in the mixing time.
> >
> > If desired, we can include a version of this argument as an appendix to the paper
> >
> > __Novelty__: The review mentions under Weaknesses that the looping behavior of STEEL is "on a high-level is similar to "resetting" the environment". We wish to emphasize that, unlike in the episodic setting, in this single trajectory setting we _do not know_ a priori when the environment has been “reset” (i.e., we don’t know the period of the cycle), or the identity of the state that it will be “reset” to. This makes the algorithmic design in this setting substantially nontrivial.
> >
> > __Assumption of provided hypothesis class and training oracle.__ The assumption of a known hypothesis class that contains the true model that fits the environment is very common in representation learning literature. For example, among representation learning works for Block MDPs, Low-Rank MDPs, and Ex-BMDPs , all of the following works (which represent major works the field) make the same realizability assumption:
> >
> > - (Du et al, 2019, “Provably efficient RL with Rich Observations via Latent State Decoding”; 279 citations; Block MDPs):  Assumption 3.1
> >
> > - (Misra et al, 2020, “Kinematic State Abstraction and Provably Efficient Rich-Observation Reinforcement Learning”; 196 citations; Block MDPs):   Assumption 2
> >
> > - (Uehara et al 2022, “Representation Learning for Online and Offline RL in Low-rank MDPs”; 155 citations; Low Rank MDPs):  Assumption 2
> >
> > - (Agarwal et al 2020, “FLAMBE: Structural complexity and representation learning of low rank MDPs”; 288 citations; Low Rank MDPs): Assumption 1
> >
> > - (Efroni et al 2022: “Provably filtering exogenous distractors using multistep inverse dynamics”; 52 citations [combined over versions of paper]; Ex-BMDPs): Assumption 2
> >
> > (continued)

---

> > > ### Author Response · Authors · 2024-11-20
> > > **Response to Review #3**
> > >
> > > Similarly, these works also assume the existence of training oracles for the function class:
> > >
> > > - (Du et al, 2019): Definition 3.1 (ERM Oracle):  Assumes oracle that can solve (possibly nonlinear) least squares regression.
> > >
> > > - (Misra et al, 2020):  (Section “Computational oracles”): Assumes access to two training oracles: one for least-squares regression and one for offline contextual bandit optimization.
> > >
> > > - (Uehara 2022): Definition 3 (Maximum Likelihood Oracle (MLE)). Assumes oracle which can find the model which maximizes the likelihood of the data
> > >
> > > - (Agarwal et al 2020): Definition 3 (Computational oracles). Requires both a Maximum Likelihood oracle and a sampling oracle
> > >
> > > - (Efroni et al 2022):  Oracle assumption in implicit – in line 5 of Algorithm 1, an argmin of a classification loss is taken over the entire encoder function class.
> > >
> > > The reviewer asks: __“What would be an example of such an oracle without prior knowledge of the true classifier? And what is the sample/computation cost of constructing such an oracle?”__
> > >
> > > Note that the oracle’s “job”  is to find a function $f$ in the function class $\mathcal{F}$ that fits the finite set of samples  that the algorithm provides to it: this is purely a computational challenge and does not require any additional samples. Therefore, there is no “sample cost” associated with the training oracle.  Technically, for *any arbitrary* finite function class, we can easily construct a (highly computationally inefficient) oracle, which simply enumerates over all of the functions and checks which one achieves the minimal loss (or, in our case, fits all of the data). Obviously this is extremely impractical: however, even if this type of “oracle” was used, the sample complexity guarantees of our algorithm would still hold.
> > >
> > > However, in practice, if one does care about computational efficiency, then a function class must be used for which there exist computationally efficient training algorithms. This reduces to the fundamental and well-studied problem of efficient learning of classifiers in computational learning theory. In the paper, we mention linear classifiers as an example of a function class where such an efficient algorithm exists; other examples include k-CNF and k-DNF formulae over features (for fixed k) and Decision Lists. In practice, the choice of hypothesis classes is domain-specific.

---

> > > > ### Comment · Reviewer_1LWB · 2024-11-25
> > > >
> > > > I appreciate the authors for their very thorough and thoughtful responses.
> > > > - The example given under "Known Upper bound on the mixing time" makes a lot of sense and clarifies why the knowledge about mixing time is necessary; I think it'll be helpful to include a version of this argument in the appendix.
> > > > - I thank the authors for providing the references on the known hypothesis class and training oracle assumption. Given that, it seems to me a fair assumption to make under the context of the literature.
> > > >
> > > > I have raised my score to 8.

---

### Official Review · Reviewer_kqTs · 2024-11-03

**Soundness:** 3
**Presentation:** 3
**Contribution:** 3
**Rating:** 8
**Confidence:** 2

**Summary:**

The paper proposes a representation learning method for Ex-BMDP called STEEL. This method identifies the small latent state space of Ex-BMDP - which encodes the essential controllable part of the MDP - while jointly learning an encoder that maps observations to the latent state. Notably, this approach can be applied without requiring "reset" commands, allowing the algorithm to learn from a single trajectory. The key idea is to repeat sequences of actions to detect cycles in the latent state space, which enables the collection of multiple i.i.d. samples to discover the latent space structure. The sample complexity of the algorithm is shown to be polynomial in the size of the latent space, the mixing rate of the Markovian exogenous process, and the complexity of the encoder function class. The algorithm is demonstrated on two problem scenarios.

**Strengths:**

* The paper is clearly written, and the analysis of the key result - specifically, the sample complexity of STEEL being polynomial in the latent space size - is supported by solid mathematical arguments. The algorithm's description is intuitive and effectively conveys its core concepts.
* Furthermore, representation learning from a single episode has been a long-standing interest in the RL community, making this paper's contribution highly relevant to the field.
* The paper provides a comprehensive literature review, effectively demonstrating the novelty of the work and differentiating it from recent existing works.

**Weaknesses:**

* The method relies on several assumptions, particularly concerning the latent state space $\mathcal{S}$. For example, the assumptions of deterministic latent dynamics and the reachability condition of the latent state space are critical for STEEL's CycleFind to function. Addressing these assumptions seems non-trivial, and overcoming them is posed as future work.
* Although the sample complexity of STEEL is polynomial in the size of the latent state space, the numerical simulations show that a substantial number of samples (millions) are required.

**Questions:**

* A discussion of the block assumption on $\mathcal{Q}$ with respect to $\mathcal{S}$ would be helpful. In many practical scenarios, the noisy nature of the emission (or observation) function can make distinguishing between two latent states directly from observations challenging, necessitating filtering techniques. It would be beneficial to clarify whether this assumption is not overly restrictive or if it cannot be easily weakened but is widely adopted.
* Is there a known lower bound for the sample complexity of Ex-BMDP under deterministic latent dynamics? Is STEEL nearly optimal under this assumption, or is there potential for further improvement?

---

> ### Author Response · Authors · 2024-11-20
> **Response To Review (#1)**
>
> Reviewer kqTs,
>
> Thanks for your constructive and thoughtful review. We will respond to most of the points that you raised now, and will follow up with additional responses later in the rebuttal period:
>
> __Restrictive assumptions: determinism and latent-state reachability__:
>
> __Determinism__:
> We agree that the determinism assumption on the endogenous dynamics is restrictive. However, recall that even in the episodic setting with resets to a single latent state, PPE (Efroni et al, 2022: the current state-of-the-art algorithm for efficiently learning Ex-BMDP latent state encoders) requires near-determinism, with only very rare deviations from deterministic behavior (one non-deterministic latent transition every $4|\mathcal{S}|$ *episodes*). In other words, tightening this assumption to full determinism does not represent a major restriction beyond what is found in previous work for an “easier” setting.
>
> Furthermore, we were recently made aware of a new paper (Mhammedi et al., to appear in NeurIPS 2024) which proposes an algorithm for sample-efficient learning in the Ex-BMDP setting with nondeterministic latent dynamics. However, the algorithm proposed in that work requires *simulator access* to the environment, meaning that the agent can reset the environment  to *any previously-observed observation* $x \in \mathcal{X}$. This suggests a general connection between the ability to handle nondeterminstic dynamics and the ability to reset the environment, leading to a “spectrum” of algorithms:
> - Mhammedi et al, 2024: full nondeterminism / reset to arbitrary states
> - Efroni et al, 2022:  near determinism (only very rare departures from determinism allowed) / reset to same state every episode
> - This work: full determinism / no resets, ever
>
> We believe that our work therefore contributes to the understanding of the Ex-BMDP setting in general, by filling out one endpoint of this spectrum. (We have added a reference to Mhammedi et al, 2024 to our paper; see the revision we have added.)
>
> (continued)

---

> > ### Author Response · Authors · 2024-11-20
> > **Response To Review (#2)**
> >
> > __Reachability__: the review mentions “the reachability condition of the latent state space“ as a weakness. Here, we argue that _any_ algorithm for single-trajectory, no-resets representation learning in the Ex-BMDP setting will *necessarily* have this limitation: that all endogenous latent states must be reachable from one another. At a high level, we argue that, if all latent states are not reachable from one another, then an algorithm must visit states in a particular order in order to explore the entire dynamics with a single trajectory. However, because the latent dynamics are not known a priori, it is impossible for an algorithm to guarantee that it visits states in the appropriate order.
> >
> > More formally, recall that the reachability assumption is that for any pair of latent states $s_1,s_2$, there exists both a path from $s_1$ to $s_2$ and a path from $s_2$ to $s_1$ . Consider any Ex-BMDP $M$ where this condition does not hold, such that $|\mathcal{A}| \geq 2$, and any learning algorithm $A$. Firstly, if there is any pair of states $s_1$ and $s_2$ where _neither_ can reach the other, then clearly a single trajectory is insufficient to learn the Ex-BMDP, because it cannot visit both $s_1$ and $s_2$. Therefore, we restrict to the case where, for each pair of states, either both are reachable from each other, or (without loss of generality) $s_2$ is reachable from $s_1$ but $s_1$ is not reachable from $s_2$.
> >
> > Consider every edge $(s,a,s’)$ of the state transition graph defined by the state transition function $T$. If, for all such edges, $s$ is reachable from $s’$, then the reachability assumption holds on the entire dynamics (because all single “steps” are invertible by some sequence of actions, so for any pair $s_1, s_2$, the path from $s_1$ to $s_2$ implies the existence of a path from $s_2$ to $s_1$). Therefore, if reachability does not hold, then there exists some edge $(s,a,s’)$ such that $s$ is not reachable from $s’$. Now, consider the first time that the algorithm $A$ encounters the state $s$. Regardless of the details of $A$, there must exist some action $a’$ such that, on this first encounter, the probability of $A$ taking action $a’$ is at least $1/|\mathcal{A}|$.
> >
> > Therefore, if $a’ = a$, then probability that the algorithm $A$ never revisits $s$, and so never takes any other action from $s$ apart from $a’$, is at least $1/|\mathcal{A}|$. Then, with substantial probability, the algorithm $A$ never explores the $|\mathcal{A}| -1$ other possible transitions from $s$, and cannot possibly learn the full dynamics of the Ex-BMDP.  (To be more precise, the algorithm’s output will not depend *at all* on the ground-truth value of $T(s,a’’)$ for $a’’ \in \mathcal{A} \setminus \\{a\\}$, and so is highly unlikely to return the correct values for these transitions on arbitrary Ex-BMDPs.)
> >
> > Alternatively, if $a’ \neq a$, then consider the alternative Ex-BMDP $M’$,  which is identical to $M$ in every way (in terms of dynamics, exogenous state, emission function, etc.), except that the effects of actions $a$ and $a’$ on the latent state $s$ are swapped. Note that before first  encountering the latent state $s$, the MDPs $M$ and $M’$  will produce identically-distributed sequences of observations, so the algorithm $A$ will behave identically on them, and have identical internal memory/state. Then, when first encountering $s$, the algorithm $M$ on $A$ will take action $a’$ with substantial probability, and then transition to $s’$ and be unable to revisit $s$.
> >
> > Therefore, for any Ex-BMDP $M$ that violates reachability, any algorithm $A$ is either likely to fail on $M$, or to fail on a slightly-modified version of $M$. In any case, no such algorithm will be able to succeed with high probability on any general class of Ex-BMDPs that does not require reachability.
> >
> > If desired, we can add a version of this discussion as an appendix.
> >
> > __Although the sample complexity of STEEL is polynomial in the size of the latent state space, the numerical simulations show that a substantial number of samples (millions) are required.__: As indicated in our response to fyCo, we are currently working on experiments that aim to show whether, on environments with certain structures of latent dynamics, STEEL can empirically outperform prior "practical" methods for controllable state representation learning. We will let you know when these results are ready.
> >
> > - __Block Assumption__: (We will return to this question later in the rebuttal period)
> >
> > - __Known lower bounds__: In our response to 1LWB, we note that at least some sample complexity term linear in $\hat{t}_\text{mix}$ is unavoidable for any algorithm in this setting.  One can also easily see that a term linear in $|\mathcal{A}|$ is also required (because every edge must be explored) and that a term linear in $1/\epsilon$ is also necessary. However, the overall sample complexity, and in particular the factors in $\mathcal{S}$, $N$, and $D$, may be improved in future work.

---

> > > ### Author Response · Authors · 2024-11-24
> > > **Follow-up About the Block Assumption**
> > >
> > > Apologies for the delay in replying to this question:
> > >
> > > The review asks: **A discussion of the block assumption on $\mathcal{Q}$ with respect to $\mathcal{S}$   would be helpful. In many practical scenarios, the noisy nature of the emission (or observation) function can make distinguishing between two latent states directly from observations challenging, necessitating filtering techniques. It would be beneficial to clarify whether this assumption is not overly restrictive or if it cannot be easily weakened but is widely adopted.**
> > >
> > >
> > > The Block assumption is indeed widely adopted in this field (See Du et al., 2019; Misra et al., 2020; Mhammedi et al., 2023; Efroni et al., 2022 and the rest of the cited works in the Related Works section in the Ex-BMDP and Block MDP settings).
> > >
> > > If we entirely  remove the block assumption from the Block MDP setting, it becomes the generic POMDP (Partially-Observed MDP) setting, which is known to **not** admit sample-efficient algorithms  (see Krishnamurthy et al 2016). However, with some additional assumptions, tractable solutions can be obtained. For example, Efroni et al. (2022c) show that if the latent state $\mathcal{S}$ always can be perfectly inferred from the last $L$ observations, then the MDP can be learned with a sample complexity that scales polynomially with $\mathcal{S}$ and does not depend on the size of the observation space $\mathcal{X}$; although it does scale exponentially in $L$. Guo et al (2023) generalize this result to a partially-observed variant of the low-rank MDP setting, and additionally consider an alternative limitation to partial-observability than the $L$-length window assumption: namely, roughly speaking,  that the _distributions_ of observations emitted from different latent states are sufficiently distinct from each other.
> > >
> > > However, in the case of Ex-BMDPs, the problem of partial observability has been less well-studied. To our knowledge, the only relevant prior work is Wu et al, 2024, which makes the sliding-$L$-length-window-decodability assumption.  However, Wu et al. only consider correctness in the limit of infinite samples, not finite sample-complexity bounds.
> > >
> > > It seems plausible that the techniques and assumptions developed in  Guo et al (2023) could be combined with the techniques developed in our work and adapted to the (infinite-horizon) Ex-BMDP setting. Indeed, this may be a promising direction. However, such an adaptation is beyond the scope of  this rebuttal, and is left for future work.
> > >
> > > Additional References:
> > > A. Krishnamurthy, A. Agarwal, and J. Langford. PAC reinforcement learning with rich observations. NeurIPS 2016.
> > >
> > > Efroni Y, Jin C, Krishnamurthy A, Miryoosefi S. Provable reinforcement learning with a short-term memory. ICML 2022c.
> > >
> > > Jiacheng Guo, Zihao Li, Huazheng Wang, Mengdi Wang, Zhuoran Yang, Xuezhou Zhang. Provably Efficient Representation Learning with Tractable Planning in Low-Rank POMDP. ICML 2023.

---

> > > > ### Comment · Reviewer_kqTs · 2024-11-25
> > > > **Response to the Authors**
> > > >
> > > > I appreciate the authors for clearly and concisely explaining the necessity of the assumptions made in this paper and comparing them to the types of assumptions commonly adopted in existing literature. In particular, I believe adding the discussion on the reachability assumption to the appendix would greatly benefit readers, as it provides an insight into the structure of the problem. I have raised my rating of the paper from 6 to 8, recommending the acceptance of the paper.

---

### Official Review · Reviewer_m7A7 · 2024-11-03

**Soundness:** 3
**Presentation:** 3
**Contribution:** 3
**Rating:** 1
**Confidence:** 3

**Summary:**

The authors violated the instructions and reduced the font size substantially for Algorithm 1 and 2. Given they took a whole 10 pages, I decided to recommend desk rejection. If the AC decides differently, please inform me accordingly.

**Strengths:**

NA

**Weaknesses:**

NA

**Questions:**

NA

---

> ### Author Response · Authors · 2024-11-20
> **Response to Review**
>
> Reviewer m7A7,
>
> We apologize for any inconvenience this formatting issue may have caused. However, use of smaller-than-standard fonts in figures, tables, and algorithms seems to be fairly common in ICLR submissions, and font size in figures, etc, was not explicitly mentioned in the formatting instructions.
>
> Regardless, in response to Reviewer fyCo's suggestion, we have moved the algorithms to an appendix (and increased their font size), and incorporated more detail into our high-level description of the algorithms in the main text. This should resolve the issue.
>
> In any case, it seems that this paper was not desk-rejected. If you are willing to provide a substantial review of our paper, we would greatly appreciate the additional feedback. (If not, we ask that you delete your review, so that it is not averaged in to our score in case the AC uses a strict cutoff.)

---

### Official Review · Reviewer_fyCo · 2024-11-03

**Soundness:** 4
**Presentation:** 3
**Contribution:** 4
**Rating:** 8
**Confidence:** 3

**Summary:**

The authors propose Single-Trajectory Exploration for Ex-BMDPs via Looping (STEEL), an algorithm to learn the endogenous (controllable) states in an Exogenous Block Markov Decision Process (Ex-BMDP) when the agent is dealing with one continuous infinite trajectory without resetting to some known states. STEEL achieves this by taking actions that result in a predictable cycle of states and iteratively updating the list of known controllable states and their transitions. They show theoretically the sample complexity and correctness of STEEL with simulations on some small environments.

**Strengths:**

- The introduction and related work highlight this work really well. It explains the existing work nicely and shows where the gaps lie and how this work attempts to extend it.
- The algorithm stands out in terms of the settings it covers compared to existing work. It deals with infinite trajectories, partial observability, and optimization with function approximators all while providing sample complexity guarantees.
- The algorithm itself is designed very well and has a lot of interesting features which include: forcing a cycle of states through the repetition of actions and detecting the unique states in a cycle using a classifier oracle.
- The limitations of the algorithm are clearly discussed with useful insights on how to extend this work in the future.

**Weaknesses:**

- Section 4 can be a bit hard to follow. To quite understand how the algorithm exactly works one has to switch between reading the section text, the pseudocode, and parts of the Appendix. I suggest moving the pseudocode to the appendix and providing further explanation of the algorithm in the main text such that the reader can get a high-level idea of how the Algorithm works from just reading section 4.
- There are parts of the algorithm that are not very intuitive and might require some further discussion. For example, it is mentioned that the dataset $D_0, D_1$ used in the CycleFind subroutine are generated in a way such that they are disjoint if $n'_{cyc}$ is equal to $n_cyc$. Intuitively, how does the selection process achieve this?
- In the experiments section, the authors mention that previous work by Lamb et al.(2023) and Levine et al. (2024) don't have theoretical correctness guarantees, which can be why it seems to have better sample efficiency than STEEL. I suggest also including the percentage of runs where these baselines get the correct states and transition probabilities and how often they fail compared to STEEL which is proven to get it right with high probability. This can add additional value to how STEEL outperforms the baselines in terms of correctness.

**Questions:**

- In the Appendix in equations 33 and 35, could you further explain how the sets $\mathcal{D}_i^\mathcal{A}$ and  $\mathcal{D}_i^\mathcal{B}$ are constructed?

---

> ### Author Response · Authors · 2024-11-20
> **Response To Review (#1)**
>
> Reviewer fyCo,
>
> Thank you for your constructive review. We will respond to parts of your review now, and we aim to follow up with additional empirical results as they become available. Note that we have already submitted a revision incorporating the suggested changes to the paper.
>
> - *"Section 4 can be a bit hard to follow.... I suggest moving the pseudocode to the appendix and providing further explanation of the algorithm in the main text such that the reader can get a high-level idea of how the Algorithm works from just reading section 4."*: We have now done this: the pseudocode is now in the Appendix, and we have substantially expanded the text of Section 4 to describe the algorithm in a more self-contained manner.
>
> -  *"There are parts of the algorithm that are not very intuitive and might require some further discussion. For example, it is mentioned that the dataset $D_0$, $D_1$ used in the CycleFind subroutine are generated in a way such that they are disjoint if  $n_{cyc}$ is equal to $n_{cyc}'$ . Intuitively, how does the selection process achieve this?":* We have added an in-depth explanation of how $D_0$ and $D_1$ are constructed to Section 4 of the paper.
>
> - *"In the experiments section, the authors mention that previous work by Lamb et al.(2023) and Levine et al. (2024) don't have theoretical correctness guarantees, which can be why it seems to have better sample efficiency than STEEL. I suggest also including the percentage of runs where these baselines get the correct states and transition probabilities and how often they fail compared to STEEL which is proven to get it right with high probability. This can add additional value to how STEEL outperforms the baselines in terms of correctness."* For the particular "multi-maze" environment in the paper, (for which we mention that Lamb et al. (2023) and Levine et al. (2024)'s algorithms have better empirical sample efficiency), Levine et al. (2024) also show that both algorithms learn the dynamics consistently, each succeeding on 20/20 random seeds. We noted in the paper that this strong performance may be partially due to the particular choice of neural network architecture chosen by Lamb et al. (2023), which has a spatial prior that favors focusing on a single maze (as also referenced by Levine et al; in Footnote 8.)
>
> However, even if we accept that STEEL underperforms  Lamb et al. (2023) and Levine et al. (2024)'s algorithms on this particular "multi-maze" environment, the benefit of STEEL is that it is _guaranteed_ to perform within a known sample-complexity bound, on _any_ environment that meets its assumptions. To underscore this point, we are currently running experiments involving a specific family of tabular Ex-BMDPs that were introduced by Levine et al. (2024), which have controllable latent dynamics that are particularly difficult for either of  Lamb et al. (2023) or Levine et al. (2024)'s algorithms to learn efficiently. Preliminarily, we have been able to show that, in apples-to-apples comparisons, STEEL empirically outperforms these prior methods on environments with this particular structure of controllable latent dynamics. We hope to have results ready to include in the paper by the end of the rebuttal period.
>
> - *"In the Appendix in equations 33 and 35, could you further explain how the sets  $\mathcal{D}^A_i$ and  $\mathcal{D}^B_i$ are constructed?"* We have added an additional figure (Figure 4) with an extensive caption, that explains how these datasets are constructed.

---

> > ### Comment · Reviewer_fyCo · 2024-11-26
> >
> > I appreciate the author's detailed response and modifications to enhance readability. I believe the message is a lot clearer now and consequently, I have updated my score. I would also appreciate if the authors add the extra experiments if the time allows for it.

---

### Author Response · Authors · 2024-11-20
**First Rebuttal Revision**

We thank all of the reviewers for their hard work. We have made initial rebuttals to review comments and submitted a revised manuscript, that includes a substantial expansion of the text of Section 4.

Additionally, while preparing the rebuttal, we noticed a (minor) flaw in the algorithm as it was implemented. We have corrected this, rerun all experiments, and corrected the reported results. The change to the results was minor, and does not affect our conclusions in any way.

---

### Author Response · Authors · 2024-11-28
**Updated Revision**

Dear Reviewers,

We have submitted an updated revision of our paper, with the following improvements:

- At reviewers kqTs and 1LWB's suggestion, we have added our discussion of the necessity of the reachability assumption and known bound on mixing time to an appendix (Appendix E).

- As mentioned to reviewer kqTs and fyCo, we have added an additional experiment, on a family of tabular  Ex-BMDPs which have controllable latent dynamics that are particularly difficult for either of Lamb et al. (2023) or Levine et al. (2024)'s algorithms to learn efficiently; we show that STEEL can outperform these methods empirically on such environments. (Appendix F)

---

### Meta-Review · Area_Chair_FLQc · 2024-12-23

**Metareview:**

### Summarization
This paper addresses the single continuous trajectory learning problem within the framework of Exogenous Block Markov Decision Process (Ex-BMDP) while the prior works mainly focus on episodic setting. The authors propose the STEEL algorithm and theoretically demonstrate the sample-efficiency of  STEEL by leveraging the structure of the Ex-BMDP setting.

### Strengths
* This paper is well structured and discusses the limitations of the proposed algorithm
* The setting of this paper presents chanllenging in collecting data, and the authors solve this problem in an ingenious manner
* This paper provides solid theoretical bounds

###  Weaknesses
* The assumptions in this paper seem too strong

Overall, this paper addresses an important problem in this field and is a good paper.

**Additional Comments On Reviewer Discussion:**

* Reviewers kqTs and 1LWB challenged that the assumptions in this paper are too strong, and the authors added a discussion about this problem in Appendix E.
* Reviewer kqTs and Reviewer fyCo suggested that the authors include experiments demonstrating that the proposed algorithm outperforms previous work. The authors updated it in Appendix F.

---

### Decision · Program_Chairs · 2025-01-22

Accept (Poster)